# A Dynamical Central Limit Theorem
# for Shallow Neural Networks

**Zhengdao Chen**[†]     **Grant M. Rotskoff**[∗]     **Joan Bruna**[†‡]     **Eric Vanden-Eijnden**[†]

## Abstract

Recent theoretical works have characterized the dynamics of wide shallow neural networks trained via gradient descent in an asymptotic mean-field limit when the width tends towards infinity. At initialization, the random sampling of the parameters leads to deviations from the mean-field limit dictated by the classical Central Limit Theorem (CLT). However, since gradient descent induces correlations among the parameters, it is of interest to analyze how these fluctuations evolve. In this work, we derive a dynamical CLT to prove that the asymptotic fluctuations around the mean limit remain bounded in mean square throughout training. The upper bound is given by a Monte-Carlo resampling error, with a variance that depends on the 2-norm of the underlying measure, which also controls the generalization error. This motivates the use of this 2-norm as a regularization term during training. Furthermore, if the mean-field dynamics converges to a measure that interpolates the training data, we prove that the asymptotic deviation eventually vanishes in the CLT scaling. We also complement these results with numerical experiments.

## 1 Introduction

Theoretical analyses of neural networks aim to understand their computational and statistical advantages seen in practice. On the computation side, the training of neural networks often succeed despite being a non-convex optimization problem known to be hard in certain settings [41, 31, 20]. On the statistics side, neural networks often generalize well despite having large numbers of parameters [70, 8]. In this context, the notion of *over-parametrization* has been useful, by providing insights into the optimization and generalization properties as the network widths tend to infinity [36, 21, 2, 4, 63, 67, 38]. In particular, under appropriate scaling, one can view shallow (a.k.a. single-hidden-layer or two-layer) networks as interacting particle systems that admit a mean-field limit. Their training dynamics can then be studied as Wasserstein Gradient Flows [47, 51, 12, 58], leading to global convergence guarantees in the mean-field limit under certain assumptions. On the statistics side, such an approach lead to powerful generalization guarantees for learning high-dimensional functions with hidden low-dimensional structures, as compared to learning in Reproducing Kernel Hilbert Spaces (RKHS) [5, 30]. However, since ultimately we are concerned with neural networks of finite width, it is key to study the deviation of finite-width networks from their infinite-width limits, and how it scales with the width $m$. At the random initial state, neurons do not interact and therefore a standard Monte-Carlo (MC) argument shows that the fluctuations in the underlying measure scale as $m^{-1/2}$, which we refer to as the Central Limit Theorem (CLT) scaling. As optimization introduces complex dependencies among the parameters, the key question is to understand how the fluctuation evolves during training. To make this investigation tractable, we aim to obtain insight on an asymptotic scale as the width grows, and focus on the evolution in time. An application of

---

∗: Department of Chemistry, Stanford University
†: Courant Institute of Mathematical Sciences, New York University
‡: Center for Data Science, New York University
Correspondence to: zc1216@nyu.edu, rotskoff@stanford.edu, bruna@nyu.edu and eve2@nyu.edu.

Grönwall's inequality shows that this asymptotic deviation remains bounded at all finite time [46], but the dependence on time is exponential, making it difficult to assess the long-time behavior.

The main focus of this paper is to investigate this question in-depth, by analyzing the interplay between the deviations from the mean-field limit and the gradient flow dynamics. First, we prove a dynamical CLT to capture how the fluctuations away from the mean-field limit evolve as a function of training time to show that the fluctuations remain on the initial $m^{-1/2}$-scale for all finite times. Next, we examine the long-time behavior of the fluctuations, proving that, in several scenarios, the long-time fluctuations are controlled by the error of Monte-Carlo resampling from the limiting measure. We focus on two main setups relevant for supervised learning and scientific computing: the unregularized case with global convergence of mean-field gradient flows to minimizers that interpolate the data, and the regularized case where the limiting measure has atomic support and is nondegenerate. In the former setup, we prove particularly that the fluctuations eventually vanish in the CLT scaling. These asymptotic predictions are complemented by empirical results in a teacher-student model.

**Related Works:**    This paper continues the line of work initiated in [47, 12, 51, 58] that studies optimization of over-parameterized shallow neural networks under the mean-field scaling. Global convergence for the unregularized setting is discussed in [47, 46, 58, 51]. In the regularized setting, [12] establishes global convergence in the mean-field limit under specific homogeneity conditions on the neuron activation. Other works that study asymptotic properties of wide neural networks include [29, 28, 6, 23, 34, 35, 69, 43, 1], notably investigating the transition between the so-called *lazy* and *active* regimes [14], corresponding respectively to linear versus nonlinear learning. Our focus is on the dynamics under the mean-field scaling, which encompasses the active, nonlinear regime.

A relevant work concerning the sparse optimization of measures is [11], where under a different metric for gradient flow and additional assumptions on the nature of the minimizer, it can be established that fluctuations vanish for sufficiently large $m$. Our results are only asymptotic in $m$ but apply to broader settings in the context of shallow neural networks. Concerning the next-order deviations of finite neural networks from their mean-field limit, [51] show that the scale of fluctuations is below that of MC resampling for unregularized problems using non-rigorous arguments. [60] provides a CLT for the fluctuations at finite time under stochastic gradient descent (SGD) and proves that the fluctuations decay in time in the case where there is a single critical point in the parameter space. Our focus is on the long-time behavior of the fluctuations in more general settings. Another relevant topic is the propagation of chaos in McKean-Vlasov systems, which study the deviations of randomly-forced interacting particle systems from their infinite-particle limits [10, 66, 65, 7]. In particular, a line of work provides uniform-in-time bounds to the fluctuations in various settings [19, 16, 55, 56, 22], but the conditions are not applicable to shallow neural networks. Concurrently to our work, [17] studies quantitative propagation of chaos of shallow neural networks trained by SGD, but the bound grows exponentially in time, and therefore cannot address the long-time behavior of the fluctuations.

Learning with neural networks exhibits the phenomenon that generalization error can decrease with the level of overparameterization [8, 64]. [48] proposes a bias-variance decomposition that contains a variance term initialization in optimization. They show in experiments that this term decreases as the width of the network increases, and justifies this theoretically under the strong assumption that model parameters remain Gaussian-distributed in the components that are irrelevant for the task, which does not hold in the scenario we consider, for example. [27] provides scaling arguments for the dependence of this term on the width of the network. Our work provides a more rigorous analysis of the dependence of this term on the width of the network and training time.

## 2   Background

### 2.1   Shallow Neural Networks and the Integral Representation

On a data space $\Omega \subseteq \mathbb{R}^d$, we consider parameterized models of the following form

$$f^{(m)}(\boldsymbol{x}) = \frac{1}{m} \sum_{i=1}^{m} \varphi(\boldsymbol{\theta}_i, \boldsymbol{x}), \tag{1}$$

where $\boldsymbol{x} \in \Omega$, $\{\boldsymbol{\theta}_i\}_{i=1}^{m} \subseteq D$ is the set of model parameters, and $\varphi : D \times \Omega \to \mathbb{R}$ is the activation function. Of particular interest are shallow neural network models, which admit a more specific form:

**Assumption 2.1 (Shallow neural networks setting)** $D = \mathbb{R} \times \hat{D}$, $\boldsymbol{\theta} = (c, \boldsymbol{z}) \in D$, and $\varphi(\boldsymbol{\theta}, \boldsymbol{x}) = c\hat{\varphi}(\boldsymbol{z}, \boldsymbol{x})$ with $\hat{\varphi} : \hat{D} \times \Omega \to \mathbb{R}$. Thus, (1) can be rewritten as $f^{(m)}(\boldsymbol{x}) = \frac{1}{m} \sum_{i=1}^{m} c_i \hat{\varphi}(\boldsymbol{z}_i, \boldsymbol{x})$.

As many of our results hold for general models of the form (1), we will invoke Assumption 2.1 only when needed. We shall also assume the following:

**Assumption 2.2** $\Omega$ *is compact;* $D$ *is an Euclidean space (or a subset thereof);* $\varphi(\boldsymbol{\theta}, \boldsymbol{x})$ *is twice differentiable in* $\boldsymbol{\theta}$; $\nabla_{\boldsymbol{\theta}} \nabla_{\boldsymbol{\theta}} \varphi(\boldsymbol{\theta}, \boldsymbol{x})$ *is Lipschitz in* $\boldsymbol{\theta}$, *uniformly in* $\boldsymbol{x}$.

The regularity assumptions are standard in the literature [11, 37, 10]. We note that they are not satisfied by ReLU units (i.e., $\hat{\varphi}(\boldsymbol{z}, \boldsymbol{x}) = \max\{0, \langle \boldsymbol{a}, \boldsymbol{x} \rangle + b\}$, where $\boldsymbol{z} = [\boldsymbol{a}, b]^{\mathsf{T}}$, with $\boldsymbol{a} \in \mathbb{R}^d$ and $b \in \mathbb{R}$), though prior work [12, 13] has considered differentiable approximations of these models.

As observed in [47, 12, 51, 58, 24], a model of the form (1) can be expressed in integral form in terms of a probability measure over $D$ as $f^{(m)} = f[\mu^{(m)}]$, where we define

$$f[\mu](\boldsymbol{x}) = \int_D \varphi(\boldsymbol{\theta}, \boldsymbol{x})\mu(d\boldsymbol{\theta}) , \qquad \mu^{(m)}(d\boldsymbol{\theta}) = \frac{1}{m} \sum_{i=1}^{m} \delta_{\boldsymbol{\theta}_i}(d\boldsymbol{\theta}) . \qquad (2)$$

Suppose we are given a dataset $\{(\boldsymbol{x}_l, y_l)\}_{l=1}^{n}$, which can be represented by an empirical data measure $\hat{\nu} = \frac{1}{n} \sum_{l=1}^{n} \delta_{\boldsymbol{x}_l}$, and $y_l = f_*(\boldsymbol{x}_l)$ are generated by an target function $f_*$ that we wish to estimate using least-squares regression. A canonical approach to this regression task is to consider an Empirical Risk Minimization (ERM) problem of the form

$$\min_{\mu \in \mathcal{P}(D)} \mathcal{L}(\mu) \quad \text{with} \quad \mathcal{L}(\mu) := \frac{1}{2} \|f[\mu] - f_*\|_{\hat{\nu}}^2 + \lambda \int_D r(\boldsymbol{\theta})\mu(d\boldsymbol{\theta}) . \qquad (3)$$

where $\mathcal{P}(D)$ is the space of probability measures on $D$, $\|f - f_*\|_{\hat{\nu}}^2 = \int_\Omega |f(\boldsymbol{x}) - f_*(\boldsymbol{x})|^2 \hat{\nu}(d\boldsymbol{x})$ denotes the function reconstruction error averaged over the data, and $\lambda \int_D r(\boldsymbol{\theta})\mu(d\boldsymbol{\theta})$ is some optional regularization term. While we can allow $r$ to be a general convex function, in Appendix F we will motivate a choice of $r$ in the shallow neural networks setting that is related to the variation norm [5] or Barron norm [44] of functions.

## 2.2 Approximation and Optimization with a Finite Number of Neurons

Integral representations with a probability measure such as those defined in (2) are amenable to efficient approximation in high dimensions via Monte-Carlo sampling. Namely, if the parameters $\boldsymbol{\theta}_i$ in $f^{(m)}$ are drawn i.i.d. from an underlying measure $\mu$ on $D$, then by the Law of Large Numbers (LLN), the resulting empirical measure $\mu^{(m)}$ converges $\mu$ almost surely, and moreover,

$$\mathbb{E}_{\mu^{(m)}} \|f[\mu^{(m)}] - f[\mu]\|_{\hat{\nu}}^2 = \frac{1}{m} \left( \int_D \|\varphi(\boldsymbol{\theta}, \cdot)\|_{\hat{\nu}}^2 \mu(d\boldsymbol{\theta}) - \|f[\mu]\|_{\hat{\nu}}^2 \right) , \qquad (4)$$

Such a Monte-Carlo estimator showcases the benefit of normalized integral representations for high-dimensional approximation, as the ambient dimension appears in the rate of approximation only through the term $\int_D \|\varphi(\boldsymbol{\theta}, \cdot)\|_{\hat{\nu}}^2 \mu(d\boldsymbol{\theta})$. In the case of shallow neural networks, this is connected to the variation norm or Barron norm of the function we wish to approximate [5, 44].

While the Monte-Carlo sampling strategy above can be seen as a 'static' approximation of a function representable as (2), it also gives rise to an efficient algorithm to optimize (3). Indeed, in terms of the empirical distribution $\mu^{(m)}$, the loss $\mathcal{L}(\mu^{(m)})$ becomes function of the parameters $\{\boldsymbol{\theta}_i\}_{i=1}^{m}$:

$$L(\boldsymbol{\theta}_1, \ldots, \boldsymbol{\theta}_m) = \frac{1}{2} \|f^{(m)} - f_*\|_{\hat{\nu}}^2 + \frac{\lambda}{m} \sum_{i=1}^{m} r(\boldsymbol{\theta}_i) . \qquad (5)$$

In the shallow neural network setting, with suitable choices of the function $r$, the regularization term corresponds to *weight decay* over the parameters.

## 2.3 From Particle to Wasserstein Gradient Flows

Expanding (5), we get

$$L(\boldsymbol{\theta}_1, \ldots, \boldsymbol{\theta}_m) = C_{f_*} - \frac{1}{m} \sum_{i=1}^{m} F(\boldsymbol{\theta}_i) + \frac{1}{2m^2} \sum_{i,j=1}^{m} K(\boldsymbol{\theta}_i, \boldsymbol{\theta}_j), \qquad (6)$$

where we have defined $C_f = \frac{1}{2}\|f\|_{\hat{\nu}}^2$, and

$$F(\boldsymbol{\theta}) = \int_\Omega f_*(\boldsymbol{x})\varphi(\boldsymbol{\theta}, \boldsymbol{x})\hat{\nu}(d\boldsymbol{x}) - \lambda r(\boldsymbol{\theta}), \qquad K(\boldsymbol{\theta}, \boldsymbol{\theta}') = \int_\Omega \varphi(\boldsymbol{\theta}, \boldsymbol{x})\varphi(\boldsymbol{\theta}', \boldsymbol{x})\hat{\nu}(d\boldsymbol{x}) . \quad (7)$$

Performing GD on $L$ amounts to discretizing in time the following ODE system for $\{\boldsymbol{\theta}_i\}_{i=1}^m$:

$$\dot{\boldsymbol{\theta}}_i = -m\partial_{\boldsymbol{\theta}_i} L(\boldsymbol{\theta}_1 \dots \boldsymbol{\theta}_m) = \nabla F(\boldsymbol{\theta}_i) - \frac{1}{m}\sum_{j=1}^m \nabla K(\boldsymbol{\theta}_i, \boldsymbol{\theta}_j) =: -\nabla V(\boldsymbol{\theta}_i, \mu_t^{(m)}). \quad (8)$$

where we defined the potential

$$V(\boldsymbol{\theta}, \mu) = -F(\boldsymbol{\theta}) + \int_D K(\boldsymbol{\theta}, \boldsymbol{\theta}')\mu(d\boldsymbol{\theta}') . \quad (9)$$

Heuristically, the 'particles' $\boldsymbol{\theta}_i$ perform GD according to the potential $V(\boldsymbol{\theta}, \mu_t^{(m)})$ which itself evolves, depending on the particles positions through their empirical measure. Such dynamics can also be expressed in terms of the empirical measure via the *continuity equation*:

$$\partial_t \mu_t^{(m)} = \nabla \cdot (\nabla V(\boldsymbol{\theta}, \mu_t^{(m)})\mu_t^{(m)}) \quad (10)$$

This equation should be understood in the weak sense by testing it against continuous functions $\chi : D \to \mathbb{R}$, and it can be interpreted as the gradient flow on the loss defined in (3) under the 2-Wasserstein metric [12, 51, 47, 58]. This insight provides powerful analytical tools to understand convergence properties, by considering the mean-field limit when $m \to \infty$.

## 2.4 Law of Large Numbers and Mean-Field Gradient Flow

From now on, we assume that the particle gradient flow is initialized in the following way:

**Assumption 2.3** *The ODE (8) is solved for the initial condition $\boldsymbol{\theta}_i(0) = \boldsymbol{\theta}_i^0$, with $\boldsymbol{\theta}_i^0$ drawn i.i.d. from a compactly supported measure $\mu_0 \in \mathcal{P}(D)$ for each $i = 1, \dots, m$.*

We use $\mathbb{P}_0$ to denote the probability measure associated with the set $\{\boldsymbol{\theta}_i^0\}_{i \in \mathbb{N}}$ with each $\boldsymbol{\theta}_i^0$ drawn i.i.d. from $\mu_0$, and use $\mathbb{E}_0$ to denote the expectation under $\mathbb{P}_0$. The Law of Large Numbers (LLN) indicates that $\mathbb{P}_0$-almost surely, $\mu_t^{(m)} \rightharpoonup \mu_t$ as $m \to \infty$, where $\mu_t$ satisfies the mean-field gradient flow [52, 12, 47, 61]:

$$\partial_t \mu_t = \nabla \cdot (\nabla V(\boldsymbol{\theta}, \mu_t)\mu_t) , \qquad \mu_{t=0} = \mu_0 . \quad (11)$$

The solution to this equation should be understood via the representation formula

$$\int_D \chi(\boldsymbol{\theta})\mu_t(d\boldsymbol{\theta}) = \int_D \chi(\boldsymbol{\Theta}_t(\boldsymbol{\theta}))\mu_0(d\boldsymbol{\theta}) , \quad (12)$$

where $\chi$ is a continuous test function $\chi : D \to \mathbb{R}$ and $\boldsymbol{\Theta}_t : D \to D$ is the *characteristic flow* associated with (10), which in direct analogy with (8) solves

$$\dot{\boldsymbol{\Theta}}_t(\boldsymbol{\theta}) = -\nabla V(\boldsymbol{\Theta}_t(\boldsymbol{\theta}), \mu_t), \qquad \boldsymbol{\Theta}_0(\boldsymbol{\theta}) = \boldsymbol{\theta} . \quad (13)$$

Using expression (9) for $V$ as well as (12), this equation can be written in closed form explicitly as

$$\dot{\boldsymbol{\Theta}}_t(\boldsymbol{\theta}) = \nabla F(\boldsymbol{\Theta}_t(\boldsymbol{\theta})) - \int_D \nabla K(\boldsymbol{\Theta}_t(\boldsymbol{\theta}), \boldsymbol{\Theta}_t(\boldsymbol{\theta}'))\mu_0(d\boldsymbol{\theta}'), \qquad \boldsymbol{\Theta}_0(\boldsymbol{\theta}) = \boldsymbol{\theta} . \quad (14)$$

It is easy to see that this equation is itself a gradient flow since it is the continuous-time limit of a proximal scheme (mirror descent), as stated in Appendix B.

## 2.5 Long-Time Properties of the Mean-Field Gradient Flow

In the shallow neural networks setting, a series of earlier work [12, 51, 47, 58] has established that under certain assumptions $\mu_t$ will converge to a global minimizer of the loss functional $\mathcal{L}$. In particular, [12] studies global convergence for the regularized loss $\mathcal{L}$ under homogeneity assumptions on $\hat{\varphi}$, and [50] considers modified dynamics using *double-lifting*. Here, to study the long time behavior of the fluctuations, we will often work with the following weaker assumptions:

**Assumption 2.4** *The solution to* (14) *exists for all time, and has a limit:*

$$\boldsymbol{\Theta}_t \to \boldsymbol{\Theta}_\infty \quad \mu_0\text{-almost surely as } t \to \infty. \tag{15}$$

**Assumption 2.5** *The limiting $\boldsymbol{\Theta}_\infty$ is a local minimizer of* (59).

With these assumptions, we have

**Proposition 2.6** *Under Assumptions* 2.3 *and* 2.4, *we have*

$$\cup_{t \geq 0} \operatorname{supp} \mu_t = \cup_{t \geq 0} \{\boldsymbol{\Theta}_t(\boldsymbol{\theta}) \, : \, \boldsymbol{\theta} \in \operatorname{supp} \mu_0\} \text{ is compact,} \tag{16}$$

*and $\mu_t \rightharpoonup \mu_\infty$ weakly as $t \to \infty$, with $\mu_\infty$ satisfying*

$$\int_D \chi(\boldsymbol{\theta}) \mu_\infty(d\boldsymbol{\theta}) = \int_D \chi(\boldsymbol{\Theta}_\infty(\boldsymbol{\theta})) \mu_0(d\boldsymbol{\theta}), \tag{17}$$

*for all continuous test function $\chi : D \to \mathbb{R}$. Additionally, if Assumption* 2.5 *also holds, then*

$$\nabla\nabla V(\boldsymbol{\Theta}_\infty(\boldsymbol{\theta}), \mu_\infty) \text{ is positive semidefinite for } \mu_0\text{-almost all } \boldsymbol{\theta} \tag{18}$$

We prove this proposition in Appendix C. Here, $\nabla\nabla V(\boldsymbol{\Theta}_\infty(\boldsymbol{\theta}), \mu_\infty)$ denotes

$$\nabla\nabla V(\boldsymbol{\Theta}_\infty(\boldsymbol{\theta}), \mu_\infty) = -\nabla\nabla F(\boldsymbol{\Theta}_\infty(\boldsymbol{\theta})) + \int_D \nabla\nabla K(\boldsymbol{\Theta}_\infty(\boldsymbol{\theta}), \boldsymbol{\Theta}_\infty(\boldsymbol{\theta}')) \mu_0(d\boldsymbol{\theta}'), \tag{19}$$

which will appear in Section 3.2 for studying the fluctuations from the mean-field limit in long time.

**Remark 2.7** *Assumptions* 2.4 *and* 2.5 *impose conditions on the initial measure $\mu_0$ [51, 47, 12]. While the convergence of gradient flows in finite-dimensional Euclidean space to local minimizers is guaranteed under mild assumptions [62, 39], its infinite-dimensional counterpart, Assumption* 2.5, *may require further technical assumptions, left for future study. Also, while Assumption* 2.4 *implies that $\mu_\infty$ is a stationary point of* (11), *Assumption* 2.5 *does not imply that $\mu_\infty$ minimizes $\mathcal{L}$.*

## 3 Fluctuations from Mean-Field Gradient Flow

The main goal of this section is to characterize the deviations of finite-width shallow networks from their mean-field evolution, by first deriving an estimate for $f_t^{(m)} - f_t$ for $t \geq 0$ (Section 3.1), and then analyzing its long-time properties (Section 3.2). The bound on the long-time fluctuations derived in Section 3.2 motivates a choice of the regularization in (3), which is also connected to generalization via the variation norm or Barron norm of functions [5, 44], as we discuss in Appendix F.

### 3.1 A Dynamical Central Limit Theorem

Let us start by defining

$$g_t^{(m)} := m^{1/2} \big( f_t^{(m)} - f_t \big). \tag{20}$$

By the static Central Limit Theorem (CLT) we know that, if we draw the initial values of the parameters $\boldsymbol{\theta}_i$ independently from $\mu_0$ as specified in Assumption 2.3, $g_{t=0}^{(m)}$ has a limit as $m \to \infty$, leading to estimates similar to (4) with $\mu^{(m)}$ and $\mu$ replaced by the initial $\mu_0^{(m)}$ and $\mu_0$, respectively. For $t > 0$, however, this estimate is not preserved by the gradient flow: the static CLT no longer applies and needs to be replaced by a dynamical variant [10, 66, 65, 60]. Next, we derive this dynamical CLT in the context of neural network optimization.

To this end let us define the discrepancy measure $\omega_t^{(m)}$ such that

$$\int_D \chi(\boldsymbol{\theta}) \omega_t^{(m)}(d\boldsymbol{\theta}) := m^{1/2} \int_D \chi(\boldsymbol{\theta}) \left( \mu_t^{(m)}(d\boldsymbol{\theta}) - \mu_t(d\boldsymbol{\theta}) \right), \tag{21}$$

for any continuous test function $\chi : D \to \mathbb{R}$. We can then represent $g_t^{(m)}$ in terms of $\omega_t^{(m)}$ as

$$g_t^{(m)} = \int_D \varphi(\boldsymbol{\theta}, \cdot) \omega_t^{(m)}(d\boldsymbol{\theta}). \tag{22}$$

Hence, we will first establish how the limit of $\omega_t^{(m)}$ as $m \to \infty$ evolves over time. This can be done upon noting that the representation formula (12) implies that

$$\int_D \chi(\boldsymbol{\theta})\omega_t^{(m)}(d\boldsymbol{\theta}) = m^{1/2} \int_D \left( \chi(\boldsymbol{\Theta}_t^{(m)}(\boldsymbol{\theta}))\mu_0^{(m)}(d\boldsymbol{\theta}) - \chi(\boldsymbol{\Theta}_t(\boldsymbol{\theta}))\mu_0(d\boldsymbol{\theta}) \right) , \qquad (23)$$

where $\boldsymbol{\Theta}_t^{(m)}$ solves (14) with $\mu_0$ replaced by $\mu_0^{(m)}$. Defining

$$\boldsymbol{T}_t^{(m)}(\boldsymbol{\theta}) = m^{1/2}\left( \boldsymbol{\Theta}_t^{(m)}(\boldsymbol{\theta}) - \boldsymbol{\Theta}_t(\boldsymbol{\theta}) \right) , \qquad (24)$$

we can write (23) as

$$\begin{aligned}
\int_D \chi(\boldsymbol{\theta})\omega_t^{(m)}(d\boldsymbol{\theta}) &= \int_D \chi(\boldsymbol{\Theta}_t(\boldsymbol{\theta}))\omega_0^{(m)}(d\boldsymbol{\theta}) \\
&+ \int_0^1 \int_D \nabla\chi\big(\boldsymbol{\Theta}_t(\boldsymbol{\theta}) + m^{-1/2}\eta \boldsymbol{T}_t^{(m)}(\boldsymbol{\theta})\big) \cdot \boldsymbol{T}_t^{(m)}(\boldsymbol{\theta})\mu_0^{(m)}(d\boldsymbol{\theta})d\eta .
\end{aligned} \qquad (25)$$

As shown in Appendix D.1, we can take the limit $m \to \infty$ of this formula to obtain:

**Proposition 3.1 (Dynamical CLT - I)** *Under Assumptions 2.2 and 2.3, $\forall t \geq 0$, as $m \to \infty$ we have $\omega_t^{(m)} \rightharpoonup \omega_t$ weakly in law with respect to $\mathbb{P}_0$, where $\omega_t$ is such that given a test function $\chi : D \to \mathbb{R}$,*

$$\int_D \chi(\boldsymbol{\theta})\omega_t(d\boldsymbol{\theta}) = \int_D \chi(\boldsymbol{\Theta}_t(\boldsymbol{\theta}))\omega_0(d\boldsymbol{\theta}) + \int_D \nabla\chi(\boldsymbol{\Theta}_t(\boldsymbol{\theta})) \cdot \boldsymbol{T}_t(\boldsymbol{\theta})\mu_0(d\boldsymbol{\theta}) . \qquad (26)$$

*Here $\omega_0$ is the Gaussian measure with mean zero and covariance*

$$\mathbb{E}_0 \left[ \omega_0(d\boldsymbol{\theta})\omega_0(d\boldsymbol{\theta}') \right] = \mu_0(d\boldsymbol{\theta})\delta_{\boldsymbol{\theta}}(d\boldsymbol{\theta}') - \mu_0(d\boldsymbol{\theta})\mu_0(d\boldsymbol{\theta}') , \qquad (27)$$

*where $\mathbb{E}_0$ denotes expectation over $\mathbb{P}_0$, and $\boldsymbol{T}_t = \lim_{m\to\infty} m^{1/2}(\boldsymbol{\Theta}_t^{(m)} - \boldsymbol{\Theta}_t)$ is the flow solution to*

$$\begin{aligned}
\dot{\boldsymbol{T}}_t(\boldsymbol{\theta}) &= -\nabla\nabla V(\boldsymbol{\Theta}_t(\boldsymbol{\theta}), \mu_t)\boldsymbol{T}_t(\boldsymbol{\theta}) - \int_D \nabla\nabla' K(\boldsymbol{\Theta}_t(\boldsymbol{\theta}), \boldsymbol{\Theta}_t(\boldsymbol{\theta}'))\boldsymbol{T}_t(\boldsymbol{\theta}')\mu_0(d\boldsymbol{\theta}') \\
&- \int_D \nabla K(\boldsymbol{\Theta}_t(\boldsymbol{\theta}), \boldsymbol{\Theta}_t(\boldsymbol{\theta}'))\omega_0(d\boldsymbol{\theta}')
\end{aligned} \qquad (28)$$

*with initial condition $\boldsymbol{T}_0 = 0$ and where $\boldsymbol{\Theta}_t$ solves (13) and $\nabla\nabla V(\boldsymbol{\Theta}_t(\boldsymbol{\theta}), \mu_t)$ is a shorthand for*

$$\nabla\nabla V(\boldsymbol{\Theta}_t(\boldsymbol{\theta}), \mu_t) = -\nabla\nabla F(\boldsymbol{\Theta}_t(\boldsymbol{\theta})) + \int_D \nabla\nabla K(\boldsymbol{\Theta}_t(\boldsymbol{\theta}), \boldsymbol{\Theta}_t(\boldsymbol{\theta}'))\mu_0(d\boldsymbol{\theta}') . \qquad (29)$$

A direct consequence of this proposition and formula (22) is:

**Corollary 3.2** *Under Assumptions 2.2 and 2.3, $\forall t \geq 0$, as $m \to \infty$ we have $g_t^{(m)} \to g_t$ pointwise in law with respect to $\mathbb{P}_0$, where $g_t$ is given in terms of the limiting measure $\omega_t$ or the flow $\boldsymbol{T}_t$ as*

$$g_t = \int_D \varphi(\boldsymbol{\theta}, \cdot)\omega_t(d\boldsymbol{\theta}) = \int_D \varphi(\boldsymbol{\Theta}_t(\boldsymbol{\theta}), \cdot)\omega_0(d\boldsymbol{\theta}) + \int_D \nabla\varphi(\boldsymbol{\Theta}_t(\boldsymbol{\theta}), \cdot) \cdot \boldsymbol{T}_t(\boldsymbol{\theta})\mu_0(d\boldsymbol{\theta}) . \qquad (30)$$

It is interesting to comment on the origin of both terms at the right hand side of (26) and, consequently, (30). The first term captures the deviations induced by fluctuations of $\mu_0^{(m)}$ around $\mu_0$ assuming that the flow $\boldsymbol{\Theta}_t^{(m)}$ is unaffected by these fluctuations, and remains equal to $\boldsymbol{\Theta}_t$. In particular, this term is the one we would obtain if we were to resample $\mu_t^{(m)}$ from $\mu_t$ at every $t \geq 0$, i.e. use $\bar{\mu}_t^{(m)} = m^{-1} \sum_{i=1}^m \delta_{\bar{\boldsymbol{\theta}}_t^i}$ with $\{\bar{\boldsymbol{\theta}}_t^i\}_{i=1}^m$ sampled i.i.d. from $\mu_t$, so that $\boldsymbol{\Theta}_t^{(m)}$ is identical to $\boldsymbol{\Theta}_t$ in (23). In this case, the limiting discrepancy measure $\bar{\omega}_t$ would simply be given by

$$\int_D \chi(\boldsymbol{\theta})\bar{\omega}_t(d\boldsymbol{\theta}) = \int_D \chi(\boldsymbol{\Theta}_t(\boldsymbol{\theta}))\omega_0(d\boldsymbol{\theta}) , \qquad (31)$$

while the associated deviation in the represented function would read

$$\bar{g}_t = \int_D \varphi(\boldsymbol{\theta}, \cdot)\bar{\omega}_t(d\boldsymbol{\theta}) = \int_D \varphi(\boldsymbol{\Theta}_t(\boldsymbol{\theta}), \cdot)\omega_0(d\boldsymbol{\theta}) . \qquad (32)$$

The second term at right hand side of (26) and (30) captures the deviations to the flow $\Theta_t$ in (14) induced by the perturbation of $\mu_0$, i.e. how much $\Theta_t^{(m)}$ differs from $\Theta_t$ in (23). In the limit as $m \to \infty$, these deviations are captured by the solution $T_t$ to (28), as is apparent from (25).

The difference between $g_t$ and $\bar{g}_t$ can also be quantified via the following Volterra equation, which can be derived from Proposition 3.1 and relates the evolution of $g_t$ to that of $\bar{g}_t$.

**Corollary 3.3 (Dynamical CLT - II)** *Under Assumptions 2.2 and 2.3, $\forall t \geq 0$, pointwise on $\Omega$, we have $g_t^{(m)} \to g_t$ in law with respect to $\mathbb{P}_0$ as $m \to \infty$, where $g_t$ solves the Volterra equation*

$$g_t(\boldsymbol{x}) + \int_0^t \int_\Omega \Gamma_{t,s}(\boldsymbol{x}, \boldsymbol{x}') g_s(\boldsymbol{x}') \hat{\nu}(d\boldsymbol{x}') ds = \bar{g}_t(\boldsymbol{x}) . \tag{33}$$

*Here $\bar{g}_t$ is given in (32) and we defined*

$$\Gamma_{t,s}(\boldsymbol{x}, \boldsymbol{x}') = \int_D \langle \nabla_{\boldsymbol{\theta}} \varphi(\Theta_t(\boldsymbol{\theta})), J_{t,s}(\boldsymbol{\theta}) \nabla_{\boldsymbol{\theta}} \varphi(\Theta_s(\boldsymbol{\theta})) \rangle \mu_0(d\boldsymbol{\theta}) , \tag{34}$$

*where $J_{t,s}$ is the solution to*

$$\frac{d}{dt} J_{t,s}(\boldsymbol{\theta}) = -\nabla\nabla V(\Theta_t(\boldsymbol{\theta}), \mu_t) J_{t,s}(\boldsymbol{\theta}), \qquad J_{s,s}(\boldsymbol{\theta}) = Id . \tag{35}$$

This corollary is proven in Appendix D.2. In a nutshell, (33) can be established using Duhamel's principle on (28) by considering all terms at the right hand side except the first as the source term (hence the role of $J_{t,s}$) and inserting the result in (30).

## 3.2 Long-Time Behavior of the Fluctuations

Next, we study the long-time behavior of $g_t$ and, in particular, evaluate

$$\lim_{t \to \infty} \mathbb{E}_0 \|g_t\|_\nu^2 = \lim_{t \to \infty} \lim_{m \to \infty} m\mathbb{E}_0 \|f_t^{(m)} - f_t\|_\nu^2. \tag{36}$$

This limit quantifies the asymptotic approximation error of $f_t^{(m)}$ around its mean field limit $f_t$ after gradient flow, i.e. if we take $m \to \infty$ first, then $t \to \infty$ – taking these limits in opposite order is of interest too but is beyond the scope of the present paper. Our main result is to show that, under certain assumptions to be specified below, the limit in (36) is not only finite but necessarily upper-bounded by $\lim_{t \to \infty} \mathbb{E}_0 \|\bar{g}_t\|_\nu^2$ with $\bar{g}_t$ given in (32). That is, the approximation error at the end of training is no higher than that obtained by resampling the mean-field measure $\mu_\infty$ defined in Proposition 2.6.

It is useful to start by considering an idealized case, namely when the initial conditions are sampled as in Assumption 2.3 with $\mu_0 = \mu_\infty$. In that case, there is no evolution at mean field level, i.e. $\Theta_t(\boldsymbol{\theta}) = \Theta_\infty(\boldsymbol{\theta}) = \boldsymbol{\theta}$, $\mu_t = \mu_\infty$, and $f_t = f_\infty = \int_D \varphi_\infty(\boldsymbol{\theta}, \cdot) \mu_\infty(d\boldsymbol{\theta})$, but the CLT fluctuations still evolve. In particular, it is easy to see that the Volterra equation in (33) for $g_t$ becomes

$$g_t(\boldsymbol{x}) + \int_0^t \int_\Omega \Gamma_{t-s}^\infty(\boldsymbol{x}, \boldsymbol{x}') g_s(\boldsymbol{x}') \hat{\nu}(d\boldsymbol{x}') ds = \bar{g}_\infty(\boldsymbol{x}) . \tag{37}$$

Here $\Gamma_{t-s}^\infty(\boldsymbol{x}, \boldsymbol{x}')$ is the Volterra kernel obtained by solving (35) with $\nabla\nabla V(\Theta_t(\boldsymbol{\theta}), \mu_t)$ replaced by $\nabla\nabla V(\boldsymbol{\theta}, \mu_\infty)$ and inserting the result in (34) with $\Theta_t(\boldsymbol{\theta}) = \boldsymbol{\theta}$ and $\mu_0 = \mu_\infty$,

$$\Gamma_{t-s}^\infty(\boldsymbol{x}, \boldsymbol{x}') = \int_D \langle \nabla_{\boldsymbol{\theta}} \varphi(\boldsymbol{\theta}, \boldsymbol{x}), e^{-(t-s)\nabla\nabla V(\boldsymbol{\theta}, \mu_\infty)} \nabla_{\boldsymbol{\theta}} \varphi(\boldsymbol{\theta}, \boldsymbol{x}') \rangle \mu_\infty(d\boldsymbol{\theta}) , \tag{38}$$

and $\bar{g}_\infty$ is the Gaussian field with variance

$$\mathbb{E}_0 \|\bar{g}_\infty\|_\nu^2 = \int_D \|\varphi(\boldsymbol{\theta}, \cdot)\|_\nu^2 \mu_\infty(d\boldsymbol{\theta}) - \|f_\infty\|_\nu^2 . \tag{39}$$

From (18) in Proposition 2.6 we know that $\nabla\nabla V(\boldsymbol{\theta}, \mu_\infty)$ is positive semidefinite for $\mu_\infty$-almost all $\boldsymbol{\theta}$. As a result, we prove in E.1 that the Volterra kernel (38) viewed as an operator on functions defined on $\Omega \times [0, T]$ is positive semidefinite. Therefore, we have

$$\int_0^T \|g_t\|_\nu^2 dt \leq \int_0^T \|g_t\|_\nu^2 dt + \int_0^T \int_0^t \int_{\Omega \times \Omega} g_t(\boldsymbol{x}) \Gamma_{t-s}^\infty(\boldsymbol{x}, \boldsymbol{x}') g_s(\boldsymbol{x}') \hat{\nu}(d\boldsymbol{x}) \hat{\nu}(d\boldsymbol{x}') ds dt$$

$$= \int_0^T \mathbb{E}_{\hat{\nu}}(g_t \bar{g}_\infty) dt \leq T^{1/2} \|\bar{g}_\infty\|_{\hat{\nu}} \left( \int_0^T \|g_t\|_\nu^2 dt \right)^{1/2} . \tag{40}$$

Together with (39), this implies that

**Theorem 3.4** *Under Assumptions [2.2], [2.3], [2.4] and [2.5], with $\mu_0 = \mu_\infty$ and $\mu_\infty$ as specified in Proposition [2.6], we have*

$$\lim_{T \to \infty} \frac{1}{T} \int_0^T \mathbb{E}_0 \|g_t\|_{\hat\nu}^2 dt \leq \int_D \|\varphi(\boldsymbol{\theta}, \cdot)\|_{\hat\nu}^2 \mu_\infty(d\boldsymbol{\theta}) - \|f_\infty\|_{\hat\nu}^2 \ . \tag{41}$$

This theorem indicates that, if we knew $\mu_\infty$ and could sample initial conditions for the parameters from it, it would still be favorable to train these parameters as this would reduce the approximation error. Of course, in practice we have no *a priori* access to $\mu_\infty$, and so the relevant question is whether (41) also holds if we sample initial conditions from any $\mu_0$ such that Proposition [2.6] holds.

In light of (30), one way to address this question is to study the long-time behavior of $\boldsymbol{T}_t$. In the setup without regularization ($\lambda = 0$), we can do so by leveraging existing results that, under certain assumptions, the mean-field gradient flow converges to a global minimizer which interpolates the training data points exactly [52, 12, 47, 60]. In this case, the following theorem shows that we can obtain stronger controls on the fluctuations than (41), which we prove in Appendix [E.2].

**Theorem 3.5 (Long-time fluctuations in the unregularized case)** *Consider the ERM setting with $\lambda = 0$ and under Assumptions [2.2], [2.3] and [2.4]. Suppose that as $t \to \infty$, $\mu_t$ converges to a global minimizer $\mu_\infty$ that interpolates the data, i.e. the function $f_\infty = \int_D \varphi(\boldsymbol{\theta}, \cdot) \mu_\infty(d\theta)$ satisfies*

$$\forall \boldsymbol{x} \in \operatorname{supp} \hat\nu \ : \ f_\infty(\boldsymbol{x}) = f_*(\boldsymbol{x}) \ , \tag{42}$$

*and, furthermore, the convergence satisfies*

$$\int_0^\infty |\mathcal{L}(\mu_t)|^{1/2} \, dt < \infty \tag{43}$$

*Then (41) holds. Additionally,*

1. *if Assumption [2.1] also holds, i.e., in the shallow neural network setting, we further have*

$$\lim_{T \to \infty} \frac{1}{T} \int_0^T \mathbb{E}_0 \|g_t\|_{\hat\nu}^2 dt = 0 \ ; \tag{44}$$

2. *if $\mu_0 = \mu_\infty$, then $\|g_t\|_{\hat\nu}$ decreases monotonically in $t$.*

Hence, in the shallow neural networks setting and under these assumptions, the fluctuations will eventually vanish in the $O(m^{-1/2})$ scale of CLT. For (43) to hold, it is sufficient that $\mathcal{L}(\mu_t)$ decays at an asymptotic rate of $O(t^{-\alpha})$ with $\alpha > 2$. We leave the search for weaker conditions for future work.

When the limiting measure $\mu_\infty$ does not necessarily interpolate the training data, we can proceed with curvature assumptions in two ways. One one hand, with Theorem [E.8] in Appendix [E.3], we prove that (41) holds under an assumption on the long-time behavior of the curvature, $\nabla\nabla V(\boldsymbol{\theta}, \mu_t)$. On the other hand, in the regularized ($\lambda > 0$) ERM setting, we can obtain the following result when the support of $\mu_\infty$ is atomic, as expected on general grounds [71, 26, 5, 9, 18]:

**Theorem 3.6 (Long-time fluctuations in the regularized case)** *Consider the ERM setting under Assumptions [2.2], [2.3] and [2.4]. Suppose further that as $t \to \infty$, $\mu_t$ converges to $\mu_\infty$ satisfying*

$$\exists \sigma > 0 \ \text{s.t.} \ \forall \boldsymbol{\theta} \in \operatorname{supp} \mu_\infty \ : \ \nabla\nabla V(\boldsymbol{\theta}, \mu_\infty) \succ \sigma Id \ , \ \text{and} \tag{45}$$

$$\Theta_t \ \text{admits an asymptotic uniform convergence rate of } O(t^{-\alpha}) \ \text{with } \alpha > 3/2. \tag{46}$$

*Then (41) holds with the "$\lim$" replaced by "$\limsup$" on its LHS.*

Theorem [3.6] is proven in Appendix [E.4] by analyzing directly the Volterra equation (33) and establishing that its solution coincides with that of (37) in the limit as $t \to \infty$, a property that we also expect to hold more generally than under the assumptions of Theorem [3.6]. In fact, we prove in Appendix [E.4] that (46) can be replaced by a weaker condition, (231). We also discuss the relation between Theorem [3.6] and the work of [11] in Appendix [E.5].

# 4   Numerical Experiments

In Figure 1, we show the results of a student-teacher experiment whose setup is described in Appendix G.1. We observe from *Column 2* that the average fluctuation of the mean-squared training loss indeed remains at a $m^{-1}$ scaling with a general tendency to decay over time. Moreover, consistently with (44) in Theorem 3.5, the fluctuation vanishes during training in the unregularized case, and hence also the training loss. Further discussions and additional experiments that consider training under the exact population loss, different initializations of the parameters as well as a non-planted target function are presented in Appendix G.

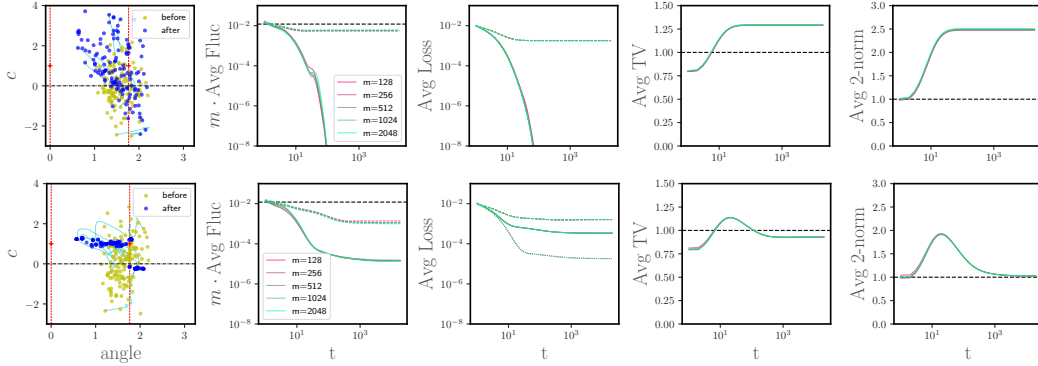

Figure 1: Results of the experiments where student shallow neural networks of different widths are trained to learn teacher networks of width 2 under the *empirical* loss. *Row 1:* using unregularized square loss; *Row 2:* using regularized square loss with $\lambda = 0.01$. In each row, *Column 1* plots the trajectory of the neurons, $\boldsymbol{\theta}_i = (c_i, \boldsymbol{z}_i)$, of a student network with width $m = 128$ during its training, with $x$-coordinate being the angle between $\boldsymbol{z}_i$ and that of a chosen teacher's neurons and $y$-coordinate being $c_i$. The yellow dots, blue dots and cyan curves mark their initial values, terminal values, and trajectory during training. *Columns 2-5* plot the average fluctuation (rescaled by $m$), average loss, average TV norm, and average 2-norm during training, respectively, computed across $\kappa = 20$ runs with different random initializations of the student network for each choice of $m$. These quantities are defined in Appendix G.1. In *Column 2*, the *solid* curves give the average fluctuation in the *training* loss, the *dashed* curves give the average fluctuation in the *population* loss computed analytically via spherical integrals, and the black horizontal *dashed* line gives an approximate value of the asymptotic Monte-Carlo bound in (41) computed in Appendix H for this setting. In *Column 3*, the *solid* curves give the total *training* loss, the *dotted* curves give the unregularized *training* loss (for the regularized case only), and the *dashed* curves give the unregularized *population* loss. In *Columns 4* and *5*, the black horizontal *dashed* line give the relevant norm of the teacher network.

# 5   Conclusions

We studied the deviation of shallow neural networks from their infinite-width limit, especially how this deviation evolves during gradient flow. In the ERM setting, we establish that under different sets of conditions, the long-term deviation under the Central Limit Theorem (CLT) scaling is controlled by a Monte Carlo (MC) resampling error, giving width-asymptotic guarantees that do not depend on the data dimension explicitly. The MC resampling bound motivates a choice of regularization that is also connected to generalization via the variation-norm function spaces.

Our results thus seem to paint a favorable picture for high-dimensional learning, in which the optimization and generalization guarantees for the idealized mean-field limit could be transferred to their finite-width counterparts. However, our results are still asymptotic, in that we take limits both in the width and time. In the face of negative results for the computational efficiency of training shallow networks [45, 41, 54, 20, 31], an important challenge is to leverage additional structure in the problem (such as the empirical data distribution [32], or the structure of the minimisers [18]) to provide nonasymptotic versions of our results, along the lines of [11] or [40]. Finally, another clear direction for future research is to extend our techniques to deep neural architectures, in light of recent works that consider deep or residual models [3, 59, 49, 42, 68, 25].

## Broader Impact

Our work contributes to the theoretical guarantees of neural network models, which are critical to one day extend their applications to the broad spectrum of scientific computing. If successful, deep learning with theoretical guarantees could transform scientific computing in domains where efficient high-dimensional function estimation is critical, such as molecular dynamics, climate science, or computational drug design.

## Acknowledgements

This work benefited from discussions with Lenaic Chizat and Carles Domingo-Enrich. Z.C. acknowledges support from the Henry MacCraken Fellowship. G.M.R. acknowledges support from the James S. McDonnell Foundation. J.B. acknowledges support from the Alfred P. Sloan Foundation, NSF RI-1816753, NSF CAREER CIF 1845360, and the Institute for Advanced Study. E. V.-E. acknowledges support by National Science Foundation (NSF) Materials Research Science and Engineering Center Program Grant No. DMR-1420073, and by NSF Grant No. DMS-1522767.

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
