[Supplementary Material]

# Contents

# Appendix

## A   Notations

We will use $\nabla\varphi(\boldsymbol{\theta},\boldsymbol{x})$ and $\nabla\nabla\varphi(\boldsymbol{\theta},\boldsymbol{x})$ to denote $\nabla_{\boldsymbol{\theta}}\varphi(\boldsymbol{\theta},\boldsymbol{x})$ and $\nabla_{\boldsymbol{\theta}}\nabla_{\boldsymbol{\theta}}\varphi(\boldsymbol{\theta},\boldsymbol{x})$, respectively. We will use $\nabla K(\boldsymbol{\theta},\boldsymbol{\theta}')$ to denote $\nabla_{\boldsymbol{\theta}}K(\boldsymbol{\theta},\boldsymbol{\theta}')$, $\nabla\nabla K(\boldsymbol{\theta},\boldsymbol{\theta}')$ to denote $\nabla_{\boldsymbol{\theta}}\nabla_{\boldsymbol{\theta}}K(\boldsymbol{\theta},\boldsymbol{\theta}')$, $\nabla'\nabla K(\boldsymbol{\theta},\boldsymbol{\theta}')$ to denote $\nabla_{\boldsymbol{\theta}'}\nabla_{\boldsymbol{\theta}}K(\boldsymbol{\theta},\boldsymbol{\theta}')$, and $\nabla'\nabla'K(\boldsymbol{\theta},\boldsymbol{\theta}')$ to denote $\nabla_{\boldsymbol{\theta}'}\nabla_{\boldsymbol{\theta}'}K(\boldsymbol{\theta},\boldsymbol{\theta}')$. We will write $V_t(\cdot)$ for $V(\cdot,\mu_t)$ and $V_\infty(\cdot)$ for $V(\cdot,\mu_\infty)$.

Let $D' = \cup_{t>0}\operatorname{supp}\mu_t$. Under Assumption 2.4 and Proposition 2.6, $D'$ is bounded, and we denote its diameter by $|D'|$. We will use $C_\varphi$, $C_{\nabla\varphi}$ and $C_{\nabla\nabla\varphi}$ to denote the supremum of $|\varphi(\boldsymbol{\theta},\boldsymbol{x})|$, $|\nabla\varphi(\boldsymbol{\theta},\boldsymbol{x})|$ and $|\nabla\nabla\varphi(\boldsymbol{\theta},\boldsymbol{x})|$ over $\boldsymbol{\theta}\in D'$ and $\boldsymbol{x}\in\operatorname{supp}\hat{\nu}$, which are all finite under Assumptions 2.2 and the boundedness of $D'$. We will use $L_{\nabla\nabla\varphi}$ to denote the (uniform-in-$\boldsymbol{x}$) Lipschitz constant of $\nabla\nabla\varphi(\boldsymbol{\theta},\boldsymbol{x})$ in $\boldsymbol{\theta}$, which is also finite under Assumption 2.2.

The following notations will be used in Appendix E.2: Assuming that $D$ is Euclidean (under Assumption 2.2), let $\mathcal{V}(D)$ denote the space of random vector fields on $D$. It becomes a Hilbert space once equipped with the inner product

$$\langle\boldsymbol{\xi}_1,\boldsymbol{\xi}_2\rangle_0 := \mathbb{E}_0\int_D\boldsymbol{\xi}_1(\boldsymbol{\theta})\cdot\boldsymbol{\xi}_2(\boldsymbol{\theta})\mu_0(d\boldsymbol{\theta}), \tag{47}$$

where $\boldsymbol{\xi}_1,\boldsymbol{\xi}_2$ denotes two random vector fields in $\mathcal{V}(D)$. This inner product gives rise to the norm

$$\|\boldsymbol{\xi}\|_0^2 := \mathbb{E}_0\int_D|\boldsymbol{\xi}(\boldsymbol{\theta})|^2\mu_0(d\boldsymbol{\theta}) \ . \tag{48}$$

For each $t$, we define $\boldsymbol{b}_t\in\mathcal{V}(D)$ as

$$\boldsymbol{b}_t(\boldsymbol{\theta}) = \int_D\nabla K(\boldsymbol{\Theta}_t(\boldsymbol{\theta}),\boldsymbol{\Theta}_t(\boldsymbol{\theta}'))\omega_0(d\boldsymbol{\theta}') \tag{49}$$

which depends on the random measure $\omega_0$. We define two linear operators, $\mathcal{A}_t^{(K)}$ and $\mathcal{A}_t^{(V)}$ on $\mathcal{V}(D)$, as

$$(\mathcal{A}_t^{(K)}\boldsymbol{\xi})(\boldsymbol{\theta}) = \int_D\nabla'\nabla K(\boldsymbol{\Theta}_t(\boldsymbol{\theta}),\boldsymbol{\Theta}_t(\boldsymbol{\theta}'))\boldsymbol{\xi}(\boldsymbol{\theta}')\mu_0(d\boldsymbol{\theta}') \tag{50}$$

$$= \int_\Omega\nabla\varphi(\boldsymbol{\Theta}_t(\boldsymbol{\theta}),\boldsymbol{x})\Big(\int_D\nabla\varphi(\boldsymbol{\Theta}_t(\boldsymbol{\theta}'),\boldsymbol{x})^\mathsf{T}\boldsymbol{\xi}(\boldsymbol{\theta}')\mu_0(d\boldsymbol{\theta}')\Big)\hat{\nu}(d\boldsymbol{x}) \ , \tag{51}$$

$$(\mathcal{A}_t^{(V)}\boldsymbol{\xi})(\boldsymbol{\theta}) = \nabla\nabla V(\boldsymbol{\Theta}_t(\boldsymbol{\theta}),\mu_t)\boldsymbol{\xi}(\boldsymbol{\theta}) \ , \tag{52}$$

for $\boldsymbol{\xi}\in\mathcal{V}(D)$. Under Assumption 2.4, we also define $\boldsymbol{b}_\infty$, $\mathcal{A}_\infty^{(K)}$, and $\mathcal{A}_\infty^{(V)}$ similarly by replacing $\boldsymbol{\Theta}_t(\cdot)$ with $\boldsymbol{\Theta}_\infty(\cdot)$.

Let $\mathcal{W}(\Omega)$ denote the space of random functions on $\Omega$. For a fixed set of data points $\{\boldsymbol{x}_l\}_{l=1}^n$, it becomes a Hilbert space once equipped with the inner product

$$\langle\eta_1,\eta_2\rangle_{\hat{\nu},0} := \mathbb{E}_0\int_\Omega\eta_1(\boldsymbol{x})\eta_2(\boldsymbol{x})\hat{\nu}(d\boldsymbol{x}) = \frac{1}{n}\mathbb{E}_0\sum_{l=1}^n\eta_1(\boldsymbol{x}_l)\eta_2(\boldsymbol{x}_l) \ , \tag{53}$$

which gives rise to the norm

$$\|\eta\|_{\hat{\nu},0}^2 := \langle \eta, \eta \rangle_{\hat{\nu},0} = \mathbb{E}_0 \|\eta\|_{\hat{\nu}}^2 . \tag{54}$$

With an abuse of notation, we will consider elements in $\mathcal{W}(\Omega)$ equivalently as random vectors on $\mathbb{R}^n$. Next, we can define $\mathcal{B}_t$ to be the operator that maps $\eta \in \mathcal{W}(\Omega)$ into the vector field

$$(\mathcal{B}_t \eta)(\boldsymbol{\theta}) = \int_\Omega \nabla\varphi(\boldsymbol{\Theta}_t(\boldsymbol{\theta}), \boldsymbol{x})\eta(\boldsymbol{x})\hat{\nu}(d\boldsymbol{x}) \tag{55}$$

in $\mathcal{V}(D)$. Its transpose is

$$(\mathcal{B}_t^{\mathsf{T}} \boldsymbol{\xi})(\boldsymbol{x}) = \int_D \nabla\varphi(\boldsymbol{\Theta}_t(\boldsymbol{\theta}), \boldsymbol{x})\boldsymbol{\xi}(\boldsymbol{\theta})\mu_0(d\boldsymbol{\theta}), \tag{56}$$

which maps a vector field $\boldsymbol{\xi} \in \mathcal{V}(D)$ back into $\mathcal{W}(\Omega)$.

## B  Proximal Scheme, Gradient Flow and Mirror Descent

**Proposition B.1**  *Given $\bar{\boldsymbol{\Theta}}_0(\boldsymbol{\theta}) = \boldsymbol{\theta}$ and $\tau > 0$, for $p \in \mathbb{N}$ let $\boldsymbol{\Theta}_{p\tau}$ be specified via*

$$\bar{\boldsymbol{\Theta}}_{p\tau} \in \operatorname{argmin}\left( \frac{1}{2\tau}\|\boldsymbol{\Theta} - \bar{\boldsymbol{\Theta}}_{(p-1)\tau}\|_0^2 + \mathcal{E}(\boldsymbol{\Theta}) , \right) \tag{57}$$

*where we defined*

$$\|\boldsymbol{\Theta}\|_0^2 = \int_D |\boldsymbol{\Theta}(\boldsymbol{\theta})|^2 \mu_0(d\boldsymbol{\theta}) \tag{58}$$

*and*

$$\mathcal{E}(\boldsymbol{\Theta}) = -\int_D F(\boldsymbol{\Theta}(\boldsymbol{\theta}))\mu_0(d\boldsymbol{\theta}) + \frac{1}{2}\int_D K(\boldsymbol{\Theta}(\boldsymbol{\theta}), \boldsymbol{\Theta}(\boldsymbol{\theta}'))\mu_0(d\boldsymbol{\theta})\mu_0(d\boldsymbol{\theta}') . \tag{59}$$

*Then*

$$\lim_{\tau \to 0} \bar{\boldsymbol{\Theta}}_{\lfloor t/\tau \rfloor \tau} = \boldsymbol{\Theta}_t \qquad \mu_0\text{-almost surely} , \tag{60}$$

*where $\boldsymbol{\Theta}_t$ solves (14).*

## C  Long-Time Properties of the Mean-Field Gradient Flow

*Proof of Proposition 2.6:* The compactness of $\cup_{t \geq 0} \operatorname{supp} \mu_t$ follows from (15) and the compactness of $\operatorname{supp} \mu_0$ assumed in Assumption 2.3. $\mu_t \rightharpoonup \mu_\infty$ follows from (12) and (15).

Under Assumption 2.4, $\boldsymbol{\Theta}_\infty$ is a local minimizer of the energy $\mathcal{E}$ defined in (59). Consider a local perturbation $\epsilon\boldsymbol{\Theta}_\Delta$ to $\boldsymbol{\Theta}$. The energy value after the perturbation is

$$\mathcal{E}(\boldsymbol{\Theta}_\infty + \epsilon\boldsymbol{\Theta}_\Delta) = -\int_D F(\boldsymbol{\Theta}_\infty(\boldsymbol{\theta}) + \epsilon\boldsymbol{\Theta}_\Delta(\boldsymbol{\theta}))\mu_0(d\boldsymbol{\theta})$$
$$+ \frac{1}{2}\int_D \int_D K(\boldsymbol{\Theta}_\infty(\boldsymbol{\theta}) + \epsilon\boldsymbol{\Theta}_\Delta(\boldsymbol{\theta}), \boldsymbol{\Theta}_\infty(\boldsymbol{\theta}') + \epsilon\boldsymbol{\Theta}_\Delta(\boldsymbol{\theta}'))\mu_0(d\boldsymbol{\theta}')\mu_0(d\boldsymbol{\theta}') . \tag{61}$$

Under Assumptions 2.2, using Taylor expansion, we have

$$F(\boldsymbol{\Theta}_\infty(\boldsymbol{\theta}) + \epsilon\boldsymbol{\Theta}_\Delta(\boldsymbol{\theta})) = F(\boldsymbol{\Theta}_\infty(\boldsymbol{\theta})) + \epsilon\nabla F(\boldsymbol{\Theta}_\infty(\boldsymbol{\theta})) \cdot \boldsymbol{\Theta}_\Delta(\boldsymbol{\theta})$$
$$+ \frac{1}{2}\epsilon^2 \langle \boldsymbol{\Theta}_\Delta(\boldsymbol{\theta}), \nabla\nabla F(\boldsymbol{\Theta}_\infty(\boldsymbol{\theta}))\boldsymbol{\Theta}_\Delta(\boldsymbol{\theta}) \rangle + O(\epsilon^3) \tag{62}$$

$$K(\boldsymbol{\Theta}_\infty(\boldsymbol{\theta}) + \epsilon\boldsymbol{\Theta}_\Delta(\boldsymbol{\theta}), \boldsymbol{\Theta}_\infty(\boldsymbol{\theta}') + \epsilon\boldsymbol{\Theta}_\Delta(\boldsymbol{\theta}'))$$
$$= K(\boldsymbol{\Theta}_\infty(\boldsymbol{\theta}), \boldsymbol{\Theta}_\infty(\boldsymbol{\theta}')) + \epsilon\nabla K(\boldsymbol{\Theta}_\infty(\boldsymbol{\theta}), \boldsymbol{\Theta}_\infty(\boldsymbol{\theta}'))\boldsymbol{\Theta}_\Delta(\boldsymbol{\theta})$$
$$+ \epsilon\nabla' K(\boldsymbol{\Theta}_\infty(\boldsymbol{\theta}), \boldsymbol{\Theta}_\infty(\boldsymbol{\theta}'))\boldsymbol{\Theta}_\Delta(\boldsymbol{\theta}') + \frac{1}{2}\epsilon^2 \langle \boldsymbol{\Theta}_\Delta(\boldsymbol{\theta}), \nabla\nabla K(\boldsymbol{\Theta}_\infty(\boldsymbol{\theta}), \boldsymbol{\Theta}_\infty(\boldsymbol{\theta}'))\boldsymbol{\Theta}_\Delta(\boldsymbol{\theta}) \rangle \tag{63}$$
$$+ \frac{1}{2}\epsilon^2 \langle \boldsymbol{\Theta}_\Delta(\boldsymbol{\theta}'), \nabla'\nabla' K(\boldsymbol{\Theta}_\infty(\boldsymbol{\theta}), \boldsymbol{\Theta}_\infty(\boldsymbol{\theta}'))\boldsymbol{\Theta}_\Delta(\boldsymbol{\theta}') \rangle$$
$$+ \epsilon^2 \langle \boldsymbol{\Theta}_\Delta(\boldsymbol{\theta}), \nabla'\nabla K(\boldsymbol{\Theta}_\infty(\boldsymbol{\theta}), \boldsymbol{\Theta}_\infty(\boldsymbol{\theta}'))\boldsymbol{\Theta}_\Delta(\boldsymbol{\theta}') \rangle + O(\epsilon^3) .$$

Hence, there is

$$
\begin{aligned}
&\mathcal{E}(\boldsymbol{\Theta}_\infty + \epsilon \boldsymbol{\Theta}_\Delta) - \mathcal{E}(\boldsymbol{\Theta}_\infty) \\
={}&\epsilon \int_D \left( -\nabla F(\boldsymbol{\Theta}_\infty(\boldsymbol{\theta})) + \int_D \nabla K(\boldsymbol{\Theta}_\infty(\boldsymbol{\theta}), \boldsymbol{\Theta}_\infty(\boldsymbol{\theta}'))\mu_0(d\boldsymbol{\theta}') \right) \boldsymbol{\Theta}_\Delta(\boldsymbol{\theta})\mu_0(d\boldsymbol{\theta}) \\
&+ \frac{1}{2}\epsilon^2 \left( \int_D \langle \boldsymbol{\Theta}_\Delta(\boldsymbol{\theta}), \left( \nabla\nabla F(\boldsymbol{\Theta}_\infty(\boldsymbol{\theta})) + \int_D \nabla\nabla K(\boldsymbol{\Theta}_\infty(\boldsymbol{\theta}), \boldsymbol{\Theta}_\infty(\boldsymbol{\theta}'))\mu_0(d\boldsymbol{\theta}') \right) \boldsymbol{\Theta}_\Delta(\boldsymbol{\theta})\rangle \mu_0(d\boldsymbol{\theta}) \right. \\
&\left. + \int_D \int_D \langle \boldsymbol{\Theta}_\Delta(\boldsymbol{\theta}), \nabla'\nabla K(\boldsymbol{\Theta}_\infty(\boldsymbol{\theta}), \boldsymbol{\Theta}_\infty(\boldsymbol{\theta}'))\boldsymbol{\Theta}_\Delta(\boldsymbol{\theta}')\rangle \mu_0(d\boldsymbol{\theta})\mu_0(d\boldsymbol{\theta}') \right) + O(\epsilon^3) \,.
\end{aligned}
\tag{64}
$$

Since $\boldsymbol{\Theta}_\Delta$ is arbitrary can $\epsilon$ can be taken arbitrarily small, we see that for $\boldsymbol{\Theta}_\infty$ to be a local minimizer, the first-order condition is, $\forall \boldsymbol{\theta} \in \operatorname{supp}\mu_0$,

$$
-\nabla F(\boldsymbol{\Theta}_\infty(\boldsymbol{\theta})) + \int_D \nabla K(\boldsymbol{\Theta}_\infty(\boldsymbol{\theta}), \boldsymbol{\Theta}_\infty(\boldsymbol{\theta}'))\mu_0(d\boldsymbol{\theta}') = 0 \,,
\tag{65}
$$

or

$$
\nabla V(\boldsymbol{\Theta}_\infty(\boldsymbol{\theta}), \mu_\infty) = 0 \,,
\tag{66}
$$

and the second-order condition is, $\forall \boldsymbol{\Theta}_\Delta$,

$$
\begin{aligned}
&\int_D \langle \boldsymbol{\Theta}_\Delta(\boldsymbol{\theta}), \left( \nabla\nabla F(\boldsymbol{\Theta}_\infty(\boldsymbol{\theta})) + \int_D \nabla\nabla K(\boldsymbol{\Theta}_\infty(\boldsymbol{\theta}), \boldsymbol{\Theta}_\infty(\boldsymbol{\theta}'))\mu_0(d\boldsymbol{\theta}') \right) \boldsymbol{\Theta}_\Delta(\boldsymbol{\theta})\rangle \mu_0(d\boldsymbol{\theta}) \\
&+ \int_D \int_D \langle \boldsymbol{\Theta}_\Delta(\boldsymbol{\theta}), \nabla'\nabla K(\boldsymbol{\Theta}_\infty(\boldsymbol{\theta}), \boldsymbol{\Theta}_\infty(\boldsymbol{\theta}'))\boldsymbol{\Theta}_\Delta(\boldsymbol{\theta}')\rangle \mu_0(d\boldsymbol{\theta})\mu_0(d\boldsymbol{\theta}') \geq 0 \,,
\end{aligned}
\tag{67}
$$

or

$$
\begin{aligned}
&\int_D \langle \boldsymbol{\Theta}_\Delta(\boldsymbol{\theta}), \nabla\nabla V(\boldsymbol{\Theta}_\infty(\boldsymbol{\theta}), \mu_\infty)\boldsymbol{\Theta}_\Delta(\boldsymbol{\theta})\rangle \mu_0(d\boldsymbol{\theta}) \\
&+ \int_D \int_D \langle \boldsymbol{\Theta}_\Delta(\boldsymbol{\theta}), \nabla'\nabla K(\boldsymbol{\Theta}_\infty(\boldsymbol{\theta}), \boldsymbol{\Theta}_\infty(\boldsymbol{\theta}'))\boldsymbol{\Theta}_\Delta(\boldsymbol{\theta}')\rangle \mu_0(d\boldsymbol{\theta})\mu_0(d\boldsymbol{\theta}') \geq 0 \,.
\end{aligned}
\tag{68}
$$

Suppose for contradiction that $\exists D^- \subseteq D$ with $\mu_0(D^-) > 0$ such that $\nabla\nabla V(\boldsymbol{\Theta}_\infty(\boldsymbol{\theta}), \mu_\infty)$ is not positive semidefinite. Define $\Lambda_\infty(\boldsymbol{\theta})$ to be the least eigenvalue of $\nabla\nabla V(\boldsymbol{\Theta}_\infty(\boldsymbol{\theta}), \mu_\infty)$. Then there is $\Lambda_\infty(\boldsymbol{\theta}) < 0$ on $D^-$. In addition, $\exists \zeta > 0$, $\exists D_0^- \subseteq D^-$ with $\mu_0(D_0^-) > 0$ such that $\Lambda_\infty(\boldsymbol{\theta}) < -\zeta$. For $\boldsymbol{\theta} \in D_0^-$, let $\boldsymbol{\Theta}_{\Delta,0}(\boldsymbol{\theta})$ be a normalized eigenvector to $\nabla\nabla V(\boldsymbol{\Theta}_\infty(\boldsymbol{\theta}), \mu_\infty)$ associated with its least eigenvalue. Moreover, for $J \in \mathbb{N}^*$ that is large enough, we can select any subset $D_J^- \subset D_0^-$ such that $\mu_0(D_J^-) = \frac{1}{J} < \mu_0(D_0^-)$. Then, define

$$
\boldsymbol{\Theta}_{\Delta,J}(\boldsymbol{\theta}) = J^{1/2} \mathbb{1}_{\boldsymbol{\theta} \in D_J^-} \boldsymbol{\Theta}_{\Delta,0}(\boldsymbol{\theta}) \,,
\tag{69}
$$

Then, there is

$$
\begin{aligned}
&\int_D \int_D \langle \boldsymbol{\Theta}_\Delta(\boldsymbol{\theta}), \nabla'\nabla K(\boldsymbol{\Theta}_\infty(\boldsymbol{\theta}), \boldsymbol{\Theta}_\infty(\boldsymbol{\theta}'))\boldsymbol{\Theta}_\Delta(\boldsymbol{\theta}')\rangle \mu_0(d\boldsymbol{\theta})\mu_0(d\boldsymbol{\theta}') \\
={}&\int_\Omega \left| \int_D \nabla\varphi(\boldsymbol{\Theta}_\infty(\boldsymbol{\theta}), \boldsymbol{x})\boldsymbol{\Theta}_{\Delta,n}\mu_0(d\boldsymbol{\theta}) \right|^2 \hat{\nu}(d\boldsymbol{x}) \\
={}&\int_\Omega \left| J^{1/2} \int_{D_J^-} \nabla\varphi(\boldsymbol{\Theta}_\infty(\boldsymbol{\theta}), \boldsymbol{x})\boldsymbol{\Theta}_{\Delta,0}\mu_0(d\boldsymbol{\theta}) \right|^2 \hat{\nu}(d\boldsymbol{x}) \\
\leq{}&C_{\nabla\varphi}^2 J^{-1} \,.
\end{aligned}
\tag{70}
$$

On the other hand

$$
\begin{aligned}
&\int_D \langle \boldsymbol{\Theta}_{\Delta,J}(\boldsymbol{\theta}), \nabla\nabla V(\boldsymbol{\Theta}_\infty(\boldsymbol{\theta}), \mu_\infty)\boldsymbol{\Theta}_{\Delta,J}(\boldsymbol{\theta})\rangle \mu_0(d\boldsymbol{\theta}) \\
={}&\int_{D_J^-} J^{-1} \langle \boldsymbol{\Theta}_{\Delta,0}(\boldsymbol{\theta}), \nabla\nabla V(\boldsymbol{\Theta}_\infty(\boldsymbol{\theta}), \mu_\infty)\boldsymbol{\Theta}_{\Delta,0}(\boldsymbol{\theta})\rangle \mu_0(d\boldsymbol{\theta}) \\
\leq{}&-\zeta \,.
\end{aligned}
\tag{71}
$$

Therefore, for $J$ large enough, we will have

$$\int_D \int_D \langle \boldsymbol{\Theta}_\Delta(\boldsymbol{\theta}), \nabla' \nabla K(\boldsymbol{\Theta}_\infty(\boldsymbol{\theta}), \boldsymbol{\Theta}_\infty(\boldsymbol{\theta}')) \boldsymbol{\Theta}_\Delta(\boldsymbol{\theta}') \rangle \mu_0(d\boldsymbol{\theta}) \mu_0(d\boldsymbol{\theta}')$$
$$+ \int_D \langle \boldsymbol{\Theta}_{\Delta,n}(\boldsymbol{\theta}), \nabla \nabla V(\boldsymbol{\Theta}_\infty(\boldsymbol{\theta}), \mu_\infty) \boldsymbol{\Theta}_{\Delta,J}(\boldsymbol{\theta}) \rangle \mu_0(d\boldsymbol{\theta}) < 0, \tag{72}$$

which contradicts (68). Hence, we can conclude that $\mu_0$-almost surely, $\nabla \nabla V(\boldsymbol{\Theta}_\infty(\boldsymbol{\theta}), \mu_\infty)$ is positive semidefinite.

## D  Derivations of the Dynamical Central Limit Theorem

### D.1  Proof of Proposition 3.1 (Dynamical CLT - I)

The following derivation is an adaptation of the approach in [10] for Vlasov interacting particle systems to our scenario. To start, $\boldsymbol{\Theta}_t$ and $\boldsymbol{\Theta}_t^{(m)}$ are governed by the following equations, respectively:

$$\dot{\boldsymbol{\Theta}}_t(\boldsymbol{\theta}) = -\nabla V(\boldsymbol{\Theta}_t(\boldsymbol{\theta}), \mu_t), \qquad \boldsymbol{\Theta}_0(\boldsymbol{\theta}) = \boldsymbol{\theta}$$
$$\dot{\boldsymbol{\Theta}}_t^{(m)}(\boldsymbol{\theta}) = -\nabla V(\boldsymbol{\Theta}_t^{(m)}(\boldsymbol{\theta}), \mu_t^{(m)}), \qquad \boldsymbol{\Theta}_0^{(m)}(\boldsymbol{\theta}) = \boldsymbol{\theta} \tag{73}$$

Taking the difference between the two equations in (73) and using the mean value theorem, we get

$$\dot{\boldsymbol{T}}_t^{(m)}(\boldsymbol{\theta})$$
$$= m^{1/2} \Big( \dot{\boldsymbol{\Theta}}_t^{(m)}(\boldsymbol{\theta}) - \dot{\boldsymbol{\Theta}}_t(\boldsymbol{\theta}) \Big)$$
$$= - m^{1/2} \Big( \nabla V(\boldsymbol{\Theta}_t^{(m)}(\boldsymbol{\theta}), \mu_t^{(m)}) - \nabla V(\boldsymbol{\Theta}_t(\boldsymbol{\theta}), \mu_t) \Big)$$
$$= - m^{1/2} \Big( \nabla V(\boldsymbol{\Theta}_t^{(m)}(\boldsymbol{\theta}), \mu_t) - \nabla V(\boldsymbol{\Theta}_t(\boldsymbol{\theta}), \mu_t) \Big) - m^{1/2} \Big( \nabla V(\boldsymbol{\Theta}_t, \mu_t^{(m)}) - \nabla V(\boldsymbol{\Theta}_t(\boldsymbol{\theta}), \mu_t) \Big)$$
$$\quad - m^{1/2} \Big[ \Big( \nabla V(\boldsymbol{\Theta}_t^{(m)}(\boldsymbol{\theta}), \mu_t^{(m)}) - \nabla V(\boldsymbol{\Theta}_t(\boldsymbol{\theta}), \mu_t^{(m)}) \Big) - \Big( \nabla V(\boldsymbol{\Theta}_t^{(m)}, \mu_t) - \nabla V(\boldsymbol{\Theta}_t(\boldsymbol{\theta}), \mu_t) \Big) \Big]$$
$$= - \nabla \nabla V(\tilde{\boldsymbol{\Theta}}_{t,1}^{(m)}(\boldsymbol{\theta}), \mu_t) \boldsymbol{T}_t^{(m)}(\boldsymbol{\theta}) - \int_D \nabla K(\boldsymbol{\Theta}_t(\boldsymbol{\theta}), \boldsymbol{\theta}') \omega_t^{(m)}(d\boldsymbol{\theta}')$$
$$\quad - m^{-1/2} \Big( \int_D \nabla \nabla K(\tilde{\boldsymbol{\Theta}}_{t,2}^{(m)}(\boldsymbol{\theta}), \boldsymbol{\theta}') \omega_t^{(m)}(d\boldsymbol{\theta}') \Big) \boldsymbol{T}_t^{(m)}(\boldsymbol{\theta}), \tag{74}$$

where $\tilde{\boldsymbol{\Theta}}_{t,1}^{(m)}(\boldsymbol{\theta})$ and $\tilde{\boldsymbol{\Theta}}_{t,2}^{(m)}(\boldsymbol{\theta})$ denote points that lie on the line segment between $\boldsymbol{\Theta}_t(\boldsymbol{\theta})$ and $\boldsymbol{\Theta}_t^{(m)}(\boldsymbol{\theta})$. Using (25), we can substitute $\omega_t^{(m)}$ in the second term at the right hand side, for which we get

$$\int_D \nabla K(\boldsymbol{\Theta}_t(\boldsymbol{\theta}), \boldsymbol{\theta}') \omega_t^{(m)}(d\boldsymbol{\theta}') = \int_D \nabla K(\boldsymbol{\Theta}_t(\boldsymbol{\theta}), \boldsymbol{\Theta}_t(\boldsymbol{\theta}')) \omega_0^{(m)}(d\boldsymbol{\theta}')$$
$$+ \int_D \nabla' \nabla K(\boldsymbol{\Theta}_t(\boldsymbol{\theta}), \tilde{\boldsymbol{\Theta}}_{t,3}^{(m)}(\boldsymbol{\theta}')) \boldsymbol{T}_t^{(m)}(\boldsymbol{\theta}') \mu_0(d\boldsymbol{\theta}')$$
$$+ m^{-1/2} \int_D \nabla' \nabla K(\boldsymbol{\Theta}_t(\boldsymbol{\theta}), \tilde{\boldsymbol{\Theta}}_{t,3}^{(m)}(\boldsymbol{\theta}')) \boldsymbol{T}_t^{(m)}(\boldsymbol{\theta}') \omega_0^{(m)}(d\boldsymbol{\theta}'). \tag{75}$$

Therefore, under Assumption 2.2, we have

$$\dot{\boldsymbol{T}}_t^{(m)}(\boldsymbol{\theta}) = -\nabla \nabla V(\tilde{\boldsymbol{\Theta}}_{t,1}^{(m)}, \mu_t) \boldsymbol{T}_t^{(m)}(\boldsymbol{\theta})$$
$$- \int_D \nabla' \nabla K(\boldsymbol{\Theta}_t(\boldsymbol{\theta}), \tilde{\boldsymbol{\Theta}}_{t,3}^{(m)}(\boldsymbol{\theta}')) \boldsymbol{T}_t^{(m)}(\boldsymbol{\theta}') \mu_0(d\boldsymbol{\theta}')$$
$$- \int_D \nabla K(\boldsymbol{\Theta}_t(\boldsymbol{\theta}), \boldsymbol{\Theta}_t(\boldsymbol{\theta}')) \omega_0^{(m)}(d\boldsymbol{\theta}') + O(m^{-1/2}). \tag{76}$$

Now, we consider the limit as $m \to \infty$. By the standard CLT, we have that $\omega_0^{(m)}(d\boldsymbol{\theta}) \rightharpoonup \omega_0(d\boldsymbol{\theta})$ weakly with respect to $\mathbb{P}_0$, where $\omega_0(d\boldsymbol{\theta})$ is the Gaussian measure with mean zero and covariance defined in (27). On the other hand, by finite-time LLN, we have $\boldsymbol{\Theta}_t^{(m)}(\boldsymbol{\theta}) \to \boldsymbol{\Theta}_t(\boldsymbol{\theta})$ pointwise, $\mathbb{P}_0$-almost surely, and as a consequence $\tilde{\boldsymbol{\Theta}}_{t,1}^{(m)}(\boldsymbol{\theta}), \bar{\boldsymbol{\Theta}}_{t,3}^{(m)}(\boldsymbol{\theta}) \to \boldsymbol{\Theta}_t(\boldsymbol{\theta})$ as well. Therefore, $\boldsymbol{T}_t^{(m)}(\boldsymbol{\theta}) \to \boldsymbol{T}_t(\boldsymbol{\theta})$ pointwise, $\mathbb{P}_0$-almost surely, where the limiting $\boldsymbol{T}_t(\boldsymbol{\theta})$ solves the equation obtained by taking the limit $m \to \infty$ on both sides of (76), which becomes (28). (28) should be solved with initial condition $\boldsymbol{T}_0(\boldsymbol{\theta}) = 0$ since $\boldsymbol{T}_0^{(m)}(\boldsymbol{\theta}) = m^{1/2}(\boldsymbol{\Theta}_0^{(m)}(\boldsymbol{\theta}) - \boldsymbol{\Theta}_0(\boldsymbol{\theta})) = 0$.

Finally, taking the limit $m \to \infty$ on both sides of the equation (25), we deduce that $\omega_t^{(m)}(d\boldsymbol{\theta}) \rightharpoonup \omega_t(d\boldsymbol{\theta})$ weakly, in law with respect to $\mathbb{P}_0$, where the limiting $\omega_t(d\boldsymbol{\theta})$ satisfies

$$\int_D \chi(\boldsymbol{\theta})\omega_t(d\boldsymbol{\theta}) = \int_D \chi(\boldsymbol{\Theta}_t(\boldsymbol{\theta}))\omega_0(d\boldsymbol{\theta}) + \int_D \nabla\chi(\boldsymbol{\Theta}_t(\boldsymbol{\theta})) \cdot \boldsymbol{T}_t(\boldsymbol{\theta})\mu_0(d\boldsymbol{\theta}) . \tag{77}$$

This ends the proof of Proposition 3.1. $\qquad\qquad\square$

### D.2    Proof of Proposition 3.3 (Dynamical CLT - II)

Recall from (28) that

$$\dot{\boldsymbol{T}}_t(\boldsymbol{\theta}) = -\nabla\nabla V(\boldsymbol{\Theta}_t(\boldsymbol{\theta}), \mu_t)\boldsymbol{T}_t(\boldsymbol{\theta}) - \int_D \nabla'\nabla K(\boldsymbol{\Theta}_t(\boldsymbol{\theta}), \boldsymbol{\Theta}_t(\boldsymbol{\theta}'))\boldsymbol{T}_t(\boldsymbol{\theta}')\mu_0(d\boldsymbol{\theta}')$$
$$- \int_D \nabla K(\boldsymbol{\Theta}_t(\boldsymbol{\theta}), \boldsymbol{\Theta}_t(\boldsymbol{\theta}'))\omega_0(d\boldsymbol{\theta}') \tag{78}$$
$$= -\nabla\nabla V(\boldsymbol{\Theta}_t(\boldsymbol{\theta}), \mu_t)\boldsymbol{T}_t(\boldsymbol{\theta}) - \int_D \nabla K(\boldsymbol{\Theta}_t(\boldsymbol{\theta}), \boldsymbol{\theta}')\omega_t(d\boldsymbol{\theta}') .$$

Since $\boldsymbol{T}_0(\boldsymbol{\theta}) = 0$, we can use Duhamel's principle to deduce that

$$\boldsymbol{T}_t(\boldsymbol{\theta}) = -\int_0^t J_{t,s}(\boldsymbol{\theta}) \int_D \nabla K(\boldsymbol{\Theta}_s(\boldsymbol{\theta}), \boldsymbol{\theta}')\omega_s(d\boldsymbol{\theta}')ds$$
$$= -\int_0^t \int_\Omega J_{t,s}(\boldsymbol{\theta})\nabla\varphi(\boldsymbol{\Theta}_s(\boldsymbol{\theta}), \boldsymbol{x}) \int_D \varphi(\boldsymbol{\theta}', \boldsymbol{x})\omega_s(d\boldsymbol{\theta}')\hat{\nu}(d\boldsymbol{x})ds \tag{79}$$
$$= -\int_0^t \int_\Omega J_{t,s}(\boldsymbol{\theta})\nabla\varphi(\boldsymbol{\Theta}_s(\boldsymbol{\theta}), \boldsymbol{x})g_s(\boldsymbol{x})\hat{\nu}(d\boldsymbol{x})ds,$$

where the tensor $J_{t,s}(\boldsymbol{\theta})$ is the Jacobian defined in Proposition 3.3. As a result

$$g_t(\boldsymbol{x}) = \int_D \varphi(\boldsymbol{\theta}, \boldsymbol{x})\omega_t(d\boldsymbol{\theta})$$
$$= \int_D \varphi(\boldsymbol{\Theta}_t(\boldsymbol{\theta}), \boldsymbol{x})\omega_0(d\boldsymbol{\theta}) + \int_D \nabla\varphi(\boldsymbol{\Theta}_t(\boldsymbol{\theta}), \boldsymbol{x}) \cdot \boldsymbol{T}_t(\boldsymbol{\theta})\mu_0(d\boldsymbol{\theta})$$
$$= \int_D \varphi(\boldsymbol{\Theta}_t(\boldsymbol{\theta}), \boldsymbol{x})\omega_0(d\boldsymbol{\theta})$$
$$- \int_D \int_0^t \int_\Omega \langle \nabla\varphi(\boldsymbol{\Theta}_t(\boldsymbol{\theta}), \boldsymbol{x}), J_{t,s}(\boldsymbol{\theta})\nabla\varphi(\boldsymbol{\Theta}_s(\boldsymbol{\theta}), \boldsymbol{x}')\rangle g_s(\boldsymbol{x}')\hat{\nu}(d\boldsymbol{x}')ds\mu_0(d\boldsymbol{\theta}) \tag{80}$$
$$= \bar{g}_t(\boldsymbol{x}) - \int_0^t \int_\Omega \int_D \langle \nabla\varphi(\boldsymbol{\Theta}_t(\boldsymbol{\theta}), \boldsymbol{x}), J_{t,s}(\boldsymbol{\theta})\nabla\varphi(\boldsymbol{\Theta}_s(\boldsymbol{\theta}), \boldsymbol{x}')\rangle \mu_0(d\boldsymbol{\theta})g_s(\boldsymbol{x}')\hat{\nu}(d\boldsymbol{x}')ds$$
$$= \bar{g}_t(\boldsymbol{x}) - \int_0^t \int_\Omega \Gamma_{t,s}(\boldsymbol{x}, \boldsymbol{x}')g_s(\boldsymbol{x}')\hat{\nu}(d\boldsymbol{x}')ds,$$

with $\bar{g}_t(\boldsymbol{x})$ and $\Gamma_{t,s}(\boldsymbol{x}, \boldsymbol{x}')$ defined in (32) and (34), respectively. This is (33). $\qquad\square$

# E  Long-Time Behavior of the Fluctuations

## E.1  Proof of Theorem 3.4 ($\mu_0 = \mu_\infty$ case)

With the argument outlined in Section 3.2, what remains to be shown is that $\Gamma^\infty_{t-s}$ is positive-semidefinite as a Volterra kernel, according to the definition in [33]. We will utilize the following known result:

**Proposition E.1 (Gripenberg et al. [33])** *Let $k : [0, \infty) \to \mathbb{R}^{n \times n}$ be a convolution-type kernel for a linear Volterra equation in $\mathbb{R}^n$. If $\forall \eta \in \mathbb{R}^n$, the function $t \mapsto \langle \eta, k(t)\eta \rangle$ is a nonnegative, nonincreasing and convex function on $(0, \infty)$, then $k$ is nonnegative, meaning that $\forall \phi : [0, \infty) \to \mathbb{R}^n$ with compact support, there is*

$$\int_0^\infty \int_0^t \langle \phi(t), k(t-s)\phi(s) \rangle \, ds \, dt \geq 0 \,. \tag{81}$$

Thus, to take advantage of this proposition, we need to verify that $\forall \eta \in \mathbb{R}^n$, $\langle \eta, \Gamma^\infty_t \eta \rangle$ is

*(1) nonnegative:*

$$
\begin{aligned}
&\langle \eta, \Gamma^\infty_t \eta \rangle \\
&= \int_{\Omega \times \Omega} \int_D \left\langle \nabla\varphi(\boldsymbol{\Theta}_\infty(\boldsymbol{\theta}), \boldsymbol{x}), e^{-t\nabla\nabla V_\infty(\boldsymbol{\Theta}_\infty(\boldsymbol{\theta}))} \nabla\varphi(\boldsymbol{\Theta}_\infty(\boldsymbol{\theta}), \boldsymbol{x}) \right\rangle \eta(\boldsymbol{x})\eta(\boldsymbol{x}')\mu_0(d\boldsymbol{\theta})\hat{\nu}(d\boldsymbol{x})\hat{\nu}(d\boldsymbol{x}') \\
&= \int_D \left\langle \boldsymbol{b}(\boldsymbol{\theta}), e^{-t\nabla\nabla V_\infty(\boldsymbol{\Theta}_\infty(\boldsymbol{\theta}))} \boldsymbol{b}(\boldsymbol{\theta}) \right\rangle \mu_0(d\boldsymbol{\theta}) \geq 0 \,,
\end{aligned}
\tag{82}
$$

where

$$\boldsymbol{b}(\boldsymbol{\theta}) = \int_\Omega \nabla\varphi(\boldsymbol{\Theta}_\infty(\boldsymbol{\theta}), \boldsymbol{x})\eta(\boldsymbol{x})\hat{\nu}(d\boldsymbol{x}) \tag{83}$$

because by assumption, $\forall \boldsymbol{\theta} \in D$, $\nabla\nabla V_\infty(\boldsymbol{\Theta}_\infty(\boldsymbol{\theta}))$ is positive semidefinite, and hence $e^{-t\nabla\nabla V_\infty(\boldsymbol{\Theta}_\infty(\boldsymbol{\theta}))}$ is a positive semidefinite operator;

*(2) nonincreasing:* Taking derivative with respect to time,

$$\frac{d}{dt}\langle \eta, \Gamma^\infty_t \eta \rangle = -\int_D \left\langle \boldsymbol{b}(\boldsymbol{\theta}), \nabla\nabla V(\boldsymbol{\Theta}_\infty(\boldsymbol{\theta}))e^{-t\nabla\nabla V_\infty(\boldsymbol{\Theta}_\infty(\boldsymbol{\theta}))} \boldsymbol{b}(\boldsymbol{\theta}) \right\rangle \mu_0(d\boldsymbol{\theta}) \leq 0, \tag{84}$$

because again, $\nabla\nabla V_\infty(\boldsymbol{\Theta}_\infty(\boldsymbol{\theta}))$ is positive semidefinite;

*(3) convex:* Taking one more derivative with respect to time,

$$\frac{d^2}{dt^2}\langle \eta, \Gamma^\infty_t \eta \rangle = \int_D \left\langle \boldsymbol{b}(\boldsymbol{\theta}), (\nabla\nabla V(\boldsymbol{\Theta}_\infty(\boldsymbol{\theta})))^2 e^{-t\nabla\nabla V_\infty(\boldsymbol{\Theta}_\infty(\boldsymbol{\theta}))} \boldsymbol{b}(\boldsymbol{\theta}) \right\rangle \mu_0(d\boldsymbol{\theta}) \geq 0, \tag{85}$$

Therefore, we can apply Proposition E.1 to conclude that $\Gamma^\infty_{t-s}$ is PSD as a Volterra kernel, and so $\int_{t_0}^T \int_{t_0}^t \langle g_t, \Gamma^\infty_{t-s} g_s \rangle \, ds \, dt \geq 0$.

## E.2  Proof of Theorem 3.5 (Unregularized case)

Recall that

$$
\begin{aligned}
\lim_{m \to \infty} m\mathbb{E}_0 \|f_t^{(m)} - f_t\|_\nu^2 = \mathbb{E}_0 \|g_t\|_\nu^2 &= \mathbb{E}_0 \int_\Omega \left| \int_D \varphi(\boldsymbol{\theta}, \boldsymbol{x})\omega_t(d\boldsymbol{\theta}) \right|^2 \hat{\nu}(d\boldsymbol{x}) \\
&= \mathbb{E}_0 \int_{D \times D} K(\boldsymbol{\theta}, \boldsymbol{\theta}')\omega_t(d\boldsymbol{\theta})\omega_t(d\boldsymbol{\theta}') \,.
\end{aligned}
\tag{86}
$$

From (26) in Proposition 3.1, this can be further expanded into

$$
\begin{aligned}
\mathbb{E}_0 & \int_{D \times D} K(\boldsymbol{\theta}, \boldsymbol{\theta}') \omega_t(d\boldsymbol{\theta}) \omega_t(d\boldsymbol{\theta}') \\
= & \ \mathbb{E}_0 \int_{D \times D} \langle \boldsymbol{T}_t(\boldsymbol{\theta}), \nabla \nabla' K(\boldsymbol{\Theta}_t(\boldsymbol{\theta}), \boldsymbol{\Theta}_t(\boldsymbol{\theta}')) \boldsymbol{T}_t(\boldsymbol{\theta}') \rangle \mu_0(d\boldsymbol{\theta}) \mu_0(d\boldsymbol{\theta}') \\
& + 2\mathbb{E}_0 \int_{D \times D} \nabla K(\boldsymbol{\Theta}_t(\boldsymbol{\theta}), \boldsymbol{\Theta}_t(\boldsymbol{\theta}')) \boldsymbol{T}_t(\boldsymbol{\theta}) \mu_0(d\boldsymbol{\theta}) \omega_0(d\boldsymbol{\theta}') \\
& + \mathbb{E}_0 \int_{D \times D} K(\boldsymbol{\Theta}_t(\boldsymbol{\theta}), \boldsymbol{\Theta}_t(\boldsymbol{\theta}')) \omega_0(d\boldsymbol{\theta}) \omega_0(d\boldsymbol{\theta}') \ .
\end{aligned}
\tag{87}
$$

The last term at the RHS is equal to $\mathbb{E}_0 \|\bar{g}_t\|_{\hat{\nu}}^2$ with $\bar{g}_t$ defined in (32). Using (27), it can be explicitly computed as

$$
\begin{aligned}
\mathbb{E}_0 \|\bar{g}_t\|_{\hat{\nu}}^2 = & \ \mathbb{E}_0 \int_{D \times D} K(\boldsymbol{\Theta}_t(\boldsymbol{\theta}), \boldsymbol{\Theta}_t(\boldsymbol{\theta}')) \omega_0(d\boldsymbol{\theta}) \omega_0(d\boldsymbol{\theta}') \\
= & \int_{D \times D} K(\boldsymbol{\Theta}_t(\boldsymbol{\theta}), \boldsymbol{\Theta}_t(\boldsymbol{\theta}')) \left( \mu_0(d\boldsymbol{\theta}) \delta_{\boldsymbol{\theta}}(d\boldsymbol{\theta}') - \mu_0(d\boldsymbol{\theta}) \mu_0(d\boldsymbol{\theta}') \right) \\
= & \int_D K(\boldsymbol{\theta}, \boldsymbol{\theta}) \mu_t(d\boldsymbol{\theta}) - \int_{D \times D} K(\boldsymbol{\theta}, \boldsymbol{\theta}) \mu_t(d\boldsymbol{\theta}) \mu_t(d\boldsymbol{\theta}') \\
= & \int_D K(\boldsymbol{\theta}, \boldsymbol{\theta}) \mu_t(d\boldsymbol{\theta}) - \|f_t\|_{\hat{\nu}}^2 \ .
\end{aligned}
\tag{88}
$$

Thus,

$$
\begin{aligned}
\lim_{t \to \infty} \mathbb{E}_0 \|\bar{g}_t\|_{\hat{\nu}}^2 = & \lim_{t \to \infty} \int_D K(\boldsymbol{\theta}, \boldsymbol{\theta}) \mu_t(d\boldsymbol{\theta}) - \|f_t\|_{\hat{\nu}}^2 \\
= & \int_D K(\boldsymbol{\theta}, \boldsymbol{\theta}) \mu_\infty(d\boldsymbol{\theta}) - \|f_\infty\|_{\hat{\nu}}^2 \\
= & \ \mathbb{E}_0 \|\bar{g}_\infty\|_{\hat{\nu}}^2
\end{aligned}
\tag{89}
$$

and so

$$
\lim_{T \to \infty} \fint_0^T \mathbb{E}_0 \|\bar{g}_t\|_{\hat{\nu}}^2 dt = \mathbb{E}_0 \|\bar{g}_\infty\|_{\hat{\nu}}^2 \ ,
\tag{90}
$$

where here and below we denote $\fint_0^t [\cdot] \ dt = \frac{1}{t} \int_0^t [\cdot] \ dt$. As a result, to prove (41) or (44) in Theorem 3.5, it suffices to establish that

$$
\lim_{T \to \infty} \fint_0^T \mathfrak{D}_t dt \leq 0 \ ,
\tag{91}
$$

or

$$
\lim_{T \to \infty} \fint_0^T \mathfrak{D}_t dt \leq -\mathbb{E}_0 \|\bar{g}_\infty\|_{\hat{\nu}}^2 \ ,
\tag{92}
$$

respectively, where we defined

$$
\begin{aligned}
\mathfrak{D}_t := & \ \mathbb{E}_0 \int_{D \times D} K(\boldsymbol{\theta}, \boldsymbol{\theta}') \omega_t(d\boldsymbol{\theta}) \omega_t(d\boldsymbol{\theta}') - \mathbb{E}_0 \int_{D \times D} K(\boldsymbol{\Theta}_t(\boldsymbol{\theta}), \boldsymbol{\Theta}_t(\boldsymbol{\theta}')) \omega_0(d\boldsymbol{\theta}) \omega_0(d\boldsymbol{\theta}') \\
= & \ \mathbb{E}_0 \int_{D \times D} \langle \boldsymbol{T}_t(\boldsymbol{\theta}), \nabla \nabla' K(\boldsymbol{\Theta}_t(\boldsymbol{\theta}), \boldsymbol{\Theta}_t(\boldsymbol{\theta}')) \boldsymbol{T}_t(\boldsymbol{\theta}') \rangle \mu_0(d\boldsymbol{\theta}) \mu_0(d\boldsymbol{\theta}') \\
& + 2\mathbb{E}_0 \int_{D \times D} \nabla K(\boldsymbol{\Theta}_t(\boldsymbol{\theta}), \boldsymbol{\Theta}_t(\boldsymbol{\theta}')) \boldsymbol{T}_t(\boldsymbol{\theta}) \mu_0(d\boldsymbol{\theta}) \omega_0(d\boldsymbol{\theta}') \ .
\end{aligned}
\tag{93}
$$

To this end, we examine (28) as an infinite-dimensional ODE. With the Hilbert space $\mathcal{V}(D)$ defined in Appendix A and $\boldsymbol{b}_t$, $\mathcal{A}_t^{(K)}$ and $\mathcal{A}_t^{(V)}$ defined by (49), (51) and (52), respectively, we can rewrite (28) as the following ODE on $\mathcal{V}(D)$:

$$
\dot{\boldsymbol{T}}_t = -(\mathcal{A}_t^{(K)} + \mathcal{A}_t^{(V)}) \boldsymbol{T}_t - \boldsymbol{b}_t,
\tag{94}
$$

We can also rewrite (93) as

$$\mathfrak{D}_t = \langle \boldsymbol{T}_t, \mathcal{A}_t^{(K)} \boldsymbol{T}_t \rangle_0 + 2\langle \boldsymbol{T}_t, \boldsymbol{b}_t \rangle_0 \ . \tag{95}$$

From (94), we can deduce that

$$\frac{1}{2}\frac{d}{dt}\|\boldsymbol{T}_t\|_0^2 = -\langle \boldsymbol{T}_t, \mathcal{A}_t^{(V)} \boldsymbol{T}_t \rangle_0 - \langle \boldsymbol{T}_t, \mathcal{A}_t^{(K)} \boldsymbol{T}_t \rangle_0 - \langle \boldsymbol{T}_t, \boldsymbol{b}_t \rangle_0, \tag{96}$$

or equivalently

$$\langle \boldsymbol{T}_t, \mathcal{A}_t^{(K)} \boldsymbol{T}_t \rangle_0 + \langle \boldsymbol{T}_t, \boldsymbol{b}_t \rangle_0 = -\frac{1}{2}\frac{d}{dt}\|\boldsymbol{T}_t\|_0^2 - \langle \boldsymbol{T}_t, \mathcal{A}_t^{(V)} \boldsymbol{T}_t \rangle_0 \ . \tag{97}$$

Therefore, we can rewrite (93) as

$$\begin{aligned}
\mathfrak{D}_t &= 2\left( \langle \boldsymbol{T}_t, \mathcal{A}_t^{(K)} \boldsymbol{T}_t \rangle_0 + \langle \boldsymbol{T}_t, \boldsymbol{b}_t \rangle_0 \right) - \langle \boldsymbol{T}_t, \mathcal{A}_t^{(K)} \boldsymbol{T}_t \rangle_0 \\
&= 2\left( -\frac{1}{2}\frac{d}{dt}\|\boldsymbol{T}_t\|_0^2 - \langle \boldsymbol{T}_t, \mathcal{A}_t^{(V)} \boldsymbol{T}_t \rangle_0 \right) - \langle \boldsymbol{T}_t, \mathcal{A}_t^{(K)} \boldsymbol{T}_t \rangle_0 \\
&= -\frac{d}{dt}\|\boldsymbol{T}_t\|_0^2 - 2\langle \boldsymbol{T}_t, \mathcal{A}_t^{(V)} \boldsymbol{T}_t \rangle_0 - \langle \boldsymbol{T}_t, \mathcal{A}_t^{(K)} \boldsymbol{T}_t \rangle_0
\end{aligned} \tag{98}$$

and as a result, since $\boldsymbol{T}_0 = 0$,

$$\fint_0^T \mathfrak{D}_t dt = -\frac{1}{T}\|\boldsymbol{T}_T\|_0^2 - 2\fint_0^T \langle \boldsymbol{T}_t, \mathcal{A}_t^{(V)} \boldsymbol{T}_t \rangle_0 dt - \fint_0^T \langle \boldsymbol{T}_t, \mathcal{A}_t^{(K)} \boldsymbol{T}_t \rangle_0 dt \ . \tag{99}$$

Note that for all $t$, $\mathcal{A}_t^{(K)}$ is a positive semidefinite (PSD) operator on $\mathcal{V}(D)$, as $\forall \boldsymbol{\xi} \in \mathcal{V}(D)$,

$$\begin{aligned}
\langle \mathcal{A}_t^{(K)} \boldsymbol{\xi}, \boldsymbol{\xi} \rangle_0 &= \mathbb{E}_0 \int_{D \times D} \langle \boldsymbol{\xi}(\boldsymbol{\theta}), \nabla\nabla' K(\boldsymbol{\Theta}_t(\boldsymbol{\theta}), \boldsymbol{\Theta}_t(\boldsymbol{\theta}'))\boldsymbol{\xi}(\boldsymbol{\theta}') \rangle \mu_0(d\boldsymbol{\theta})\mu_0(d\boldsymbol{\theta}') \\
&= \mathbb{E}_0 \int_\Omega \left| \int_D \nabla\varphi(\boldsymbol{\Theta}_t(\boldsymbol{\theta})) \cdot \boldsymbol{\xi}(\boldsymbol{\theta})\mu_0(d\boldsymbol{\theta}) \right|^2 \hat{\nu}(d\boldsymbol{x}) \geq 0 \ .
\end{aligned} \tag{100}$$

This implies that $\fint_0^T \langle \boldsymbol{T}_t, \mathcal{A}_t^{(K)} \boldsymbol{T}_t \rangle_0 dt \geq 0$. Hence, to establish (91), it is sufficient to show that

$$\lim_{T \to \infty} \fint_0^T \langle \boldsymbol{T}_t, \mathcal{A}_t^{(V)} \boldsymbol{T}_t \rangle_0 dt = 0 \ . \tag{101}$$

To this end, we need two lemmas that are proved below in Appendices E.2.1 and E.2.2, respectively:

**Lemma E.2** *Assuming* (42) *and* (43) *together with Assumptions* 2.2, 2.3 *and* 2.4*, we have*

$$\int_0^\infty \|\mathcal{A}_t^{(V)}\|_0 dt < \infty \tag{102}$$

$$\int_0^\infty \|\mathcal{A}_\infty^{(K)} - \mathcal{A}_t^{(K)}\|_0 dt < \infty \tag{103}$$

$$\int_0^\infty \|\boldsymbol{b}_t - \boldsymbol{b}_\infty\|_0 dt < \infty \tag{104}$$

**Lemma E.3** *Assuming* (42) *and* (43) *together with Assumptions* 2.2, 2.3 *and* 2.4*, we have*

$$\sup_{t<\infty} \|\boldsymbol{T}_t\|_0^2 < \infty \ . \tag{105}$$

With these two lemmas, we can show that

$$\begin{aligned}
\left| \fint_0^T \langle \boldsymbol{T}_t, \mathcal{A}_t^{(V)} \boldsymbol{T}_t \rangle_0 dt \right| &\leq \fint_0^T \|\mathcal{A}_t^{(V)}\|_0 \|\boldsymbol{T}_t\|_0^2 dt \\
&\leq \left( \fint_0^T \|\mathcal{A}_t^{(V)}\|_0 dt \right) \sup_{t<\infty} \|\boldsymbol{T}_t\|_0^2 \\
&< \infty \ ,
\end{aligned} \tag{106}$$

and therefore (101) is satisfied. This finishes the proof of (41) under (42) and (43) together with Assumptions 2.2, 2.3 and 2.4.

Next, we show (44) under the additional condition of Assumption 2.1. Thanks to (99) and (101), it is sufficient to establish that

$$\lim_{T\to\infty}\fint_0^T \langle \boldsymbol{T}_t, \mathcal{A}_t^{(K)}\boldsymbol{T}_t\rangle_0 dt = \mathbb{E}_0\|\bar{g}_\infty\|_\nu^2 \ . \tag{107}$$

Heuristically, if $\boldsymbol{T}_\infty := \lim_{t\to\infty}\boldsymbol{T}_t$ exists, then from (94), it has to satisfy

$$-\boldsymbol{b}_\infty = \left(\mathcal{A}_\infty^{(V)} + \mathcal{A}_\infty^{(K)}\right)\boldsymbol{T}_\infty = \mathcal{A}_\infty^{(K)}\boldsymbol{T}_\infty \ , \tag{108}$$

as $\mathcal{A}_\infty^{(V)} = 0$ (because $\nabla\nabla V(\boldsymbol{\theta}, \mu_\infty) = \int_\Omega \varphi(\boldsymbol{\theta}, \boldsymbol{x})(f_\infty(\boldsymbol{x}) - f_*(\boldsymbol{x}))\hat{\nu}(d\boldsymbol{x}) = 0$ under the assumption of (42)). This equation implies that

$$(\boldsymbol{T}_\infty)^{||} = -\left(\mathcal{A}_\infty^{(K)}\right)^\dagger \boldsymbol{b}_\infty \ , \tag{109}$$

where $(\boldsymbol{T}_\infty)^{||}$ denotes the component of $\boldsymbol{T}_\infty$ in the range of $\mathcal{A}_\infty^{(K)}$, and $\left(\mathcal{A}_\infty^{(K)}\right)^\dagger$ denotes the Moore-Penrose pseudoinverse of $\mathcal{A}_\infty^{(K)}$. As a result,

$$\begin{aligned}
\langle \boldsymbol{T}_\infty, \mathcal{A}_\infty^{(K)}\boldsymbol{T}_\infty\rangle_0 &= \langle (\boldsymbol{T}_\infty)^{||}, \mathcal{A}_\infty^{(K)}(\boldsymbol{T}_\infty)^{||}\rangle_0 \\
&= \langle -\left(\mathcal{A}_\infty^{(K)}\right)^\dagger \boldsymbol{b}_\infty, -\mathcal{A}_\infty^{(K)}\left(\mathcal{A}_\infty^{(K)}\right)^\dagger \boldsymbol{b}_\infty\rangle_0 \\
&= \langle \boldsymbol{b}_\infty, \left(\mathcal{A}_\infty^{(K)}\right)^\dagger \boldsymbol{b}_\infty\rangle_0 \ .
\end{aligned} \tag{110}$$

Rigorously, without assuming the existence of $\boldsymbol{T}_\infty$, we can establish that

**Lemma E.4** *Assuming (42) and (43) together with Assumptions 2.2, 2.3 and 2.4, we have*

$$\lim_{t\to\infty}\fint_0^t \langle \boldsymbol{T}_s, \mathcal{A}_s^{(K)}\boldsymbol{T}_s\rangle_0 ds \geq \langle \boldsymbol{b}_\infty, \left(\mathcal{A}_\infty^{(K)}\right)^\dagger \boldsymbol{b}_\infty\rangle_0 \ . \tag{111}$$

*As a consequence,*

$$\lim_{t\to\infty}\fint_0^t \mathbb{E}_0\|g_s\|_\nu^2 dt \leq \mathbb{E}_0\|\bar{g}_\infty\|_\nu^2 - \langle \boldsymbol{b}_\infty, \left(\mathcal{A}_\infty^{(K)}\right)^\dagger \boldsymbol{b}_\infty\rangle_0 \ . \tag{112}$$

This lemma is proved in E.2.3. It implies that we only need to show that

$$\langle \boldsymbol{b}_\infty, \left(\mathcal{A}_\infty^{(K)}\right)^\dagger \boldsymbol{b}_\infty\rangle_0 = \mathbb{E}_0\|\bar{g}_\infty\|_\nu^2 \ . \tag{113}$$

This requires us to further exploit the relationship among $\mathcal{A}_\infty^{(K)}$, $\boldsymbol{b}_\infty$ and $\bar{g}_\infty$. With the Hilbert space $\mathcal{W}(\Omega)$ defined in Appendix A and $\mathcal{B}_t$ defined by (55), we can rewrite (51) as

$$\mathcal{A}_t^{(K)} = \mathcal{B}_t\mathcal{B}_t^\mathsf{T} \ . \tag{114}$$

Further, recall that

$$g_t = \int_D \varphi(\boldsymbol{\theta}, \cdot)\omega_t(d\boldsymbol{\theta}) = \int_D \varphi(\boldsymbol{\Theta}_t(\boldsymbol{\theta}), \cdot)\omega_0(d\boldsymbol{\theta}) + \int_D \nabla\varphi(\boldsymbol{\Theta}_t(\boldsymbol{\theta}), \cdot) \cdot \boldsymbol{T}_t(\boldsymbol{\theta})\mu_0(d\boldsymbol{\theta}) \tag{115}$$

$$\bar{g}_t = \int_D \varphi(\boldsymbol{\theta}, \cdot)\bar{\omega}_t(d\boldsymbol{\theta}) = \int_D \varphi(\boldsymbol{\Theta}_t(\boldsymbol{\theta}), \cdot)\omega_0(d\boldsymbol{\theta}) \ . \tag{116}$$

Therefore, we can write

$$g_t = \bar{g}_t + \mathcal{B}_t^\mathsf{T}\boldsymbol{T}_t \ , \tag{117}$$

and

$$\boldsymbol{b}_t = \mathcal{B}_t\bar{g}_t, \tag{118}$$

Similar formulas hold when we replace $t$ by $\infty$. With these relations, we see that

$$\left\langle \boldsymbol{b}_\infty, \left(\mathcal{A}_\infty^{(K)}\right)^\dagger \boldsymbol{b}_\infty \right\rangle_0 = \left\langle \mathcal{B}_\infty \bar{g}_\infty, (\mathcal{B}_\infty \mathcal{B}_\infty^\mathsf{T})^\dagger \mathcal{B}_\infty \bar{g}_\infty \right\rangle_0 \tag{119}$$
$$= \mathbb{E}_0 \| (\mathcal{B}_\infty)^\dagger \mathcal{B}_\infty \bar{g}_\infty \|_{\hat{\nu}}^2 ,$$

because $(\mathcal{B}_\infty \mathcal{B}_\infty^\mathsf{T})^\dagger = (\mathcal{B}_\infty^\mathsf{T})^\dagger (\mathcal{B}_\infty)^\dagger$. Since $(\mathcal{B}_\infty)^\dagger \mathcal{B}_\infty$ is the projection operator (which becomes an $n \times n$ matrix once we identify each random function in $\mathcal{W}(\Omega)$ with a random vector in $\mathbb{R}^n$) onto the range of $\mathcal{B}_\infty^\mathsf{T}$, it is then sufficient to prove that

**Lemma E.5** *Under Assumptions 2.1, 2.2, 2.3 and 2.4, $\mathbb{P}_0$-almost surely, $\bar{g}_\infty \in \mathrm{Ran}(\mathcal{B}_\infty^\mathsf{T})$.*

Lemma E.5 is proven in Appendix E.2.4 and it concludes the proof of (44) in Theorem 3.5.

To show that $\|g_t\|_{\hat{\nu}}$ decreases monotonically when $\mu_0 = \mu_\infty$, note that in this case $\mu_t = \mu_\infty, \forall t \geq 0$, and so $\mathcal{A}_t^{(V)} = \mathcal{A}_\infty^{(V)} = 0$, $\mathcal{A}_t^{(K)} = \mathcal{A}_\infty^{(K)}$ and $\boldsymbol{b}_t = \boldsymbol{b}_\infty, \forall t \geq 0$. Thus, (94) becomes

$$\dot{\boldsymbol{T}}_t = -\mathcal{A}_\infty^{(K)} \boldsymbol{T}_t - \boldsymbol{b}_\infty, \tag{120}$$

As will be shown in Lemma E.6, $\boldsymbol{b}_\infty$ is in the range of $\mathcal{A}_\infty^{(K)}$. Therefore, defining

$$\boldsymbol{u}_\infty = (\mathcal{A}_\infty^{(K)})^\dagger \boldsymbol{b}_\infty , \tag{121}$$

and

$$\boldsymbol{z}_t = \boldsymbol{T}_t + \boldsymbol{u}_\infty , \tag{122}$$

there is

$$\dot{\boldsymbol{z}}_t = -\mathcal{A}_\infty^{(K)} \boldsymbol{z}_t , \tag{123}$$

whose solution can be written analytically as

$$\boldsymbol{z}_t = e^{-t\mathcal{A}_\infty^{(K)}} \boldsymbol{z}_0 = e^{-t\mathcal{A}_\infty^{(K)}} \boldsymbol{u}_\infty . \tag{124}$$

Thus,

$$\boldsymbol{T}_t = \boldsymbol{z}_t - \boldsymbol{u}_\infty = -(I - e^{-t\mathcal{A}_\infty^{(K)}}) \boldsymbol{u}_\infty \tag{125}$$

Therefore, as $\boldsymbol{b}_\infty = \mathcal{B}_\infty \bar{g}_\infty$, there is

$$\begin{aligned} g_t &= \bar{g}_\infty + \mathcal{B}_\infty^\mathsf{T} \boldsymbol{T}_t \\ &= \bar{g}_\infty - \mathcal{B}_\infty^\mathsf{T} (I - e^{-t\mathcal{A}_\infty^{(K)}}) \boldsymbol{u}_\infty \\ &= \bar{g}_\infty - \mathcal{B}_\infty^\mathsf{T} (I - e^{-t\mathcal{A}_\infty^{(K)}}) (\mathcal{A}_\infty^{(K)})^\dagger \mathcal{B}_\infty \bar{g}_\infty . \end{aligned} \tag{126}$$

Hence,

$$|g_\infty|^2 = |\bar{g}_\infty|^2 - 2(*) + (**) , \tag{127}$$

where

$$\begin{aligned} (*) &= (\mathcal{B}_\infty \bar{g}_\infty)^\mathsf{T} (I - e^{-t\mathcal{A}_\infty^{(K)}}) (\mathcal{A}_\infty^{(K)})^\dagger \mathcal{B}_\infty \bar{g}_\infty \\ &= \boldsymbol{b}_\infty^\mathsf{T} (I - e^{-t\mathcal{A}_\infty^{(K)}}) (\mathcal{A}_\infty^{(K)})^\dagger \boldsymbol{b}_\infty \end{aligned} \tag{128}$$

and

$$\begin{aligned} (**) &= (\mathcal{B}_\infty \bar{g}_\infty)^\mathsf{T} (\mathcal{A}_\infty^{(K)})^\dagger (I - e^{-t\mathcal{A}_\infty^{(K)}}) \mathcal{B}_\infty \mathcal{B}_\infty^\mathsf{T} (I - e^{-t\mathcal{A}_\infty^{(K)}}) (\mathcal{A}_\infty^{(K)})^\dagger \mathcal{B}_\infty \bar{g}_\infty \\ &= \boldsymbol{b}_\infty^\mathsf{T} (I - e^{-t\mathcal{A}_\infty^{(K)}}) \mathcal{B}_\infty \mathcal{B}_\infty^\mathsf{T} (I - e^{-t\mathcal{A}_\infty^{(K)}}) \boldsymbol{b}_\infty . \end{aligned} \tag{129}$$

In the ERM setting, $\mathcal{A}_\infty^{(K)}$ is PSD with a finite number of nonzero eigenspaces. Consider a set of its orthonormal eigenfunctions that span those nonzero eigenspaces, $v_1, ..., v_k$, corresponding to eigenvalues $\lambda_1, ..., \lambda_k > 0$, respectively. As $\boldsymbol{b}_\infty$ is in the range of $\mathcal{A}_\infty^{(K)}$ by Lemma E.6, we can decompose it as

$$\boldsymbol{b}_\infty = \sum_{i=1}^k c_i v_i \tag{130}$$

for some real numbers $c_i$'s. Thus, we can write

$$
\begin{aligned}
(*) &= \left( \sum_{i=1}^{k} c_i v_i \right)^{\mathsf{T}} (I - e^{-t\mathcal{A}_{\infty}^{(K)}})(\mathcal{A}_{\infty}^{(K)})^{\dagger} \left( \sum_{j=1}^{k} c_j v_j \right) \\
&= \left( \sum_{i=1}^{k} c_i v_i \right)^{\mathsf{T}} \left( \sum_{j=1}^{k} c_j \lambda_j^{-1}(1 - e^{-\lambda_j t}) v_j \right) \\
&= \sum_{i=1}^{k} \lambda_j^{-1}(1 - e^{-\lambda_j t}) c_i^2 \,,
\end{aligned}
\tag{131}
$$

$$
\begin{aligned}
(**) &= \left( \sum_{i=1}^{k} c_i v_i \right)^{\mathsf{T}} (\mathcal{A}_{\infty}^{(K)})^{\dagger}(I - e^{-t\mathcal{A}_{\infty}^{(K)}})\mathcal{B}_{\infty}\mathcal{B}_{\infty}^{\mathsf{T}}(I - e^{-t\mathcal{A}_{\infty}^{(K)}})(\mathcal{A}_{\infty}^{(K)})^{\dagger} \left( \sum_{j=1}^{k} c_j v_j \right) \\
&= \left( \sum_{i=1}^{k} c_i v_i \right)^{\mathsf{T}} (\mathcal{A}_{\infty}^{(K)})^{\dagger}(I - e^{-t\mathcal{A}_{\infty}^{(K)}})\mathcal{A}_{\infty}^{(K)}(I - e^{-t\mathcal{A}_{\infty}^{(K)}})(\mathcal{A}_{\infty}^{(K)})^{\dagger} \left( \sum_{j=1}^{k} c_j v_j \right) \\
&= \left( \sum_{i=1}^{k} c_i v_i \right)^{\mathsf{T}} \left( \sum_{j=1}^{k} \lambda_j^{-1} \left( 1 - e^{-\lambda_j t} \right)^2 c_j v_j \right) \\
&= \sum_{i=1}^{k} \lambda_j^{-1} \left( 1 - e^{-\lambda_j t} \right)^2 c_i^2 \,.
\end{aligned}
\tag{132}
$$

Therefore,

$$
\begin{aligned}
|g_{\infty}|^2 &= |\bar{g}_{\infty}|^2 - 2 \sum_{i=1}^{k} \lambda_j^{-1} \left( 1 - e^{-\lambda_j t} \right) c_i^2 + \sum_{i=1}^{k} \lambda_j^{-1} \left( 1 - e^{-\lambda_j t} \right)^2 c_i^2 \\
&= |\bar{g}_{\infty}|^2 + \sum_{i=1}^{k} \lambda_j^{-1} \left( 1 - e^{-\lambda_j t} \right) \left( -1 - e^{-\lambda_j t} \right) c_i^2 \\
&= |\bar{g}_{\infty}|^2 - \sum_{i=1}^{k} \lambda_j^{-1} \left( 1 - e^{-2\lambda_j t} \right) c_i^2 \,,
\end{aligned}
\tag{133}
$$

which is decreasing in time. This completes the proof of Theorem 3.5. $\qquad\square$

### E.2.1  Proof of Lemma E.2

*Proof of* (102): $\int_0^{\infty} \|\mathcal{A}_t^{(V)}\|_0 dt < \infty$.

By the definition of the operator norm induced by $\|\cdot\|_0$ on $\mathcal{V}(D)$, $\|\mathcal{A}_t^{(V)}\|_0$ is the smallest number $C_t$ such that $\forall \boldsymbol{\xi}$, there is

$$
\|\mathcal{A}_t^{(V)}\|_0 = \sup_{\boldsymbol{\xi} \in \mathcal{V}(D), \|\boldsymbol{\xi}\|_0 \neq 0} \frac{\left| \langle \boldsymbol{\xi}, \mathcal{A}_t^{(V)} \boldsymbol{\xi} \rangle_0 \right|}{\|\boldsymbol{\xi}\|_0^2} \,.
\tag{134}
$$

In the unregularized case, a straightforward bound of $\left|\langle\boldsymbol{\xi},\mathcal{A}_t^{(V)}\boldsymbol{\xi}\rangle_0\right|$ is

$$
\begin{aligned}
\left|\langle\boldsymbol{\xi},\mathcal{A}_t^{(V)}\boldsymbol{\xi}\rangle_0\right| &= \left|\mathbb{E}_0\int_D\langle\boldsymbol{\xi}(\boldsymbol{\theta}),\nabla\nabla V(\boldsymbol{\Theta}_t(\boldsymbol{\theta}),\mu_t)\boldsymbol{\xi}(\boldsymbol{\theta})\rangle\mu_0(d\boldsymbol{\theta})\right|\\
&= \left|\mathbb{E}_0\int_D\int_\Omega\langle\boldsymbol{\xi}(\boldsymbol{\theta}),\nabla\nabla\varphi(\boldsymbol{\Theta}_t(\boldsymbol{\theta}),\boldsymbol{x})\boldsymbol{\xi}(\boldsymbol{\theta})\rangle(f_t(\boldsymbol{x})-f_*(\boldsymbol{x}))\hat{\nu}(d\boldsymbol{x})\,\mu_0(d\boldsymbol{\theta})\right|\\
&\leq \mathbb{E}_0\int_D\int_\Omega C_{\nabla\nabla\varphi}\,|\boldsymbol{\xi}(\boldsymbol{\theta})|^2\,|f_t(\boldsymbol{x})-f_*(\boldsymbol{x})|\,\hat{\nu}(d\boldsymbol{x})\mu_0(d\boldsymbol{\theta})\\
&= C_{\nabla\nabla\varphi}\|\boldsymbol{\xi}\|_0^2\int_\Omega|f_t(\boldsymbol{x})-f_*(\boldsymbol{x})|\,\hat{\nu}(d\boldsymbol{x})\\
&\leq n^{1/2}C_{\nabla\nabla\varphi}\|\boldsymbol{\xi}\|_0^2\|f_t-f_*\|_{\hat{\nu}}\\
&= n^{1/2}C_{\nabla\nabla\varphi}\|\boldsymbol{\xi}\|_0^2\,(\mathcal{L}(\mu_t))^{1/2}\ .
\end{aligned}
\tag{135}
$$

Thus, we have

$$
\|\mathcal{A}_t^{(V)}\|_0 \leq n^{1/2}C_{\nabla\nabla\varphi}\|\boldsymbol{\xi}\|_0^2\,(\mathcal{L}(\mu_t))^{1/2}\ .
\tag{136}
$$

By the assumption (43), we thus have

$$
\int_0^\infty\|\mathcal{A}_t^{(V)}\|_0 dt \leq n^{1/2}C_{\nabla\nabla\varphi}\int_0^\infty(\mathcal{L}(\mu_t))^{1/2}\,dt < \infty
\tag{137}
$$

which gives us the desired bound. $\qquad\square$

*Proof of* (103): $\int_0^\infty\|\mathcal{A}_\infty^{(K)}-\mathcal{A}_t^{(K)}\|_0 dt < \infty$.
We have

$$
\begin{aligned}
&\langle\boldsymbol{\xi},(\mathcal{A}_t^{(K)}-\mathcal{A}_\infty^{(K)})\boldsymbol{\xi}\rangle_0\\
&=\mathbb{E}_0\int_\Omega\left(\left(\int_D\nabla\varphi(\boldsymbol{\Theta}_t(\boldsymbol{\theta}),\boldsymbol{x})\cdot\boldsymbol{\xi}(\boldsymbol{\theta})\mu_0(d\boldsymbol{\theta})\right)^2-\left(\int_D\nabla\varphi(\boldsymbol{\Theta}_\infty(\boldsymbol{\theta}),\boldsymbol{x})\cdot\boldsymbol{\xi}(\boldsymbol{\theta})\mu_0(d\boldsymbol{\theta})\right)^2\right)\hat{\nu}(d\boldsymbol{x})\\
&=\mathbb{E}_0\int_\Omega\left(\int_D\big(\nabla\varphi(\boldsymbol{\Theta}_t(\boldsymbol{\theta}),\boldsymbol{x})+\nabla\varphi(\boldsymbol{\Theta}_\infty(\boldsymbol{\theta}),\boldsymbol{x})\big)\cdot\boldsymbol{\xi}(\boldsymbol{\theta})\mu_0(d\boldsymbol{\theta})\right)\\
&\quad\times\left(\int_D\big(\nabla\varphi(\boldsymbol{\Theta}_t(\boldsymbol{\theta}),\boldsymbol{x})-\nabla\varphi(\boldsymbol{\Theta}_\infty(\boldsymbol{\theta}),\boldsymbol{x})\big)\cdot\boldsymbol{\xi}(\boldsymbol{\theta})\mu_0(d\boldsymbol{\theta})\right)\hat{\nu}(d\boldsymbol{x})\ .
\end{aligned}
\tag{138}
$$

Hence, the absolute value of the expression above is upper-bounded by

$$
\begin{aligned}
&\mathbb{E}_0\Big(\int_D|\nabla\varphi(\boldsymbol{\Theta}_t(\boldsymbol{\theta}),\boldsymbol{x})+\nabla\varphi(\boldsymbol{\Theta}_\infty(\boldsymbol{\theta}),\boldsymbol{x})||\boldsymbol{\xi}(\boldsymbol{\theta})|\mu_0(d\boldsymbol{\theta})\\
&\quad\times\int_D|\nabla\varphi(\boldsymbol{\Theta}_t(\boldsymbol{\theta}),\boldsymbol{x})-\nabla\varphi(\boldsymbol{\Theta}_\infty(\boldsymbol{\theta}),\boldsymbol{x})||\boldsymbol{\xi}(\boldsymbol{\theta})|\mu_0(d\boldsymbol{\theta})\Big)\\
&\leq 2C_{\nabla\varphi}C_{\nabla\nabla\varphi}\|\boldsymbol{\xi}\|_0^2\Big(\int_D|\boldsymbol{\Theta}_t(\boldsymbol{\theta})-\boldsymbol{\Theta}_\infty(\boldsymbol{\theta})|^2\mu_0(d\boldsymbol{\theta})\Big)^{1/2}\ .
\end{aligned}
\tag{139}
$$

Thus, by the assumption (43), we have

$$
\begin{aligned}
\int_0^\infty\|\mathcal{A}_\infty^{(K)}-\mathcal{A}_t^{(K)}\|_0 dt &\leq 2C_{\nabla\varphi}C_{\nabla\nabla\varphi}\int_0^\infty\left(\int_D|\boldsymbol{\Theta}_t(\boldsymbol{\theta})-\boldsymbol{\Theta}_\infty(\boldsymbol{\theta})|^2\mu_0(d\boldsymbol{\theta})\right)^{1/2}dt\\
&\leq 2C_{\nabla\varphi}C_{\nabla\nabla\varphi}\int_0^\infty(\mathcal{L}(\mu_t))^{1/2}\,dt\\
&<\infty
\end{aligned}
\tag{140}
$$

$\qquad\square$

*Proof of* (104): $\int_0^\infty\|\boldsymbol{b}_t-\boldsymbol{b}_\infty\|_0 dt < \infty$.

There is

$$
\boldsymbol{b}_t(\boldsymbol{\theta}) - \boldsymbol{b}_\infty(\boldsymbol{\theta})
$$
$$
= \int_D \left( \nabla K(\boldsymbol{\Theta}_t(\boldsymbol{\theta}), \boldsymbol{\Theta}_t(\boldsymbol{\theta}')) - \nabla K(\boldsymbol{\Theta}_\infty(\boldsymbol{\theta}), \boldsymbol{\Theta}_\infty(\boldsymbol{\theta}'))) \omega_0(d\boldsymbol{\theta}')
$$
$$
= \int_D \int_\Omega \nabla\varphi(\boldsymbol{\Theta}_t(\boldsymbol{\theta}), \boldsymbol{x}) \cdot \nabla\varphi(\boldsymbol{\Theta}_t(\boldsymbol{\theta}'), \boldsymbol{x})^\mathsf{T} - \nabla\varphi(\boldsymbol{\Theta}_\infty(\boldsymbol{\theta}), \boldsymbol{x}) \cdot \nabla\varphi(\boldsymbol{\Theta}_\infty(\boldsymbol{\theta}'), \boldsymbol{x})^\mathsf{T} \hat{\nu}(d\boldsymbol{x}) \omega_0(d\boldsymbol{\theta}')
$$
$$
= \int_D \int_\Omega \nabla\varphi(\boldsymbol{\Theta}_t(\boldsymbol{\theta}), \boldsymbol{x}) \cdot \nabla\varphi(\boldsymbol{\Theta}_t(\boldsymbol{\theta}'), \boldsymbol{x})^\mathsf{T} - \nabla\varphi(\boldsymbol{\Theta}_t(\boldsymbol{\theta}), \boldsymbol{x}) \cdot \nabla\varphi(\boldsymbol{\Theta}_\infty(\boldsymbol{\theta}'), \boldsymbol{x})^\mathsf{T} \hat{\nu}(d\boldsymbol{x}) \omega_0(d\boldsymbol{\theta}')
$$
$$
+ \int_D \int_\Omega \nabla\varphi(\boldsymbol{\Theta}_t(\boldsymbol{\theta}), \boldsymbol{x}) \cdot \nabla\varphi(\boldsymbol{\Theta}_\infty(\boldsymbol{\theta}'), \boldsymbol{x})^\mathsf{T} - \nabla\varphi(\boldsymbol{\Theta}_\infty(\boldsymbol{\theta}), \boldsymbol{x}) \cdot \nabla\varphi(\boldsymbol{\Theta}_\infty(\boldsymbol{\theta}'), \boldsymbol{x})^\mathsf{T} \omega_0(d\boldsymbol{\theta}')
$$
$$
= \int_\Omega \nabla\varphi(\boldsymbol{\Theta}_t(\boldsymbol{\theta}), \boldsymbol{x}) \cdot \left( \int_D \left( \nabla\varphi(\boldsymbol{\Theta}_t(\boldsymbol{\theta}'), \boldsymbol{x}) - \nabla\varphi(\boldsymbol{\Theta}_\infty(\boldsymbol{\theta}'), \boldsymbol{x}) \right) \omega_0(d\boldsymbol{\theta}') \right)^\mathsf{T} \hat{\nu}(d\boldsymbol{x})
$$
$$
+ \int_\Omega \left( \nabla\varphi(\boldsymbol{\Theta}_t(\boldsymbol{\theta}), \boldsymbol{x}) - \nabla\varphi(\boldsymbol{\Theta}_\infty(\boldsymbol{\theta}), \boldsymbol{x}) \right) \left( \int_D \nabla\varphi(\boldsymbol{\Theta}_\infty(\boldsymbol{\theta}'), \boldsymbol{x}) \omega_0(d\boldsymbol{\theta}') \right)^\mathsf{T} \hat{\nu}(d\boldsymbol{x}) \, .
$$

(141)

Thus,

$$
\mathbb{E}_0 \left| \boldsymbol{b}_t(\boldsymbol{\theta}) - \boldsymbol{b}_\infty(\boldsymbol{\theta}) \right|^2
$$
$$
\leq \mathbb{E}_0 \left| \int_\Omega \nabla\varphi(\boldsymbol{\Theta}_t(\boldsymbol{\theta}), \boldsymbol{x}) \cdot \left( \int_D \left( \nabla\varphi(\boldsymbol{\Theta}_t(\boldsymbol{\theta}'), \boldsymbol{x}) - \nabla\varphi(\boldsymbol{\Theta}_\infty(\boldsymbol{\theta}'), \boldsymbol{x}) \right) \omega_0(d\boldsymbol{\theta}') \right)^\mathsf{T} \hat{\nu}(d\boldsymbol{x}) \right|^2
$$
$$
+ \mathbb{E}_0 \left| \int_\Omega \left( \nabla\varphi(\boldsymbol{\Theta}_t(\boldsymbol{\theta}), \boldsymbol{x}) - \nabla\varphi(\boldsymbol{\Theta}_\infty(\boldsymbol{\theta}), \boldsymbol{x}) \right) \left( \int_D \nabla\varphi(\boldsymbol{\Theta}_\infty(\boldsymbol{\theta}'), \boldsymbol{x}) \omega_0(d\boldsymbol{\theta}') \right)^\mathsf{T} \hat{\nu}(d\boldsymbol{x}) \right|^2
$$
$$
\leq \int_\Omega \left| \nabla\varphi(\boldsymbol{\Theta}_t(\boldsymbol{\theta}), \boldsymbol{x}) \right|^2 \mathbb{E}_0 \left| \int_D \left( \nabla\varphi(\boldsymbol{\Theta}_t(\boldsymbol{\theta}'), \boldsymbol{x}) - \nabla\varphi(\boldsymbol{\Theta}_\infty(\boldsymbol{\theta}'), \boldsymbol{x}) \right) \omega_0(d\boldsymbol{\theta}') \right|^2 \hat{\nu}(d\boldsymbol{x})
$$
$$
+ \int_\Omega \left| \nabla\varphi(\boldsymbol{\Theta}_t(\boldsymbol{\theta}), \boldsymbol{x}) - \nabla\varphi(\boldsymbol{\Theta}_\infty(\boldsymbol{\theta}), \boldsymbol{x}) \right|^2 \mathbb{E}_0 \left| \int_D \nabla\varphi(\boldsymbol{\Theta}_\infty(\boldsymbol{\theta}'), \boldsymbol{x}) \omega_0(d\boldsymbol{\theta}') \right|^2 \hat{\nu}(d\boldsymbol{x})
$$
$$
\leq C_{\nabla\varphi}^2 \int_\Omega \mathbb{E}_0 \left| \int_D \left( \nabla\varphi(\boldsymbol{\Theta}_t(\boldsymbol{\theta}'), \boldsymbol{x}) - \nabla\varphi(\boldsymbol{\Theta}_\infty(\boldsymbol{\theta}'), \boldsymbol{x}) \right) \omega_0(d\boldsymbol{\theta}') \right|^2 \hat{\nu}(d\boldsymbol{x})
$$
$$
+ C_{\nabla\nabla\varphi}^2 \left| \boldsymbol{\Theta}_t(\boldsymbol{\theta}) - \boldsymbol{\Theta}_\infty(\boldsymbol{\theta}) \right|^2 \int_\Omega \mathbb{E}_0 \left| \int_D \nabla\varphi(\boldsymbol{\Theta}_\infty(\boldsymbol{\theta}'), \boldsymbol{x}) \omega_0(d\boldsymbol{\theta}') \right|^2 \hat{\nu}(d\boldsymbol{x}) \, .
$$

(142)

By the property of $\omega_0$, there is

$$
\mathbb{E}_0 \left| \int_D \chi(\boldsymbol{\theta}) \omega_0(d\boldsymbol{\theta}) \right|^2 = \int_D \left| \chi(\boldsymbol{\theta}) - \int_D \chi(\boldsymbol{\theta}') \mu_0(d\boldsymbol{\theta}') \right|^2 \mu_0(d\boldsymbol{\theta})
$$
$$
\leq \int_D \left| \chi(\boldsymbol{\theta}) \right|^2 \mu_0(d\boldsymbol{\theta})
$$

(143)

for a test function $\chi$ on $D$. Thus,

$$
\mathbb{E}_0 \left| \boldsymbol{b}_t(\boldsymbol{\theta}) - \boldsymbol{b}_\infty(\boldsymbol{\theta}) \right|^2 \leq C_{\nabla\varphi}^2 \int_\Omega \int_D \left| \nabla\varphi(\boldsymbol{\Theta}_t(\boldsymbol{\theta}'), \boldsymbol{x}) - \nabla\varphi(\boldsymbol{\Theta}_\infty(\boldsymbol{\theta}'), \boldsymbol{x}) \right|^2 \mu_0(d\boldsymbol{\theta}')
$$
$$
+ C_{\nabla\nabla\varphi}^2 \left| \boldsymbol{\Theta}_t(\boldsymbol{\theta}) - \boldsymbol{\Theta}_\infty(\boldsymbol{\theta}) \right|^2 \int_\Omega \int_D \left| \nabla\varphi(\boldsymbol{\Theta}_\infty(\boldsymbol{\theta}'), \boldsymbol{x}) \right|^2 \mu_0(d\boldsymbol{\theta}') \hat{\nu}(d\boldsymbol{x})
$$
$$
\leq C_{\nabla\varphi}^2 C_{\nabla\nabla\varphi}^2 \int_D \left| \boldsymbol{\Theta}_t(\boldsymbol{\theta}') - \boldsymbol{\Theta}_\infty(\boldsymbol{\theta}') \right|^2 \mu_0(d\boldsymbol{\theta}')
$$
$$
+ C_{\nabla\nabla\varphi}^2 C_{\nabla\varphi}^2 \left| \boldsymbol{\Theta}_t(\boldsymbol{\theta}) - \boldsymbol{\Theta}_\infty(\boldsymbol{\theta}) \right|^2 \, .
$$

(144)

Therefore,

$$
\begin{aligned}
\|\boldsymbol{b}_t - \boldsymbol{b}_\infty\|_0^2 = & \mathbb{E}_0 \int_D |\boldsymbol{b}_t(\boldsymbol{\theta}) - \boldsymbol{b}_\infty(\boldsymbol{\theta})|^2 \mu_0(d\boldsymbol{\theta}) \\
\leq & 2C_{\nabla\varphi}^2 C_{\nabla\nabla\varphi}^2 \int_D |\boldsymbol{\Theta}_t(\boldsymbol{\theta}) - \boldsymbol{\Theta}_\infty(\boldsymbol{\theta})|^2 \mu_0(d\boldsymbol{\theta}) \ .
\end{aligned}
\tag{145}
$$

Since

$$
\begin{aligned}
\int_D |\boldsymbol{\Theta}_t(\boldsymbol{\theta}) - \boldsymbol{\Theta}_\infty(\boldsymbol{\theta})|^2 \mu_0(d\boldsymbol{\theta}) \leq & \int_D \int_t^\infty \left| \dot{\boldsymbol{\Theta}}_s(\boldsymbol{\theta}) \right|^2 ds \mu_0(d\boldsymbol{\theta}) \\
= & \int_D \int_t^\infty |\nabla V(\boldsymbol{\Theta}_t(\boldsymbol{\theta}), \mu_t)|^2 ds \mu_0(d\boldsymbol{\theta}) \\
= & -\int_t^\infty \frac{d}{ds} \mathcal{L}(\mu_s) ds \\
= & \mathcal{L}(\mu_t) - \mathcal{L}(\mu_\infty) \\
= & \mathcal{L}(\mu_t)
\end{aligned}
\tag{146}
$$

we can conclude that

$$
\int_0^\infty \|\boldsymbol{b}_t - \boldsymbol{b}_\infty\|_0 dt \leq \int_0^\infty |\mathcal{L}(\mu_t)|^{1/2} \, dt < \infty \ .
\tag{147}
$$

$\square$

### E.2.2  Proof of Lemma E.3

Our goal is to show that $\|\boldsymbol{T}_t\|_0$ remains bounded for all time. First note that, for all $t$, $\mathcal{A}_t^{(K)}$ is a positive semidefinite (PSD) operator on $\mathcal{V}(D)$ since

$$
\begin{aligned}
\langle \mathcal{A}_t^{(K)} \boldsymbol{\xi}, \boldsymbol{\xi} \rangle_0 = & \mathbb{E}_0 \int_{D \times D} \langle \boldsymbol{\xi}(\boldsymbol{\theta}), \nabla\nabla' K(\boldsymbol{\Theta}_t(\boldsymbol{\theta}), \boldsymbol{\Theta}_t(\boldsymbol{\theta}')) \boldsymbol{\xi}(\boldsymbol{\theta}') \rangle \mu_0(d\boldsymbol{\theta}) \mu_0(d\boldsymbol{\theta}') \\
= & \mathbb{E}_0 \int_\Omega \left| \int_D \nabla\varphi(\boldsymbol{\Theta}_t(\boldsymbol{\theta})) \cdot \boldsymbol{\xi}(\boldsymbol{\theta}) \mu_0(d\boldsymbol{\theta}) \right|^2 \hat{\nu}(d\boldsymbol{x}) \geq 0 \ .
\end{aligned}
\tag{148}
$$

Second, by Assumption 2.4, for $\mu_0$-almost-every $\boldsymbol{\theta} \in D$, $\boldsymbol{\Theta}_\infty(\boldsymbol{\theta}) = \lim_{t\to\infty} \boldsymbol{\Theta}_t(\boldsymbol{\theta})$ exists, which allows us to define $\boldsymbol{b}_\infty$, $\mathcal{A}_\infty^{(K)}$, and $\mathcal{A}_\infty^{(V)}$ similarly to (49), (51) and (52) by replacing $\boldsymbol{\Theta}_t(\cdot)$ with $\boldsymbol{\Theta}_\infty(\cdot)$. Since we assume that

$$
\forall \boldsymbol{x}_k \in \operatorname{supp} \hat{\nu} \quad : \quad f_\infty(\boldsymbol{x}_k) = \int_D \varphi(\boldsymbol{\theta}, \boldsymbol{x}_k) \mu_\infty(d\boldsymbol{\theta}) = f_*(\boldsymbol{x}_k)
\tag{149}
$$

we have

$$
\forall \boldsymbol{\theta} \in D \quad : \quad \nabla\nabla V(\boldsymbol{\theta}, \mu_\infty) = \int_\Omega \nabla\nabla\varphi(\boldsymbol{\theta}, \boldsymbol{x})(f_\infty(\boldsymbol{x}) - f_*(\boldsymbol{x})) d\boldsymbol{x} = 0 \ .
\tag{150}
$$

This implies that $\mathcal{A}_\infty^{(V)}$ is the zero operator on $\mathcal{V}(D)$.

Third, we have the following observation:

**Lemma E.6** *Under Assumptions 2.2, 2.3 and 2.4, $\boldsymbol{b}_t \in \operatorname{Ran}(\mathcal{A}_t^{(K)})$ for all $t$, and $\boldsymbol{b}_\infty \in \operatorname{Ran}(\mathcal{A}_\infty^{(K)})$. Specifically, $\exists \tilde{\boldsymbol{u}}_\infty \in \mathcal{V}(D)$ such that $\|\boldsymbol{u}_\infty\|_0 < \infty$ and $\mathcal{A}_\infty^{(K)} \tilde{\boldsymbol{u}}_\infty = \boldsymbol{b}_\infty$.*

*Proof of Lemma E.6:* Recall from (118) that $\boldsymbol{b}_\infty = \mathcal{B}_\infty \bar{g}_\infty$. Define $\tilde{\boldsymbol{u}}_\infty = \mathcal{B}_\infty (\mathcal{B}_\infty^\mathsf{T} \mathcal{B}_\infty)^\dagger \bar{g}_\infty$. We claim that $\mathcal{A}_\infty^{(K)} \tilde{\boldsymbol{u}}_\infty = \boldsymbol{b}_\infty$, because

$$
\begin{aligned}
\mathcal{A}_\infty^{(K)} \tilde{\boldsymbol{u}}_\infty = & \left( \mathcal{B}_\infty \mathcal{B}_\infty^\mathsf{T} \right) \mathcal{B}_\infty \left( \mathcal{B}_\infty^\mathsf{T} \mathcal{B}_\infty \right)^\dagger \bar{g}_\infty \\
= & \mathcal{B}_\infty \mathcal{B}_\infty^\mathsf{T} \left( \mathcal{B}_\infty \left( \mathcal{B}_\infty \right)^\dagger \right) (\mathcal{B}_\infty^\mathsf{T})^\dagger \bar{g}_\infty \\
= & \mathcal{B}_\infty \left( \mathcal{B}_\infty^\mathsf{T} (\mathcal{B}_\infty^\mathsf{T})^\dagger \right) \bar{g}_\infty \\
= & \mathcal{B}_\infty \bar{g}_\infty \\
= & \boldsymbol{b}_\infty \ ,
\end{aligned}
\tag{151}
$$

where the third equality is because $\mathcal{B}_\infty \,(\mathcal{B}_\infty)^\dagger$ is the projection operator onto $\mathrm{Ran}(\mathcal{B}_\infty) = \mathrm{Nul}^\perp(\mathcal{B}_\infty^\mathsf{T})$, and the fourth equality is because $\mathcal{B}_\infty^\mathsf{T}\,(\mathcal{B}_\infty^\mathsf{T})^\dagger$ is the projection operator onto $\mathrm{Ran}(\mathcal{B}_\infty^\mathsf{T}) = \mathrm{Nul}^\perp(\mathcal{B}_\infty)$.

It remains to establish that $\|\tilde{\boldsymbol{u}}_\infty\|_0 < \infty$. To show this, we see that

$$
\begin{aligned}
&\int_D |\tilde{\boldsymbol{u}}_\infty(\boldsymbol{\theta})|^2 \mu_0(d\boldsymbol{\theta})\\
&= \int_D \int_{\Omega \times \Omega} \Big( \nabla\varphi(\boldsymbol{\Theta}_\infty(\boldsymbol{\theta}), \boldsymbol{x})\big(\mathcal{M}_\infty^\dagger \bar{g}_\infty\big)(\boldsymbol{x})\Big)\\
&\qquad\qquad \cdot \Big(\nabla\varphi(\boldsymbol{\Theta}_\infty(\boldsymbol{\theta}), \boldsymbol{x}')\big((\mathcal{M}_\infty)_\infty^\dagger \bar{g}_\infty\big)(\boldsymbol{x}')\Big)\hat{\nu}(d\boldsymbol{x})\hat{\nu}(d\boldsymbol{x}')\mu_0(d\boldsymbol{\theta}')\\
&= \int_\Omega \int_\Omega M(\boldsymbol{x},\boldsymbol{x}',\mu_\infty)\big(\mathcal{M}_\infty^\dagger \bar{g}_\infty\big)(\boldsymbol{x})\big(\mathcal{M}_\infty^\dagger \bar{g}_\infty\big)(\boldsymbol{x}')\hat{\nu}(d\boldsymbol{x})\hat{\nu}(d\boldsymbol{x}')\\
&= \int_\Omega \big(\mathcal{M}_\infty^\dagger \bar{g}_\infty\big)(\boldsymbol{x}) \cdot \bar{g}_\infty(\boldsymbol{x})\hat{\nu}(d\boldsymbol{x})\\
&\le \lambda_{\min}^{-1} \int_\Omega |\bar{g}_\infty(\boldsymbol{x})|^2 \hat{\nu}(d\boldsymbol{x})\,,
\end{aligned}
\tag{152}
$$

where $\lambda_{\min}$ is the least nonzero eigenvalue of the matrix $\mathcal{M}_\infty$ (and hence $\lambda_{\min}^{-1}$ is the largest eigenvalue of $\mathcal{M}_\infty^\dagger$). Since

$$
\begin{aligned}
\mathbb{E}_0|\bar{g}_\infty(\boldsymbol{x})|^2 &= \mathbb{E}_0 \Big| \int_D \varphi(\boldsymbol{\Theta}_\infty(\boldsymbol{\theta}), \boldsymbol{x})\omega_0(d\boldsymbol{\theta}) \Big|^2\\
&= \int_D \Big( \varphi(\boldsymbol{\Theta}_\infty(\boldsymbol{\theta}), \boldsymbol{x}) - \int_D \varphi(\boldsymbol{\Theta}_\infty(\boldsymbol{\theta}'), \boldsymbol{x})\mu_0(d\boldsymbol{\theta}') \Big)^2 \mu_0(d\boldsymbol{\theta})\\
&\le \int_D \big|\varphi(\boldsymbol{\Theta}_\infty(\boldsymbol{\theta}), \boldsymbol{x})\big|^2 \mu_0(d\boldsymbol{\theta})\,,
\end{aligned}
\tag{153}
$$

there is

$$
\begin{aligned}
\|\tilde{\boldsymbol{u}}_\infty\|_0^2 &\le \mathbb{E}_0 \int_D |\tilde{\boldsymbol{u}}_\infty(\boldsymbol{\theta})|^2 \mu_0(d\boldsymbol{\theta})\\
&\le \lambda_{\min}^{-1} \int_\Omega \int_D \big(\varphi(\boldsymbol{\Theta}_\infty(\boldsymbol{\theta}), \boldsymbol{x})\big)^2 \mu_0(d\boldsymbol{\theta})\nu(d\boldsymbol{x})\\
&\le \lambda_{\min}^{-1} C_\varphi^2 < \infty\,,
\end{aligned}
\tag{154}
$$

(*End of the proof of Lemma E.6*) □

Coming back to the prof of Lemma E.3, we have shown that, as $t \to \infty$, (94) approaches the asymptotic dynamics

$$
\dot{\boldsymbol{T}}_t = -\mathcal{A}_\infty^{(K)}\boldsymbol{T}_t - \boldsymbol{b}_\infty,
\tag{155}
$$

with $\mathcal{A}_\infty^{(K)}$ positive semidefinite and $\boldsymbol{b}_\infty$ in the range of $\mathcal{A}_\infty^{(K)}$. This is a stable system. Hence, the rest of the task is to examine what happens at finite time. To do so, we perform a change-of-variable with

$$
\boldsymbol{z}_t = \boldsymbol{T}_t + \tilde{\boldsymbol{u}}_\infty,
\tag{156}
$$

with

$$
\boldsymbol{u}_\infty = \mathcal{B}_\infty(\mathcal{B}_\infty^\mathsf{T}\mathcal{B}_\infty)^\dagger \bar{g}_\infty
\tag{157}
$$

as is defined in the proof of Lemma E.6. The dynamics of $\boldsymbol{z}_t$ is governed by

$$
\begin{aligned}
\dot{\boldsymbol{z}}_t = \dot{\boldsymbol{T}}_t &= -(\mathcal{A}_t^{(K)} + \mathcal{A}_t^{(V)})\boldsymbol{T}_t - \boldsymbol{b}_t\\
&= -\mathcal{A}_t^{(K)}\boldsymbol{z}_t - \mathcal{A}_t^{(V)}\boldsymbol{z}_t - \big(\boldsymbol{b}_t - (\mathcal{A}_t^{(K)} + \mathcal{A}_t^{(V)})\tilde{\boldsymbol{u}}_\infty\big)\,.
\end{aligned}
\tag{158}
$$

Thus, in integral form,

$$
\boldsymbol{z}_t = \Pi(t,0)\boldsymbol{z}_0 + \int_0^t \Pi(t,s)\big(-\mathcal{A}_s^{(V)}\boldsymbol{z}_s - (\boldsymbol{b}_s - (\mathcal{A}_s^{(K)} + \mathcal{A}_s^{(V)})\tilde{\boldsymbol{u}}_\infty)\big)ds,
\tag{159}
$$

where $\Pi(t, s)$ is the fundamental solution (a.k.a. Green's function) associated with the time-variant homogeneous system

$$\dot{z}_t = -\mathcal{A}_t^{(K)} z_t . \tag{160}$$

Since $\mathcal{A}_t^{(K)}$ is positive semidefinite for all $t$, there is $\|\Pi(t, s)\|_0 \leq 1$ for $t > s$, where with a slight abuse of notation we also use $\| \cdot \|_0$ for the operator norm. Hence,

$$
\begin{aligned}
\|z_t\|_0 \leq & \|\Pi(t, 0)\|_0 \|z_0\|_0 + \int_0^t \|\Pi(t, s)\|_0 \Big( \|\mathcal{A}_s^{(V)}\|_0 \|z_s\|_0 + \|b_s - (\mathcal{A}_s^{(K)} + \mathcal{A}_s^{(V)}) \tilde{u}_\infty\|_0 \Big) ds \\
\leq & \|z_0\|_0 + \int_0^t \Big( \|\mathcal{A}_s^{(V)}\|_0 \|z_s\|_0 + \|b_s - (\mathcal{A}_s^{(K)} + \mathcal{A}_s^{(V)}) \tilde{u}_\infty\|_0 \Big) ds .
\end{aligned}
\tag{161}
$$

By Grönwall's inequality, we thus have

$$\|z_t\|_0 \leq \Big( \|z_0\|_0 + \int_0^t \|b_s - (\mathcal{A}_s^{(K)} + \mathcal{A}_s^{(V)}) \tilde{u}_\infty\|_0 ds \Big) e^{\int_0^t \|\mathcal{A}_s^{(V)}\|_0 ds} . \tag{162}$$

Therefore, $\|z_t\|_0$ remains bounded for all time if we can show that

$$\int_0^\infty \|b_t - (\mathcal{A}_t^{(K)} + \mathcal{A}_t^{(V)}) \tilde{u}_\infty\|_0 dt < \infty, \qquad \int_0^\infty \|\mathcal{A}_t^{(V)}\|_0 dt < \infty . \tag{163}$$

Since

$$\|b_t - (\mathcal{A}_t^{(K)} + \mathcal{A}_t^{(V)}) \tilde{u}_\infty\|_0 \leq \|b_t - b_\infty\|_0 + \|(\mathcal{A}_t^{(K)} - \mathcal{A}_\infty^{(K)}) \tilde{u}_\infty\|_0 + \|\mathcal{A}_\infty^{(V)} \tilde{u}_\infty\|_0 \tag{164}$$

we see that (163) is guaranteed by Lemmas E.2 and E.6.

This completes the proof of Lemma E.3. $\qquad\square$

### E.2.3 Proof of Lemma E.4

From E.3, we have that

$$\lim_{t\to\infty} \left\| \fint_0^t \dot{T}_s ds \right\|_0 = \lim_{t\to\infty} \left\| \frac{1}{t} (T_t - T_0) ds \right\|_0 = 0 . \tag{165}$$

By (94), we then obtain that

$$\lim_{t\to\infty} \left\| \fint_0^t \Big( \mathcal{A}_s^{(K)} T_s + b_s \Big) ds + \fint_0^t \mathcal{A}_s^{(V)} T_s ds \right\|_0 = 0 . \tag{166}$$

By (102) in Lemma E.2 as well as Lemma E.3, we know that

$$\lim_{t\to\infty} \left\| \fint_0^t \mathcal{A}_s^{(V)} T_s ds \right\|_0 = 0 . \tag{167}$$

Therefore,

$$\lim_{t\to\infty} \left\| \fint_0^t \Big( \mathcal{A}_s^{(K)} T_s + b_s \Big) ds \right\|_0 = 0 . \tag{168}$$

Next, by (103) and (104) in Lemma E.2 as well as Lemma E.3, we know that

$$\lim_{t\to\infty} \left\| \fint_0^t \Big( \mathcal{A}_s^{(K)} T_s + b_s \Big) ds - \fint_0^t \Big( \mathcal{A}_\infty^{(K)} T_s + b_\infty \Big) ds \right\|_0 = 0 . \tag{169}$$

Therefore,

$$\lim_{t\to\infty} \left\| \fint_0^t \Big( \mathcal{A}_\infty^{(K)} T_s + b_\infty \Big) ds \right\|_0 = 0 . \tag{170}$$

With $\tilde{u}_\infty$ defined in (157), as $b_\infty = \mathcal{A}_\infty^{(K)} u_\infty$, there is

$$\lim_{t\to\infty} \left\| \mathcal{A}_\infty^{(K)} \Big( \fint_0^t T_s ds - u_\infty \Big) \right\|_0 = 0 . \tag{171}$$

Let $\boldsymbol{\xi}^{\|}$ denote the component of a vector field $\boldsymbol{\xi} \in \mathcal{V}(D)$ that is in the range of $\mathcal{A}_\infty^{(K)}$. In the ERM setting, $\mathcal{A}_\infty^{(K)}$ has a least nonzero eigenvalue that is positive, and hence the above implies that

$$\lim_{t\to\infty} \left\| \left( \fint_0^t \boldsymbol{T}_s ds - \tilde{\boldsymbol{u}}_\infty \right)^{\|} \right\|_0 = 0 \tag{172}$$

or

$$\lim_{t\to\infty} \left\| \left( \fint_0^t \boldsymbol{T}_s ds \right)^{\|} - \tilde{\boldsymbol{u}}_\infty \right\|_0 = 0 \tag{173}$$

and therefore, as $\mathrm{Nul}(\mathcal{A}_\infty^{(K)}) = \mathrm{Nul}(\mathcal{B}_\infty \mathcal{B}_\infty^\mathsf{T}) = \mathrm{Nul}(\mathcal{B}_\infty^\mathsf{T})$, it follows that

$$\lim_{t\to\infty} \left\| \mathcal{B}_\infty^\mathsf{T} \left( \fint_0^t \boldsymbol{T}_s ds \right) - \mathcal{B}_\infty^\mathsf{T} \tilde{\boldsymbol{u}}_\infty \right\|_0 = 0 \,. \tag{174}$$

Similar to (103), it can be shown that $\int_0^\infty \|\mathcal{B}_t - \mathcal{B}_\infty\|_0 dt < \infty$. Therefore, we have

$$\lim_{t\to\infty} \left\| \left( \fint_0^t \mathcal{B}_s^\mathsf{T} \boldsymbol{T}_s ds \right) - \mathcal{B}_\infty^\mathsf{T} \tilde{\boldsymbol{u}}_\infty \right\|_0 = 0 \,. \tag{175}$$

Now,

$$\fint_0^t \langle \boldsymbol{T}_s, \mathcal{A}_s^{(K)} \boldsymbol{T}_s \rangle_0 ds = \fint_0^t \langle \mathcal{B}_s^\mathsf{T} \boldsymbol{T}_s, \mathcal{B}_s^\mathsf{T} \boldsymbol{T}_s \rangle_{\hat{\nu},0} ds$$
$$\geq \left\langle \left( \fint_0^t \mathcal{B}_s^\mathsf{T} \boldsymbol{T}_s ds \right), \left( \fint_0^t \mathcal{B}_s^\mathsf{T} \boldsymbol{T}_s ds \right) \right\rangle_{\hat{\nu},0} \,. \tag{176}$$

Hence,

$$\lim_{t\to\infty} \fint_0^t \langle \boldsymbol{T}_s, \mathcal{A}_s^{(K)} \boldsymbol{T}_s \rangle_0 ds \geq \lim_{t\to\infty} \left\langle \left( \fint_0^t \mathcal{B}_s^\mathsf{T} \boldsymbol{T}_s ds \right), \left( \fint_0^t \mathcal{B}_s^\mathsf{T} \boldsymbol{T}_s ds \right) \right\rangle_{\hat{\nu},0}$$
$$= \langle \mathcal{B}_\infty^\mathsf{T} \tilde{\boldsymbol{u}}_\infty, \mathcal{B}_\infty^\mathsf{T} \tilde{\boldsymbol{u}}_\infty \rangle_{\hat{\nu},0}$$
$$= \left\langle \mathcal{B}_\infty^\mathsf{T} \left( \mathcal{A}_\infty^{(K)} \right)^\dagger \boldsymbol{b}_\infty, \mathcal{B}_\infty^\mathsf{T} \left( \mathcal{A}_\infty^{(K)} \right)^\dagger \boldsymbol{b}_\infty \tilde{\boldsymbol{u}}_\infty \right\rangle_{\hat{\nu},0} \tag{177}$$
$$= \left\langle \left( \mathcal{A}_\infty^{(K)} \right)^\dagger \boldsymbol{b}_\infty, \left( \mathcal{A}_\infty^{(K)} \right) \left( \mathcal{A}_\infty^{(K)} \right)^\dagger \boldsymbol{b}_\infty \right\rangle_0$$
$$= \left\langle \boldsymbol{b}_\infty, \left( \mathcal{A}_\infty^{(K)} \right)^\dagger \boldsymbol{b}_\infty \right\rangle_0 \,.$$

$\square$

### E.2.4   Proof of Lemma E.5

Since

$$\bar{g}_\infty(\boldsymbol{x}) = \int_D \varphi(\boldsymbol{\theta}, \boldsymbol{x}) \omega_0(d\boldsymbol{\theta}) \,, \tag{178}$$

we know that when viewed as an $n$-dimensional random vector, $\bar{g}_\infty$ has the distribution

$$\bar{g}_\infty \sim \mathcal{N}(0, \bar{C}_\infty) \,, \tag{179}$$

where

$$\left( \bar{C}_\infty \right)_{ij} := \mathbb{E}_0 \left[ \bar{g}_\infty(\boldsymbol{x}_i) \bar{g}_\infty(\boldsymbol{x}_j) \right]$$
$$= \int_D \varphi(\boldsymbol{\theta}, \boldsymbol{x}_i) \varphi(\boldsymbol{\theta}, \boldsymbol{x}_j) \mu_\infty(d\boldsymbol{\theta}) - \int_D \varphi(\boldsymbol{\theta}, \boldsymbol{x}_i) \mu_\infty(d\boldsymbol{\theta}) \int_D \varphi(\boldsymbol{\theta}', \boldsymbol{x}_j) \mu_\infty(d\boldsymbol{\theta}') \,, \tag{180}$$

by the covariance of $\omega_0$, (27). Thus, we decompose $\bar{C}_\infty$ as $\bar{C}_\infty = \bar{C}_\infty^{(1)} - \bar{C}_\infty^{(2)}$, with

$$\left( \bar{C}_\infty^{(1)} \right)_{ij} = \int_D \varphi(\boldsymbol{\theta}, \boldsymbol{x}_i) \varphi(\boldsymbol{\theta}, \boldsymbol{x}_j) \mu_\infty(d\boldsymbol{\theta}) \,, \tag{181}$$

$$\left(\bar{C}_\infty^{(2)}\right)_{ij} = \int_D \varphi(\boldsymbol{\theta}, \boldsymbol{x}_i)\mu_\infty(d\boldsymbol{\theta}) \int_D \varphi(\boldsymbol{\theta}', \boldsymbol{x}_j)\mu_\infty(d\boldsymbol{\theta}') . \tag{182}$$

Since $\bar{C}_\infty$ is PSD, its square root $\left(\bar{C}_\infty\right)^{1/2}$ is well-defined. By the property of multivariate Gaussian, we can write

$$\bar{g}_\infty \overset{\mathrm{d}}{=} \left(\bar{C}_\infty\right)^{1/2} w , \tag{183}$$

where $\overset{\mathrm{d}}{=}$ denotes equality in distribution, and $w \in \mathbb{R}^n$ follows the distribution

$$w \sim \mathcal{N}(0, \mathrm{Id}_n) . \tag{184}$$

This means that almost surely, $\bar{g}_\infty \in \mathrm{Ran}\left(\left(\bar{C}_\infty\right)^{1/2}\right)$, and which would imply that $\bar{g}_\infty \in \mathrm{Ran}\left(\bar{C}_\infty\right)$. This means that almost surely, we can write

$$\bar{g}_\infty = \bar{C}_\infty^{(1)} w^{(1)} - \bar{C}_\infty^{(2)} w^{(2)} \tag{185}$$

for some pair of $w^{(1)}, w^{(2)} \in \mathbb{R}^n$. Our goal is then to show that both $\bar{C}_\infty^{(1)} w^{(1)}$ and $\bar{C}_\infty^{(2)} w^{(2)}$ belong to $\mathrm{Ran}(\mathcal{B}_\infty^\intercal)$. Here, under Assumption 2.1, since $\varphi(\boldsymbol{\theta}, \boldsymbol{x}) = c\hat{\varphi}(\boldsymbol{z}, \boldsymbol{x})$ when $\boldsymbol{\theta} = [c \quad \boldsymbol{z}]^\intercal$, there is

$$\nabla\varphi(\boldsymbol{\theta}, \boldsymbol{x}) = \begin{bmatrix} \hat{\varphi}(\boldsymbol{z}, \boldsymbol{x}) \\ c\nabla_{\boldsymbol{z}}\hat{\varphi}(\boldsymbol{z}, \boldsymbol{x}) \end{bmatrix} . \tag{186}$$

Therefore, first, we have

$$\begin{aligned}
\left(\bar{C}_\infty^{(1)} w^{(1)}\right)_i &= \int_D \varphi(\boldsymbol{\theta}, \boldsymbol{x}_i)\left(\sum_{j=1}^n \varphi(\boldsymbol{\theta}, \boldsymbol{x}_j)w_j^{(1)}\right)\mu_\infty(d\boldsymbol{\theta}) \\
&= \int_D \nabla\varphi(\boldsymbol{\theta}, \boldsymbol{x}_i)^\intercal \cdot \begin{bmatrix} c(\boldsymbol{\theta})\left(\sum_{j=1}^n \varphi(\boldsymbol{\theta}, \boldsymbol{x}_j)w_j^{(1)}\right) \\ 0 \end{bmatrix}\mu_\infty(d\boldsymbol{\theta}) \\
&= \mathcal{B}_\infty^\intercal \boldsymbol{\xi}^{(1)} ,
\end{aligned} \tag{187}$$

with

$$\boldsymbol{\xi}(\boldsymbol{\theta})^{(1)} = \begin{bmatrix} c(\boldsymbol{\theta})\left(\sum_{j=1}^n \varphi(\boldsymbol{\theta}, \boldsymbol{x}_j)w_j^{(1)}\right) \\ 0 \end{bmatrix} . \tag{188}$$

This means that $\left(\bar{C}_\infty^{(1)} w^{(1)}\right) \in \mathrm{Ran}(\mathcal{B}_\infty^\intercal)$.

Second, there is

$$\begin{aligned}
\left(\bar{C}_\infty^{(2)} w^{(2)}\right)_i &= \left(\int_D \varphi(\boldsymbol{\theta}, \boldsymbol{x}_i)\mu_\infty(d\boldsymbol{\theta})\right)\left(\sum_{j=1}^n w_j^{(2)} \int_D \varphi(\boldsymbol{\theta}', \boldsymbol{x}_j)\mu_\infty(d\boldsymbol{\theta}')\right) \\
&= \int_D \nabla\varphi(\boldsymbol{\theta}, \boldsymbol{x}_i)^\intercal \cdot \begin{bmatrix} c(\boldsymbol{\theta})\left(\sum_{j=1}^n w_j^{(2)} \int_D \varphi(\boldsymbol{\theta}', \boldsymbol{x}_j)\mu_\infty(d\boldsymbol{\theta}')\right) \\ 0 \end{bmatrix}\mu_\infty(d\boldsymbol{\theta}) \\
&= \mathcal{B}_\infty^\intercal \boldsymbol{\xi}^{(2)} ,
\end{aligned} \tag{189}$$

with

$$\boldsymbol{\xi}(\boldsymbol{\theta})^{(2)} = \begin{bmatrix} c(\boldsymbol{\theta})\left(\sum_{j=1}^n w_j^{(2)} \int_D \varphi(\boldsymbol{\theta}', \boldsymbol{x}_j)\mu_\infty(d\boldsymbol{\theta}')\right) \\ 0 \end{bmatrix} \tag{190}$$

This means that $\left(\bar{C}_\infty^{(2)} w^{(2)}\right) \in \mathrm{Ran}(\mathcal{B}_\infty^\intercal)$. Hence the lemma is proved. $\qquad\square$

### E.3 Long-time fluctuations under curvature assumptions

When the limiting measure $\mu_\infty$ does not necessarily interpolate the training data, such as in the regularized case, we have the following condition on $\boldsymbol{T}_t$ which guarantees that (41) holds:

**Lemma E.7** *If*

$$\lim_{T\to\infty}\mathbb{E}_0\int_0^T\int_D\langle\boldsymbol{T}_t(\boldsymbol{\theta}),\nabla\nabla V(\boldsymbol{\Theta}_t(\boldsymbol{\theta}),\mu_t)\boldsymbol{T}_t(\boldsymbol{\theta})\rangle\mu_0(d\boldsymbol{\theta})dt\ge 0\,, \tag{191}$$

*(including when this limit is $+\infty$) then* (41) *holds.*

*Proof of Lemma E.7:* With $\mathfrak{D}_t$ defined in (93), for (41) to hold, it is sufficient to show that

$$\lim_{T\to\infty}\fint_0^T\mathfrak{D}_t dt\le 0\,. \tag{192}$$

Recall from (99) that

$$\fint_0^T\mathfrak{D}_t dt=-\frac{1}{T}\|\boldsymbol{T}_T\|_0^2-2\fint_0^T\langle\boldsymbol{T}_t,\mathcal{A}_t^{(V)}\boldsymbol{T}_t\rangle_0 dt-\fint_0^T\langle\boldsymbol{T}_t,\mathcal{A}_t^{(K)}\boldsymbol{T}_t\rangle_0 dt\,. \tag{193}$$

Since $\boldsymbol{T}_0=0$ and $\mathcal{A}_t^{(K)}$ is PSD, we see that the assumption (191) is sufficient. $\qquad\square$

Note that condition (191) is natural since we know from Proposition 2.6 that $\lim_{t\to\infty}\nabla\nabla V(\boldsymbol{\Theta}_t(\boldsymbol{\theta}),\mu_t)=\nabla\nabla V(\boldsymbol{\Theta}_\infty(\boldsymbol{\theta}),\mu_\infty)$ exists and is positive semidefinite $\mu_0$-almost surely. This lemma allow us to derive the following result:

**Theorem E.8 (Long-time fluctuations under assumptions on the curvature)** *Let* $\Lambda_t(\boldsymbol{\theta})$ *denote the smallest eigenvalue of* $\nabla\nabla V(\boldsymbol{\Theta}_t(\boldsymbol{\theta}),\mu_t)$ *(defined in* (29)*) and assume that for a constant $C$ (to be specified in the proof) such that*

$$-\int_D\min\{\Lambda_t(\boldsymbol{\theta}),0\}\mu_0(d\boldsymbol{\theta})=O(e^{-Ct})\qquad as\ t\to\infty. \tag{194}$$

*Then* (191) *and hence* (41) *hold.*

**Remark E.9** *To intuitively understand* (194)*, note that we know from* (18) *in Proposition 2.6 that $\Lambda_t(\boldsymbol{\theta})\to 0$ $\mu_0$-almost surely as $t\to\infty$. Condition* (194) *can therefore be satisfied by having $\Lambda_t(\boldsymbol{\theta})$ converge to zero sufficiently fast in the regions of $D$ where it is negative, or having the measure of these regions with respect to $\mu_0$ converge to zero sufficiently fast, or both.*

*Proof of Theorem E.8:* Our goal is to verify (191) in order to apply Lemma E.7. We first see that

$$\begin{aligned}
&\mathbb{E}_0\int_D\langle\boldsymbol{T}_t(\boldsymbol{\theta}),\nabla\nabla V(\boldsymbol{\Theta}_t(\boldsymbol{\theta}),\mu_t)\boldsymbol{T}_t(\boldsymbol{\theta})\rangle\mu_0(d\boldsymbol{\theta})\\
\ge&\mathbb{E}_0\int_D\lambda_{\min}(\nabla\nabla V(\boldsymbol{\Theta}_t(\boldsymbol{\theta}),\mu_t))|\boldsymbol{T}_t(\boldsymbol{\theta})|^2\mu_0(d\boldsymbol{\theta})\\
\ge&\mathbb{E}_0\int_D\min\{\lambda_{\min}(\nabla\nabla V(\boldsymbol{\Theta}_t(\boldsymbol{\theta}),\mu_t)),0\}|\boldsymbol{T}_t(\boldsymbol{\theta})|^2\mu_0(d\boldsymbol{\theta})\\
=&\int_D\min\{\lambda_{\min}(\nabla\nabla V(\boldsymbol{\Theta}_t(\boldsymbol{\theta}),\mu_t)),0\}\left(\mathbb{E}_0|\boldsymbol{T}_t(\boldsymbol{\theta})|^2\right)\mu_0(d\boldsymbol{\theta})\\
\ge&\int_D\min\{\lambda_{\min}(\nabla\nabla V(\boldsymbol{\Theta}_t(\boldsymbol{\theta}),\mu_t)),0\}\left(\sup_{\boldsymbol{\theta}\in\mathrm{supp}\,\mu_0}\mathbb{E}_0|\boldsymbol{T}_t(\boldsymbol{\theta})|^2\right)\mu_0(d\boldsymbol{\theta})\\
\ge&\|\boldsymbol{T}_t\|_{\sup}^2\left(\int_D\min\{\lambda_{\min}(\nabla\nabla V(\boldsymbol{\Theta}_t(\boldsymbol{\theta}),\mu_t)),0\}\mu_0(d\boldsymbol{\theta})\right)\,,
\end{aligned} \tag{195}$$

where we define, for $\boldsymbol{\xi}\in\mathcal{V}(D)$,

$$\|\boldsymbol{\xi}\|_{\sup}:=\sup_{\boldsymbol{\theta}\in\mathrm{supp}\,\mu_0}\left(\mathbb{E}_0|\boldsymbol{\xi}(\boldsymbol{\theta})|^2\right)^{1/2}\,, \tag{196}$$

which is a norm on $\mathcal{V}(D)$.

Hence, if we assume that $\left|\int_D\min\{\lambda_{\min}(\nabla\nabla V(\boldsymbol{\theta},\mu_t)),0\}\mu_0(d\boldsymbol{\theta})\right|$ is small asymptotically, then what remains is to upper-bound $\|\boldsymbol{T}_t\|_{\sup}$. Recall from (94) that the dynamics of $\boldsymbol{T}_t$ is governed by

$$\dot{\boldsymbol{T}}_t=-(\mathcal{A}_t^{(K)}+\mathcal{A}_t^{(V)})\boldsymbol{T}_t-\boldsymbol{b}_t, \tag{197}$$

Thus, in the $\|\cdot\|_{\text{sup}}$ norm defined above, we have

$$
\begin{aligned}
\frac{d}{dt}\|\boldsymbol{T}_t\|_{\text{sup}} \leq &\| -(\mathcal{A}_t^{(K)} + \mathcal{A}_t^{(V)})\boldsymbol{T}_t - \boldsymbol{b}_t\|_{\text{sup}} \\
\leq &\|\mathcal{A}_t^{(K)}\boldsymbol{T}_t\|_{\text{sup}} + \|\mathcal{A}_t^{(V)}\boldsymbol{T}_t\|_{\text{sup}} + \|\boldsymbol{b}_t\|_{\text{sup}} .
\end{aligned}
\tag{198}
$$

We then want to bound the growth of $\|\boldsymbol{T}_t\|_{\text{sup}}$ by upper-bounding the RHS. Note that for $\boldsymbol{\xi} \in \mathcal{V}(D)$,

$$
\begin{aligned}
\|\mathcal{A}_t^{(V)}\boldsymbol{\xi}\|_{\text{sup}}^2 &= \sup_{\boldsymbol{\theta}\in D} \mathbb{E}_0 |(\mathcal{A}_t^{(V)}\boldsymbol{\xi})(\boldsymbol{\theta})|^2 \\
&= \sup_{\boldsymbol{\theta}\in D} \mathbb{E}_0 |\nabla\nabla V(\boldsymbol{\Theta}_t(\boldsymbol{\theta}), \mu_t)\boldsymbol{\xi}(\boldsymbol{\theta})|^2 \\
&\leq \sup_{\boldsymbol{\theta}\in D} |\nabla\nabla V(\boldsymbol{\Theta}_t(\boldsymbol{\theta}), \mu_t)|^2 \mathbb{E}_0 |\boldsymbol{\xi}(\boldsymbol{\theta})|^2 \\
&\leq (C_{\nabla\nabla\varphi}C_\varphi + \lambda)^2 \sup_{\boldsymbol{\theta}\in D} \mathbb{E}_0 |\boldsymbol{\xi}(\boldsymbol{\theta})|^2 \\
&= (C_{\nabla\nabla\varphi}C_\varphi + \lambda)^2 \|\boldsymbol{\xi}\|_{\text{sup}}^2 ,
\end{aligned}
\tag{199}
$$

$$
\begin{aligned}
\|\mathcal{A}_t^{(K)}\boldsymbol{\xi}\|_{\text{sup}}^2 &= \sup_{\boldsymbol{\theta}\in D} \mathbb{E}_0 |(\mathcal{A}_t^{(K)}\boldsymbol{\xi})(\boldsymbol{\theta})|^2 \\
&= \sup_{\boldsymbol{\theta}\in D} \mathbb{E}_0 \left| \int_D \nabla'\nabla K(\boldsymbol{\Theta}_t(\boldsymbol{\theta}), \boldsymbol{\Theta}_t(\boldsymbol{\theta}'))\boldsymbol{\xi}(\boldsymbol{\theta}')\mu_0(d\boldsymbol{\theta}') \right|^2 \\
&\leq \sup_{\boldsymbol{\theta}\in D} \mathbb{E}_0 \int_D |\nabla'\nabla K(\boldsymbol{\Theta}_t(\boldsymbol{\theta}), \boldsymbol{\Theta}_t(\boldsymbol{\theta}'))|^2 |\boldsymbol{\xi}(\boldsymbol{\theta}')|^2 \mu_0(d\boldsymbol{\theta}') \\
&\leq \sup_{\boldsymbol{\theta}\in D} (C_{\nabla\varphi})^4 \int_D \mathbb{E}_0 |\boldsymbol{\xi}(\boldsymbol{\theta}')|^2 \mu_0(d\boldsymbol{\theta}') \\
&\leq (C_{\nabla\varphi})^4 \sup_{\boldsymbol{\theta}'\in D} \mathbb{E}_0 |\boldsymbol{\xi}(\boldsymbol{\theta}')|^2 \\
&= (C_{\nabla\varphi})^4 \|\boldsymbol{\xi}\|_{\text{sup}}^2 .
\end{aligned}
\tag{200}
$$

Thus,

$$
\|\mathcal{A}_t^{(K)}\boldsymbol{T}_t\|_{\text{sup}} + \|\mathcal{A}_t^{(V)}\boldsymbol{T}_t\|_{\text{sup}} \leq (C_{\nabla\varphi}^2 + C_{\nabla\nabla\varphi}C_\varphi + \lambda)\|\boldsymbol{T}_t\|_{\text{sup}} .
\tag{201}
$$

To bound $\|\boldsymbol{b}_t\|_{\text{sup}}$, we recall that

$$
\begin{aligned}
\boldsymbol{b}_t(\boldsymbol{\theta}) &= \int_D \nabla K(\boldsymbol{\Theta}_t(\boldsymbol{\theta}), \boldsymbol{\Theta}_t(\boldsymbol{\theta}'))\omega_0(d\boldsymbol{\theta}') \\
&= \int_\Omega \nabla\varphi(\boldsymbol{\Theta}_t(\boldsymbol{\theta}), \boldsymbol{x})\bar{g}_t(\boldsymbol{x})\hat{\nu}(d\boldsymbol{x}) ,
\end{aligned}
\tag{202}
$$

with

$$
\bar{g}_t(\boldsymbol{x}) = \int_D \varphi(\boldsymbol{\Theta}_t(\boldsymbol{\theta}), \boldsymbol{x})\omega_0(d\boldsymbol{\theta}) .
\tag{203}
$$

This implies that $\forall \boldsymbol{\theta} \in \operatorname{supp}\mu_0$,

$$
|\boldsymbol{b}_t(\boldsymbol{\theta})| \leq \frac{1}{n} C_{\nabla\varphi} \sum_{l=1}^n |\bar{g}_t(\boldsymbol{x}_l)|
\tag{204}
$$

and so

$$
\begin{aligned}
\mathbb{E}_0 |\boldsymbol{b}_t(\boldsymbol{\theta})|^2 &\leq C_{\nabla\varphi}^2 \mathbb{E}_0 \left( \frac{1}{n} \sum_{l=1}^n |\bar{g}_t(\boldsymbol{x}_l)| \right)^2 \\
&\leq C_{\nabla\varphi}^2 \mathbb{E}_0 \left( \frac{1}{n} \sum_{l=1}^n |\bar{g}_t(\boldsymbol{x}_l)|^2 \right) \\
&\leq C_{\nabla\varphi}^2 \frac{1}{n} \sum_{l=1}^n \mathbb{E}_0 |\bar{g}_t(\boldsymbol{x}_l)|^2 .
\end{aligned}
\tag{205}
$$

On the other hand, similar to (153), we have

$$
\begin{aligned}
\mathbb{E}_0|\bar{g}_t(\boldsymbol{x})|^2 = \mathbb{E}_0\Big|\int_D \varphi(\boldsymbol{\Theta}_t(\boldsymbol{\theta}), \boldsymbol{x})\omega_0(d\boldsymbol{\theta})\Big|^2 & \\
= \int_D \left(\varphi(\boldsymbol{\Theta}_t(\boldsymbol{\theta}), \boldsymbol{x}) - \int_D \varphi(\boldsymbol{\Theta}_t(\boldsymbol{\theta}'), \boldsymbol{x})\mu_0(d\boldsymbol{\theta}')\right)^2 \mu_0(d\boldsymbol{\theta}) & \\
\leq \int_D \big|\varphi(\boldsymbol{\Theta}_t(\boldsymbol{\theta}), \boldsymbol{x})\big|^2 \mu_0(d\boldsymbol{\theta}) & \\
\leq (C_\varphi)^2 \, , &
\end{aligned}
\tag{206}
$$

Thus, there is $\forall \boldsymbol{\theta} \in \operatorname{supp}\mu_0$,

$$
\mathbb{E}_0|\boldsymbol{b}_t(\boldsymbol{\theta})|^2 \leq (C_{\nabla\varphi})^2(C_\varphi)^2 \tag{207}
$$

and so

$$
\|\boldsymbol{b}_t\|_{\sup} \leq C_{\nabla\varphi}C_\varphi \, . \tag{208}
$$

Therefore, based on (198), we have

$$
\frac{d}{dt}\|\boldsymbol{T}_t\|_{\sup} \leq ((C_{\nabla\varphi})^2 + C_{\nabla\nabla\varphi}C_\varphi + \lambda)\|\boldsymbol{T}_t\|_{\sup} + C_{\nabla\varphi}C_\varphi \, . \tag{209}
$$

Since $\boldsymbol{T}_0 = 0$, we thus have

$$
\begin{aligned}
\|\boldsymbol{T}_t\|_{\sup} \leq & C_{\nabla\varphi}C_\varphi \int_0^t e^{((C_{\nabla\varphi})^2 + C_{\nabla\nabla\varphi}C_\varphi + \lambda)(t-s)} ds \\
= & C_{\nabla\varphi}C_\varphi e^{((C_{\nabla\varphi})^2 + C_{\nabla\nabla\varphi}C_\varphi + \lambda)t} \int_0^t e^{-((C_{\nabla\varphi})^2 + C_{\nabla\nabla\varphi}C_\varphi + \lambda)s} ds \\
\leq & \frac{C_{\nabla\varphi}C_\varphi}{(C_{\nabla\varphi})^2 + C_{\nabla\nabla\varphi}C_\varphi + \lambda} e^{((C_{\nabla\varphi})^2 + C_{\nabla\nabla\varphi}C_\varphi + \lambda)t}
\end{aligned}
\tag{210}
$$

Now, using (195), we see that in order for (191) to hold, it is sufficient to have

$$
\lim_{t\to\infty} e^{((C_{\nabla\varphi})^2 + C_{\nabla\nabla\varphi}C_\varphi + \lambda)t}\left(\int_D \min\left\{\lambda_{\min}(\nabla\nabla V(\boldsymbol{\theta}, \mu_t)), 0\right\}\mu_0(d\boldsymbol{\theta})\right) = 0 \tag{211}
$$

and therefore sufficient to have

$$
-\int_D \min\left\{\lambda_{\min}(\nabla\nabla V(\boldsymbol{\theta}, \mu_t)), 0\right\}\mu_0(d\boldsymbol{\theta}) \sim O\left(e^{-((C_{\nabla\varphi})^2 + C_{\nabla\nabla\varphi}C_\varphi + \lambda)t}\right) \tag{212}
$$

$\square$

To intuitively understand (194), note that we know from (18) in Proposition 2.6 that $\Lambda_t(\boldsymbol{\theta}) \to 0$ $\mu_0$-almost surely as $t \to \infty$. Condition (194) can therefore be satisfied by having $\Lambda_t(\boldsymbol{\theta})$ converge to zero sufficiently fast in the regions of $D$ where it is negative, or having the measure of these regions with respect to $\mu_0$ converge to zero sufficiently fast, or both.

### E.4 Proof of Theorem 3.6 (Regularized case)

Recall from Proposition 3.3 that the dynamics of $g_t$ is governed by

$$
g_t(\boldsymbol{x}) + \int_0^t \int_\Omega \Gamma_{t,s}(\boldsymbol{x}, \boldsymbol{x}')g_s(\boldsymbol{x}')\hat{\nu}(d\boldsymbol{x}')ds = \bar{g}_t(\boldsymbol{x}) \, , \tag{213}
$$

with

$$
\Gamma_{t,s}(\boldsymbol{x}, \boldsymbol{x}') = \int_D \langle\nabla\varphi(\boldsymbol{\Theta}_t(\boldsymbol{\theta}), \boldsymbol{x}), J_{t,s}(\boldsymbol{\theta})\nabla\varphi(\boldsymbol{\Theta}_s(\boldsymbol{\theta}), \boldsymbol{x}')\rangle\mu_0(d\boldsymbol{\theta}) \, , \tag{214}
$$

with $J_{t,s}$ being the Jacobian of the flow $\boldsymbol{\Theta}_t$.

In the ERM setting, $\operatorname{supp}\hat{\nu}$ is singular, and we have $\hat{\nu}(d\boldsymbol{x}) = n^{-1}\sum_{l=1}^n \delta_{\boldsymbol{x}_l}(d\boldsymbol{x})$, where $n$ is the total number of training data points. With the definitions given in Appendix A, we will also continue to consider $g_t$ and $\bar{g}_t$ equivalently as $n$-dimensional vectors,

$$
\begin{pmatrix} g_t(\boldsymbol{x}_1) & \cdots & g_t(\boldsymbol{x}_n) \end{pmatrix}^T \, , \qquad \begin{pmatrix} \bar{g}_t(\boldsymbol{x}_1) & \cdots & \bar{g}_t(\boldsymbol{x}_n) \end{pmatrix}^T \, , \tag{215}
$$

respectively. Thus, $\Gamma_{t,s}$ can also be represented by the $n \times n$ matrix

$$\begin{pmatrix} \Gamma_{t,s}(\boldsymbol{x}_1, \boldsymbol{x}_1) & \cdots & \Gamma_{t,s}(\boldsymbol{x}_1, \boldsymbol{x}_n) \\ \vdots & & \vdots \\ \Gamma_{t,s}(\boldsymbol{x}_n, \boldsymbol{x}_1) & \cdots & \Gamma_{t,s}(\boldsymbol{x}_n, \boldsymbol{x}_n) \end{pmatrix} . \tag{216}$$

Under such an abuse of notations, we can simplify (213) into

$$g_t + \int_0^t \Gamma_{t,s} g_s ds = \bar{g}_t . \tag{217}$$

Thus, the goal is to prove that

$$\lim_{t \to \infty} \sup \fint_0^t \mathbb{E}_0 \|g_t\|_{\hat{\nu}}^2 dt \le \mathbb{E}_0 \|\bar{g}_\infty\|_{\hat{\nu}}^2 . \tag{218}$$

In fact, we will prove that for any realization of the randomness of $\mathbb{P}_0$, there is

$$\lim_{t \to \infty} \sup \fint_0^t \|g_t\|_{\hat{\nu}}^2 dt \le \|\bar{g}_\infty\|_{\hat{\nu}}^2 . \tag{219}$$

As in (38), we also define

$$\Gamma_{t-s}^\infty(\boldsymbol{x}, \boldsymbol{x}') = \int_D \langle \nabla\varphi(\boldsymbol{\theta}, \boldsymbol{x}), e^{-(t-s)\nabla\nabla V_\infty(\boldsymbol{\theta})} \nabla\varphi(\boldsymbol{\theta}, \boldsymbol{x}') \rangle \mu_\infty(d\boldsymbol{\theta}) , \tag{220}$$

where for simplicity, we write $V_t(\cdot)$ for $V(\cdot, \mu_t)$ and $V_\infty(\cdot)$ for $V(\cdot, \mu_\infty)$. Then the heuristic argument outlined in Section 3.2 before Theorem 3.4 amounts to rewriting (217) as

$$g_t + \int_0^t \Gamma_{t-s}^\infty g_s ds = \bar{g}_t + \int_0^t (\Gamma_{t-s}^\infty - \Gamma_{t,s}) g_s ds \tag{221}$$

and then arguing that 1) $\Gamma^\infty$ is a nonnegative convolution-type Volterra kernel, and 2) the second term on the RHS is small. Rigorously, we need to introduce an extra level of complication: for every $t_0 > 0$, we can rewrite (217) into

$$\begin{aligned} g_t &= \bar{g}_t - \int_{t_0}^t \Gamma_{t,s} g_s ds - \int_0^{t_0} \Gamma_{t,s} g_s ds \\ &= \bar{g}_t - \int_{t_0}^t \Gamma_{t-s}^\infty g_s ds + \int_{t_0}^t (\Gamma_{t-s}^\infty - \Gamma_{t,s}) g_s ds - \int_0^{t_0} \Gamma_{t,s} g_s ds . \end{aligned} \tag{222}$$

Then, for any $T > t_0$, by multiplying $g_t$ and integrating from $t_0$ to $T$, we get

$$\begin{aligned} &\int_{t_0}^T \|g_t\|_{\hat{\nu}}^2 dt + \int_{t_0}^T \int_{t_0}^t \langle g_t, \Gamma_{t-s}^\infty g_s \rangle_{\hat{\nu}} ds dt \\ &\le \int_{t_0}^T \langle g_t, \bar{g}_t \rangle_{\hat{\nu}} dt + \int_{t_0}^T \langle g_t, \int_{t_0}^t (\Gamma_{t-s}^\infty - \Gamma_{t,s}) g_s ds \rangle_{\hat{\nu}} dt + \int_{t_0}^T \langle g_t, \int_0^{t_0} \Gamma_{t,s} g_s ds \rangle_{\hat{\nu}} dt . \end{aligned} \tag{223}$$

Then firstly, the second term on the LHS is nonnegative because of the nonnegativity of $\Gamma_t^\infty$ as a convolution-type Volterra kernel, as proven in Appendix E.1.

Hence, we have

$$\begin{aligned} \int_{t_0}^T \|g_t\|_{\hat{\nu}}^2 dt &\le \int_{t_0}^T \langle g_t, \bar{g}_t \rangle_{\hat{\nu}} dt + \int_{t_0}^T \langle g_t, \int_{t_0}^t (\Gamma_{t-s}^\infty - \Gamma_{t,s}) g_s ds \rangle_{\hat{\nu}} dt \\ &+ \int_{t_0}^T \left\langle g_t, \int_0^{t_0} \Gamma_{t,s} g_s ds \right\rangle_{\hat{\nu}} dt . \end{aligned} \tag{224}$$

By Cauchy-Schwartz,

$$\int_{t_0}^T \langle g_t, \bar{g}_t \rangle_{\hat{\nu}} dt \le \left( \int_{t_0}^T \|g_t\|_{\hat{\nu}}^2 dt \right)^{\frac{1}{2}} \left( \int_{t_0}^T \|\bar{g}_t\|_{\hat{\nu}}^2 dt \right)^{\frac{1}{2}} , \tag{225}$$

$$\int_{t_0}^{T} \left\langle g_t, \int_{t_0}^{t} (\Gamma_{t-s}^{\infty} - \Gamma_{t,s}) g_s ds \right\rangle_{\hat{\nu}} dt$$

$$\leq \left( \int_{t_0}^{T} \|g_t\|_{\hat{\nu}}^2 dt \right)^{\frac{1}{2}} \left( \int_{t_0}^{T} \left\| \int_{t_0}^{t} (\Gamma_{t-s}^{\infty} - \Gamma_{t,s}) g_s ds \right\|_{\hat{\nu}}^2 dt \right)^{\frac{1}{2}} \quad (226)$$

$$\leq \left( \int_{t_0}^{T} \|g_t\|_{\hat{\nu}}^2 dt \right) \left( \int_{t_0}^{T} \int_{t_0}^{t} \|\Gamma_{t-s}^{\infty} - \Gamma_{t,s}\|_{\hat{\nu}}^2 ds dt \right)^{\frac{1}{2}},$$

and

$$\int_{t_0}^{T} \left\langle g_t, \int_{0}^{t_0} \Gamma_{t,s} g_s ds \right\rangle_{\hat{\nu}} dt$$

$$\leq \left( \int_{t_0}^{T} \|g_t\|_{\hat{\nu}}^2 dt \right)^{\frac{1}{2}} \left( \int_{t_0}^{T} \left\| \int_{0}^{t_0} \Gamma_{t,s} g_s ds \right\|_{\hat{\nu}}^2 dt \right)^{\frac{1}{2}}$$

$$\leq \left( \int_{t_0}^{T} \|g_t\|_{\hat{\nu}}^2 dt \right)^{\frac{1}{2}} \left( \int_{t_0}^{T} \left( \int_{0}^{t_0} \|\Gamma_{t,s}\|_{\hat{\nu}}^2 ds \right) \left( \int_{0}^{t_0} \|g_s\|_{\hat{\nu}}^2 ds \right) dt \right)^{\frac{1}{2}} \quad (227)$$

$$\leq \left( \int_{t_0}^{T} \|g_t\|_{\hat{\nu}}^2 dt \right)^{\frac{1}{2}} \left( \int_{0}^{t_0} \|g_t\|_{\hat{\nu}}^2 dt \right)^{\frac{1}{2}} \left( \int_{t_0}^{T} \int_{0}^{t_0} \|\Gamma_{t,s}\|_{\hat{\nu}}^2 ds dt \right)^{\frac{1}{2}}.$$

Therefore, putting everything together, we have

$$\left( \int_{t_0}^{T} \|g_t\|_{\hat{\nu}}^2 dt \right)^{\frac{1}{2}} \leq \left( \int_{t_0}^{T} \|\bar{g}_t\|_{\hat{\nu}}^2 dt \right)^{\frac{1}{2}}$$

$$+ \left( \int_{t_0}^{T} \|g_t\|_{\hat{\nu}}^2 dt \right)^{\frac{1}{2}} \left( \int_{t_0}^{T} \int_{t_0}^{t} \|\Gamma_{t-s}^{\infty} - \Gamma_{t,s}\|_{\hat{\nu}}^2 ds dt \right)^{\frac{1}{2}} \quad (228)$$

$$+ \left( \int_{0}^{t_0} \|g_t\|_{\hat{\nu}}^2 dt \right)^{\frac{1}{2}} \left( \int_{t_0}^{T} \int_{0}^{t_0} \|\Gamma_{t,s}\|_{\hat{\nu}}^2 ds dt \right)^{\frac{1}{2}},$$

and hence, using $\fint_a^b \cdot dt$ to denote the averaged integral $\frac{1}{b-a} \int_a^b \cdot dt$,

$$\left( \fint_{t_0}^{T} \|g_t\|_{\hat{\nu}}^2 dt \right)^{\frac{1}{2}} \leq \left( \fint_{t_0}^{T} \|\bar{g}_t\|_{\hat{\nu}}^2 dt \right)^{\frac{1}{2}}$$

$$+ \left( \fint_{t_0}^{T} \|g_t\|_{\hat{\nu}}^2 dt \right)^{\frac{1}{2}} \left( \int_{t_0}^{T} \int_{t_0}^{t} \|\Gamma_{t-s}^{\infty} - \Gamma_{t,s}\|_{\hat{\nu}}^2 ds dt \right)^{\frac{1}{2}} \quad (229)$$

$$+ \left( \int_{0}^{t_0} \|g_t\|_{\hat{\nu}}^2 dt \right)^{\frac{1}{2}} \left( \fint_{t_0}^{T} \int_{0}^{t_0} \|\Gamma_{t,s}\|_{\hat{\nu}}^2 ds dt \right)^{\frac{1}{2}},$$

or

$$\left( 1 - \left[ \int_{t_0}^{T} \int_{t_0}^{t} \|\Gamma_{t-s}^{\infty} - \Gamma_{t,s}\|_{\hat{\nu}}^2 ds dt \right]^{\frac{1}{2}} \right) \left( \fint_{t_0}^{T} \|g_t\|_{\hat{\nu}}^2 dt \right)^{\frac{1}{2}}$$

$$\leq \left( \fint_{t_0}^{T} \|\bar{g}_t\|_{\hat{\nu}}^2 dt \right)^{\frac{1}{2}} + \left( \int_{0}^{t_0} \|g_t\|_{\hat{\nu}}^2 dt \right)^{\frac{1}{2}} \left( \fint_{t_0}^{T} \int_{0}^{t_0} \|\Gamma_{t,s}\|_{\hat{\nu}}^2 ds dt \right)^{\frac{1}{2}}. \quad (230)$$

**Lemma E.10** *Under all assumptions in Theorem 3.6 except for (46) being replaced by a weaker condition,*

$$\int_0^\infty \int_D \left( |\boldsymbol{\Theta}_t(\boldsymbol{\theta}) - \boldsymbol{\Theta}_\infty(\boldsymbol{\theta})| + |U_t(\boldsymbol{\theta})|^2 \right) e^{C_1(U_t(\boldsymbol{\theta}) + \bar{U}_t)} \mu_0(d\boldsymbol{\theta})dt < \infty , \tag{231}$$

*we have*

$$\lim_{t_0 \to \infty} \int_{t_0}^\infty \int_{t_0}^t \|\Gamma_{t-s}^\infty - \Gamma_{t,s}\|_{\hat{\nu}}^2 ds dt = 0 \tag{232}$$

*and $\forall t_0 > 0$,*

$$\lim_{T \to \infty} \fint_{t_0}^T \int_0^{t_0} \|\Gamma_{t,s}\|^2 ds dt = 0 . \tag{233}$$

*We will prove in Appendix E.4.2 that (46) indeed implies (231).*

The lemma will be proved in Appendix E.4.1, and let us first proceed with the proof of the theorem assuming this lemma. Suppose for contradiction that (219) does not hold, meaning that

$$\lim_{T \to \infty} \sup \left( \fint_0^T \|g_t\|_{\hat{\nu}}^2 dt \right)^{\frac{1}{2}} = \|\bar{g}_\infty\|_{\hat{\nu}} + \epsilon \tag{234}$$

for some $\epsilon > 0$. We will select a pair of $t_0$ and $T$ for which the inequality (230) cannot be satisfied. Firstly, by the convergence of $\bar{g}_t$ to $\bar{g}_\infty$, $\exists t_a > 0$ such that $\forall t_1, t_2 > t_a$,

$$\left( \fint_{t_1}^{t_2} \|\bar{g}_t\|_{\hat{\nu}}^2 dt \right)^{\frac{1}{2}} \leq \|\bar{g}_\infty\|_{\hat{\nu}} + \tfrac{1}{6}\epsilon . \tag{235}$$

Secondly, by our assumption (234) and the first part of Lemma E.10, $\exists t_0 > t_a$ such that both

$$\left( \fint_0^{t_0} \|g_t\|_{\hat{\nu}}^2 dt \right)^{\frac{1}{2}} \leq \|\bar{g}_\infty\|_{\hat{\nu}} + 2\epsilon \tag{236}$$

and

$$\int_{t_0}^\infty \int_{t_0}^t \|\Gamma_{t-s}^\infty - \Gamma_{t,s}\|_{\hat{\nu}}^2 ds dt < \frac{\epsilon}{6\|\bar{g}_\infty\|_{\hat{\nu}} + 3\epsilon} \tag{237}$$

are satisfied. In particular, (236) implies

$$\left( \int_0^{t_0} |g_t|^2 dt \right)^{\frac{1}{2}} \leq t_0^{\frac{1}{2}} \cdot (|\bar{g}_\infty| + 2\epsilon) \tag{238}$$

Let

$$\delta = \left( \frac{\epsilon}{6 t_0^{\frac{1}{2}} \cdot (\|\bar{g}_\infty\|_{\hat{\nu}} + 2\epsilon)} \right)^2 > 0 . \tag{239}$$

By the second part of Lemma E.10, $\exists t_b > t_0$ such that $\forall T > t_b$,

$$\fint_{t_0}^T \int_0^{t_0} \|\Gamma_{t,s}\|^2 ds dt < \delta \tag{240}$$

so that the last term in (230) satisfies

$$\left( \int_0^{t_0} \|g_t\|_{\hat{\nu}}^2 dt \right)^{\frac{1}{2}} \left( \fint_{t_0}^T \int_0^{t_0} \|\Gamma_{t,s}\|^2 ds dt \right)^{\frac{1}{2}} < \tfrac{1}{6}\epsilon \tag{241}$$

By our assumption (234), we can choose a $T > t_b$ such that

$$\left( \fint_0^T \|g_t\|_{\hat{\nu}}^2 dt \right)^{\frac{1}{2}} \geq \|\bar{g}_\infty\|_{\hat{\nu}} + \tfrac{2}{3}\epsilon . \tag{242}$$

Since

$$\left( \fint_0^{t_0} \|g_t\|_{\hat{\nu}}^2 dt \right)^{\frac{1}{2}} \leq \|\bar{g}_\infty\|_{\hat{\nu}} + 2\epsilon, \tag{243}$$

we can assume without loss of generality that $\frac{T}{t_0}$ is large enough so that

$$\Big(\fint_{t_0}^T \|g_t\|_{\hat{\nu}}^2 dt\Big)^{\frac{1}{2}} \geq \|\bar{g}_\infty\|_{\hat{\nu}} + \tfrac{1}{2}\epsilon . \tag{244}$$

Thus, back to the inequality (230), the LHS is strictly lower-bounded by

$$\big(\|\bar{g}_\infty\|_{\hat{\nu}} + \tfrac{1}{2}\epsilon\big)\big(1 - \frac{\epsilon}{6\|\bar{g}_\infty\|_{\hat{\nu}} + 3\epsilon}\big) = \|\bar{g}_\infty\|_{\hat{\nu}} + \tfrac{1}{3}\epsilon, \tag{245}$$

whereas the RHS is strictly upper-bounded by

$$\|\bar{g}_\infty\|_{\hat{\nu}} + \tfrac{1}{6}\epsilon + \tfrac{1}{6}\epsilon = \|\bar{g}_\infty\|_{\hat{\nu}} + \tfrac{1}{3}\epsilon . \tag{246}$$

This gives a contradiction, and hence we have proved Theorem 3.6. □

### E.4.1  Proof of Lemma E.10

It remains to prove Lemma E.10. To do so we will need an auxiliary result, that we state and prove first:

**Lemma E.11** *Let $\Delta\Gamma_{t,s} := \Gamma_{t,s} - \Gamma_{t-s}^\infty$. If $\nabla\nabla V$ is uniformly positive definite with eigenvalues lower-bounded by $\lambda$, then there exists constants $C$ and $C'$ whose values depend on $|D'|$, $C_\varphi$, $C_{\nabla\varphi}$, $C_{\nabla\nabla\varphi}$, and $L_{\nabla\nabla\varphi}$ such that*

$$\|\Delta\Gamma_{t,s}\|_{\hat{\nu}} \leq Ce^{-\lambda(t-s)} \int_D \Big(|\Delta\Theta_t(\boldsymbol{\theta})| + (|\Delta\Theta_s(\boldsymbol{\theta})| + U_s(\boldsymbol{\theta}))\, e^{C'(U_s(\boldsymbol{\theta}) + \bar{U}_s)}\Big)\mu_0(d\boldsymbol{\theta}) \tag{247}$$

*where $\Delta\Theta_t(\boldsymbol{\theta}) = \Theta_t(\boldsymbol{\theta}) - \Theta_\infty(\boldsymbol{\theta})$.*

*Proof of Lemma E.11:* To bound $\|\Delta\Gamma_{t,s}\|_{\hat{\nu}}$, we bound $\|\Delta\Gamma_{t,s}\eta\|_{\hat{\nu}}$ for $\eta \in \mathbb{R}^n$. Note that $\Delta\Gamma_{t,s}\eta$ can be obtained in the following way. Consider the two systems

$$\begin{cases} \dfrac{d}{dt}\boldsymbol{\xi}_t(\boldsymbol{\theta}) = -\nabla\nabla V_t(\Theta_t(\boldsymbol{\theta}))\boldsymbol{\xi}_t(\boldsymbol{\theta}) \\[2mm] \boldsymbol{\xi}_s(\boldsymbol{\theta}) = \displaystyle\int_\Omega \nabla\varphi(\Theta_s(\boldsymbol{\theta}), \boldsymbol{x}')\eta(\boldsymbol{x}')\hat{\nu}(d\boldsymbol{x}') \end{cases} \tag{248}$$

$$\begin{cases} \dfrac{d}{dt}\boldsymbol{\xi}_t'(\boldsymbol{\theta}) = -\nabla\nabla V_\infty(\Theta_\infty(\boldsymbol{\theta}))\boldsymbol{\xi}_t'(\boldsymbol{\theta}) \\[2mm] \boldsymbol{\xi}_s'(\boldsymbol{\theta}) = \displaystyle\int_\Omega \nabla\varphi(\Theta_\infty(\boldsymbol{\theta}), \boldsymbol{x}')\eta(\boldsymbol{x}')\hat{\nu}(d\boldsymbol{x}') \end{cases} \tag{249}$$

Then there is

$$\begin{aligned} (\Gamma_{t,s}\eta)(\boldsymbol{x}) &= \int_D \nabla\varphi(\Theta_t(\boldsymbol{\theta}), \boldsymbol{x}) \cdot \boldsymbol{\xi}_t(\boldsymbol{\theta})\mu_0(d\boldsymbol{\theta}) \\ (\Gamma_{t-s}^\infty\eta)(\boldsymbol{x}) &= \int_D \nabla\varphi(\Theta_\infty(\boldsymbol{\theta}), \boldsymbol{x}) \cdot \boldsymbol{\xi}_t'(\boldsymbol{\theta})\mu_0(d\boldsymbol{\theta}) \end{aligned} \tag{250}$$

and hence

$$\begin{aligned} (\Delta\Gamma_{t,s}\eta)(\boldsymbol{x}) &= \int_D \nabla\varphi(\Theta_t(\boldsymbol{\theta}), \boldsymbol{x})\boldsymbol{\xi}_t(\boldsymbol{\theta})\mu_0(d\boldsymbol{\theta}) - \int_D \nabla\varphi(\Theta_\infty(\boldsymbol{\theta}), \boldsymbol{x})\boldsymbol{\xi}_t'(\boldsymbol{\theta})\mu_0(d\boldsymbol{\theta}) \\ &= \int_D \nabla\varphi(\Theta_t(\boldsymbol{\theta}), \boldsymbol{x}) \cdot \big(\boldsymbol{\xi}_t(\boldsymbol{\theta}) - \boldsymbol{\xi}_t'(\boldsymbol{\theta})\big)\mu_0(d\boldsymbol{\theta}) \\ &\quad + \int_D \big(\nabla\varphi(\Theta_t(\boldsymbol{\theta}), \boldsymbol{x}) - \nabla\varphi(\Theta_\infty(\boldsymbol{\theta}), \boldsymbol{x})\big) \cdot \boldsymbol{\xi}_t'(\boldsymbol{\theta})\mu_0(d\boldsymbol{\theta}) . \end{aligned} \tag{251}$$

We will first try to bound $\boldsymbol{\xi}_t(\boldsymbol{\theta}) - \boldsymbol{\xi}_t'(\boldsymbol{\theta})$ as a function of $\eta$. Define $\Delta\boldsymbol{\xi}_t(\boldsymbol{\theta}) = \boldsymbol{\xi}_t(\boldsymbol{\theta}) - \boldsymbol{\xi}_t'(\boldsymbol{\theta})$. Then

$$\begin{aligned} \frac{d}{dr}\Delta\boldsymbol{\xi}_r(\boldsymbol{\theta}) =\ & -\big(\nabla\nabla V_r(\Theta_r(\boldsymbol{\theta})) - \nabla\nabla V_\infty(\Theta_\infty(\boldsymbol{\theta}))\big)\boldsymbol{\xi}_r - \nabla\nabla V_\infty(\Theta_\infty(\boldsymbol{\theta}))\Delta\boldsymbol{\xi}_r(\boldsymbol{\theta}) \\ =\ & -\nabla\nabla V_\infty(\Theta_\infty(\boldsymbol{\theta}))\Delta\boldsymbol{\xi}_r(\boldsymbol{\theta}) \\ & -\big(\nabla\nabla V_r(\Theta_r(\boldsymbol{\theta})) - \nabla\nabla V_\infty(\Theta_\infty(\boldsymbol{\theta}))\big)\boldsymbol{\xi}_r' \\ & -\big(\nabla\nabla V_r(\Theta_r(\boldsymbol{\theta})) - \nabla\nabla V_\infty(\Theta_\infty(\boldsymbol{\theta}))\big)\Delta\boldsymbol{\xi}_r . \end{aligned} \tag{252}$$

Thus,

$$\Delta\boldsymbol{\xi}_t(\boldsymbol{\theta}) = e^{-(t-s)\nabla\nabla V_\infty(\boldsymbol{\Theta}_\infty(\boldsymbol{\theta}))}\Delta\boldsymbol{\xi}_s(\boldsymbol{\theta})$$
$$+ \int_s^t e^{-(t-r)\nabla\nabla V_\infty(\boldsymbol{\Theta}_\infty(\boldsymbol{\theta}))}\big(\nabla\nabla V_r(\boldsymbol{\Theta}_r(\boldsymbol{\theta})) - \nabla\nabla V_\infty(\boldsymbol{\Theta}_\infty(\boldsymbol{\theta}))\big)\boldsymbol{\xi}_r'(\boldsymbol{\theta})dr$$
$$+ \int_s^t e^{-(t-r)\nabla\nabla V_\infty(\boldsymbol{\Theta}_\infty(\boldsymbol{\theta}))}\big(\nabla\nabla V_r(\boldsymbol{\Theta}_r(\boldsymbol{\theta})) - \nabla\nabla V_\infty(\boldsymbol{\Theta}_\infty(\boldsymbol{\theta}))\big)\Delta\boldsymbol{\xi}_r(\boldsymbol{\theta})dr \,. \tag{253}$$

Since $\nabla\nabla V_\infty(\boldsymbol{\Theta}_\infty(\boldsymbol{\theta})) - \lambda I_d$ is positive semidefinite, we first have

$$|\boldsymbol{\xi}_r'(\boldsymbol{\theta})| \le e^{-\lambda(r-s)}|\boldsymbol{\xi}_s'(\boldsymbol{\theta})| \tag{254}$$

as well as

$$|\Delta\boldsymbol{\xi}_t(\boldsymbol{\theta})| \le e^{-\lambda(t-s)}|\Delta\boldsymbol{\xi}_s(\boldsymbol{\theta})|$$
$$+ \int_s^t e^{-\lambda(t-r)}\|\nabla\nabla V_r(\boldsymbol{\Theta}_r(\boldsymbol{\theta})) - \nabla\nabla V_\infty(\boldsymbol{\Theta}_\infty(\boldsymbol{\theta}))\|\,|\boldsymbol{\xi}_r'(\boldsymbol{\theta})|dr$$
$$+ \int_s^t e^{-\lambda(t-r)}\|\nabla\nabla V_r(\boldsymbol{\Theta}_r(\boldsymbol{\theta})) - \nabla\nabla V_\infty(\boldsymbol{\Theta}_\infty(\boldsymbol{\theta}))\|\,|\Delta\boldsymbol{\xi}_r(\boldsymbol{\theta})|dr$$
$$\le e^{-\lambda(t-s)}|\Delta\boldsymbol{\xi}_s(\boldsymbol{\theta})|$$
$$+ \int_s^t e^{-\lambda(t-s)}\|\nabla\nabla V_r(\boldsymbol{\Theta}_r(\boldsymbol{\theta})) - \nabla\nabla V_\infty(\boldsymbol{\Theta}_\infty(\boldsymbol{\theta}))\|\,|\boldsymbol{\xi}_s'(\boldsymbol{\theta})|dr$$
$$+ \int_s^t e^{-\lambda(t-r)}\|\nabla\nabla V_r(\boldsymbol{\Theta}_r(\boldsymbol{\theta})) - \nabla\nabla V_\infty(\boldsymbol{\Theta}_\infty(\boldsymbol{\theta}))\|\,|\Delta\boldsymbol{\xi}_r(\boldsymbol{\theta})|dr \,. \tag{255}$$

To prepare for an application of Gronwall's inequality, we introduce a change-of-variable by defining, for $r \in [s,t]$,

$$\overline{\Delta\boldsymbol{\xi}}_r(\boldsymbol{\theta}) = e^{\lambda(t-s)}\Delta\boldsymbol{\xi}_r(\boldsymbol{\theta}) \,. \tag{256}$$

Then we can rewrite the equation above as

$$|\overline{\Delta\boldsymbol{\xi}}_t(\boldsymbol{\theta})| = e^{\lambda(r-s)}|\Delta\boldsymbol{\xi}_t(\boldsymbol{\theta})|$$
$$\le |\Delta\boldsymbol{\xi}_s(\boldsymbol{\theta})| + \int_s^t \|\nabla\nabla V_r(\boldsymbol{\Theta}_r(\boldsymbol{\theta})) - \nabla\nabla V_\infty(\boldsymbol{\Theta}_\infty(\boldsymbol{\theta}))\|\,|\boldsymbol{\xi}_s'(\boldsymbol{\theta})|dr$$
$$+ \int_s^t e^{\lambda(r-s)}\|\nabla\nabla V_r(\boldsymbol{\Theta}_r(\boldsymbol{\theta})) - \nabla\nabla V_\infty(\boldsymbol{\Theta}_\infty(\boldsymbol{\theta}))\|\,|\Delta\boldsymbol{\xi}_r(\boldsymbol{\theta})|dr$$
$$\le |\overline{\Delta\boldsymbol{\xi}}_s(\boldsymbol{\theta})| + \int_s^t \|\nabla\nabla V_r(\boldsymbol{\Theta}_r(\boldsymbol{\theta})) - \nabla\nabla V_\infty(\boldsymbol{\Theta}_\infty(\boldsymbol{\theta}))\|\,|\boldsymbol{\xi}_s'(\boldsymbol{\theta})|dr$$
$$+ \int_s^t \|\nabla\nabla V_r(\boldsymbol{\Theta}_r(\boldsymbol{\theta})) - \nabla\nabla V_\infty(\boldsymbol{\Theta}_\infty(\boldsymbol{\theta}))\|\,|\overline{\Delta\boldsymbol{\xi}}_r(\boldsymbol{\theta})|dr \,. \tag{257}$$

Thus, by Gronwall's inequality,

$$|\overline{\Delta\boldsymbol{\xi}}_t(\boldsymbol{\theta})| \le \left(|\overline{\Delta\boldsymbol{\xi}}_s(\boldsymbol{\theta})| + \int_s^t \|\nabla\nabla V_r(\boldsymbol{\Theta}_r(\boldsymbol{\theta})) - \nabla\nabla V_\infty(\boldsymbol{\Theta}_\infty(\boldsymbol{\theta}))\|\,|\boldsymbol{\xi}_s'(\boldsymbol{\theta})|dr\right)$$
$$\times e^{\int_s^t \|\nabla\nabla V_r(\boldsymbol{\Theta}_r(\boldsymbol{\theta})) - \nabla\nabla V_\infty(\boldsymbol{\Theta}_\infty(\boldsymbol{\theta}))\|dr} \,, \tag{258}$$

or, back in the original variable that we are interested in,

$$|\Delta\boldsymbol{\xi}_t(\boldsymbol{\theta})| \le \left(|\Delta\boldsymbol{\xi}_s(\boldsymbol{\theta})| + \int_s^t \|\nabla\nabla V_r(\boldsymbol{\Theta}_r(\boldsymbol{\theta})) - \nabla\nabla V_\infty(\boldsymbol{\Theta}_\infty(\boldsymbol{\theta}))\|\,|\boldsymbol{\xi}_s'(\boldsymbol{\theta})|dr\right)$$
$$\times e^{-\lambda(t-s)+\int_s^t \|\nabla\nabla V_r(\boldsymbol{\Theta}_r(\boldsymbol{\theta})) - \nabla\nabla V_\infty(\boldsymbol{\Theta}_\infty(\boldsymbol{\theta}))\|dr} \,. \tag{259}$$

Now, we have

$$|\Delta\Gamma_{t,s}\eta(\boldsymbol{x})|$$

$$\leq \|\int_D \nabla\varphi(\boldsymbol{\Theta}_t(\boldsymbol{\theta}),\boldsymbol{x})^\mathsf{T}\cdot\Delta\boldsymbol{\xi}_t(\boldsymbol{\theta})\mu_0(d\boldsymbol{\theta})\|_{\hat\nu}$$

$$+ \|\int_D \left(\nabla\varphi(\boldsymbol{\Theta}_t(\boldsymbol{\theta}),\boldsymbol{x}) - \nabla\varphi(\boldsymbol{\Theta}_\infty(\boldsymbol{\theta}),\boldsymbol{x})\right)^\mathsf{T}\boldsymbol{\xi}_t'(\boldsymbol{\theta})\mu_0(d\boldsymbol{\theta})\|_{\hat\nu}$$

$$\leq C_{\nabla\varphi}\int_D |\Delta\boldsymbol{\xi}_t(\boldsymbol{\theta})|\mu_0(d\boldsymbol{\theta}) + C_{\nabla\nabla\varphi}\int_D |\Delta\boldsymbol{\Theta}_t(\boldsymbol{\theta})||\boldsymbol{\xi}_t'(\boldsymbol{\theta})|\mu_0(d\boldsymbol{\theta}) \quad (260)$$

$$\leq C_{\nabla\varphi}e^{-\lambda(t-s)}\int_D \left(|\Delta\boldsymbol{\xi}_s(\boldsymbol{\theta})| + \int_s^t \|\nabla\nabla V_r(\boldsymbol{\Theta}_r(\boldsymbol{\theta})) - \nabla\nabla V_\infty(\boldsymbol{\Theta}_\infty(\boldsymbol{\theta}))\||\boldsymbol{\xi}_s'(\boldsymbol{\theta})|dr\right)$$

$$e^{\int_s^t \|\nabla\nabla V_r(\boldsymbol{\Theta}_r(\boldsymbol{\theta})) - \nabla\nabla V_\infty(\boldsymbol{\Theta}_\infty(\boldsymbol{\theta}))\|dr}\mu_0(d\boldsymbol{\theta})$$

$$+ C_{\nabla\nabla\varphi}e^{-\lambda(t-s)}\int_D |\Delta\boldsymbol{\Theta}_t(\boldsymbol{\theta})||\boldsymbol{\xi}_s'(\boldsymbol{\theta})|\mu_0(d\boldsymbol{\theta}) .$$

Note that we have,

$$|\boldsymbol{\xi}_s'(\boldsymbol{\theta})| = |\int_\Omega \nabla\varphi(\boldsymbol{\Theta}_\infty(\boldsymbol{\theta}),\boldsymbol{x}')\eta(\boldsymbol{x}')\hat\nu(d\boldsymbol{x}')| \leq C_{\nabla\varphi}\sup_{1\leq l\leq n}|\eta(\boldsymbol{x}_l)| \leq n^{\frac{1}{2}}C_{\nabla\varphi}\|\eta\|_{\hat\nu}, \quad (261)$$

$$|\Delta\boldsymbol{\xi}_s(\boldsymbol{\theta})| = |\int_\Omega \left(\nabla\varphi(\boldsymbol{\Theta}_s(\boldsymbol{\theta}),\boldsymbol{x}) - \nabla\varphi(\boldsymbol{\Theta}_\infty(\boldsymbol{\theta}),\boldsymbol{x})\right)\eta(\boldsymbol{x}')\hat\nu(d\boldsymbol{x}')|$$

$$\leq \int_\Omega |\nabla\varphi(\boldsymbol{\Theta}_s(\boldsymbol{\theta}),\boldsymbol{x}') - \nabla\varphi(\boldsymbol{\Theta}_\infty(\boldsymbol{\theta}),\boldsymbol{x}')||\eta(\boldsymbol{x}')|\hat\nu(d\boldsymbol{x}') \quad (262)$$

$$\leq n^{\frac{1}{2}}C_{\nabla\nabla\varphi}|\Delta\boldsymbol{\Theta}_s(\boldsymbol{\theta})|\|\eta\|_{\hat\nu}$$

and, since $\nabla\nabla V_r(\boldsymbol{\theta}) = \int_\Omega \nabla\nabla\varphi(\boldsymbol{\theta},\boldsymbol{x})(f_r(\boldsymbol{x}) - f_*(\boldsymbol{x}))\hat\nu(d\boldsymbol{x})$ and $\nabla\nabla V_\infty(\boldsymbol{\theta}) = \int_\Omega \nabla\nabla\varphi(\boldsymbol{\theta},\boldsymbol{x})(f_\infty(\boldsymbol{x}) - f_*(\boldsymbol{x}))\hat\nu(d\boldsymbol{x})$,

$$\|\nabla\nabla V_r(\boldsymbol{\Theta}_r(\boldsymbol{\theta})) - \nabla\nabla V_\infty(\boldsymbol{\Theta}_\infty(\boldsymbol{\theta}))\|$$
$$\leq \|\nabla\nabla V_r(\boldsymbol{\Theta}_r(\boldsymbol{\theta})) - \nabla\nabla V_r(\boldsymbol{\Theta}_\infty(\boldsymbol{\theta}))\|$$
$$+ \|\nabla\nabla V_r(\boldsymbol{\Theta}_\infty(\boldsymbol{\theta})) - \nabla\nabla V_\infty(\boldsymbol{\Theta}_\infty(\boldsymbol{\theta}))\| \quad (263)$$
$$\leq L_{\nabla\nabla\varphi}C_\varphi|\Delta\boldsymbol{\Theta}_r(\boldsymbol{\theta})| + C_{\nabla\nabla\varphi}\|\Delta f_r\|_{\hat\nu,\infty},$$

where we use $\|f\|_{\hat\nu,\infty}$ to denote $\sup_{\boldsymbol{x}\in\text{supp}\,\hat\nu}|f(\boldsymbol{x})|$ and we defined $\Delta f_t = f_t - f_\infty$.

As a result, we have

$$\|\Delta\Gamma_{t,s}\|_{\hat\nu}$$
$$\leq \|\eta\|_{\hat\nu}^{-1}\|\Delta\Gamma_{t,s}\eta\|_{\hat\nu}$$
$$\leq C_{\nabla\varphi}e^{-\lambda(t-s)}\int_D \left(C_{\nabla\nabla\varphi}|\Delta\boldsymbol{\Theta}_s(\boldsymbol{\theta})| + \int_s^t \left(L_{\nabla\nabla\varphi}C_\varphi|\Delta\boldsymbol{\Theta}_r(\boldsymbol{\theta})| + C_{\nabla\nabla\varphi}\|\Delta f_r\|_{\hat\nu,\infty}\right)C_{\nabla\varphi}dr\right)$$
$$\times e^{\int_s^t L_{\nabla\nabla\varphi}C_\varphi|\Delta\boldsymbol{\Theta}_r(\boldsymbol{\theta})|+C_{\nabla\nabla\varphi}\|\Delta f_r\|_{\hat\nu,\infty}dr}\mu_0(d\boldsymbol{\theta})$$
$$+ C_{\nabla\nabla\varphi}e^{-\lambda(t-s)}\int_D C_{\nabla\varphi}|\Delta\boldsymbol{\Theta}_t(\boldsymbol{\theta})|\mu_0(d\boldsymbol{\theta}) .$$
$$(264)$$

Therefore, using $C_0$, $C_1$, etc. to represent constants that depend on $C_\varphi$, $C_{\nabla\varphi}$, $C_{\nabla\nabla\varphi}$, $C_{\nabla\nabla\varphi}$ and $L_{\nabla\nabla\varphi}$, we have

$$\|\Delta\Gamma_{t,s}\|_{\hat\nu}$$
$$\leq C_0 e^{-\lambda(t-s)}\Big(\int_D |\Delta\boldsymbol{\Theta}_t(\boldsymbol{\theta})|\mu_0(d\boldsymbol{\theta}) + \int_D |\Delta\boldsymbol{\Theta}_s(\boldsymbol{\theta})|e^{C_1\int_s^t |\Delta\boldsymbol{\Theta}_r(\boldsymbol{\theta})|+\|\Delta f_r\|_{\hat\nu,\infty}dr}\mu_0(d\boldsymbol{\theta})$$
$$+ \int_D \big(\int_s^t |\Delta\boldsymbol{\Theta}_r(\boldsymbol{\theta})| + \|f_r - f_\infty\|_{\hat\nu,\infty}dr\big)e^{C_1\int_s^t |\Delta\boldsymbol{\Theta}_r(\boldsymbol{\theta})|+\|\Delta f_r\|_{\hat\nu,\infty}dr}\mu_0(d\boldsymbol{\theta})\Big) .$$
$$(265)$$

Note that $\|\Delta f_r\|_{\hat{\nu},\infty}$ can be further upper-bounded by $C_\varphi \int_D |\Delta\Theta_r(\boldsymbol{\theta})|\mu_0(d\boldsymbol{\theta})$. Furthermore, defining

$$\overline{\Delta\Theta_t} = \int_D |\Delta\Theta_t(\boldsymbol{\theta})|\mu_0(d\boldsymbol{\theta}) \tag{266}$$

we can write the bound above as

$$\|\Delta\Gamma_{t,s}\|_{\hat{\nu}} \leq C_0 e^{-\lambda(t-s)}\Big( \int_D |\Delta\Theta_t(\boldsymbol{\theta})|\mu_0(d\boldsymbol{\theta}) + \int_D |\Delta\Theta_s(\boldsymbol{\theta})|e^{C_1 \int_s^t |\Delta\Theta_r(\boldsymbol{\theta})|+\overline{\Delta\Theta_r}dr}\mu_0(d\boldsymbol{\theta})$$
$$+ \int_D \big(\int_s^t |\Delta\Theta_r(\boldsymbol{\theta})| + \overline{\Delta\Theta_r}dr\big)e^{C_1 \int_s^t |\Delta\Theta_r(\boldsymbol{\theta})|+\overline{\Delta\Theta_r}dr}\mu_0(d\boldsymbol{\theta})\Big) . \tag{267}$$

Finally, let

$$U_t(\boldsymbol{\theta}) = \int_t^\infty |\Delta\Theta_t(\boldsymbol{\theta})|dt \tag{268}$$

and

$$\bar{U}_t = \int_D U_t(\boldsymbol{\theta})\mu_0(d\boldsymbol{\theta}) = \int_t^\infty \overline{\Delta\Theta_t}dt . \tag{269}$$

Then there is

$$\|\Delta\Gamma_{t,s}\|_{\hat{\nu}} \leq C_0 e^{-\lambda(t-s)} \int_D \Big( |\Delta\Theta_t(\boldsymbol{\theta})| + \big(|\Delta\Theta_s(\boldsymbol{\theta})| + U_s(\boldsymbol{\theta}) + \bar{U}_s\big) e^{C_1(U_s(\boldsymbol{\theta})+\bar{U}_s)} \Big) \mu_0(d\boldsymbol{\theta})$$
$$\leq 2C_0 e^{-\lambda(t-s)} \int_D \Big( |\Delta\Theta_t(\boldsymbol{\theta})| + (|\Delta\Theta_s(\boldsymbol{\theta})| + U_s(\boldsymbol{\theta})) e^{C_1(U_s(\boldsymbol{\theta})+\bar{U}_s)} \Big) \mu_0(d\boldsymbol{\theta}) . \tag{270}$$

*(End of the proof of Lemma E.11.)* □

*Proof of Lemma E.10:* Lemma E.11 entails that, $\exists C, C' > 0$ such that

$$\|\Delta\Gamma_{t,s}\|_{\hat{\nu}}^2 \leq C e^{-2\lambda(t-s)} \left( \int_D \Big( |\Delta\Theta_t(\boldsymbol{\theta})| + \big(|\Delta\Theta_s(\boldsymbol{\theta})| + U_s(\boldsymbol{\theta})\big)e^{C'(U_s(\boldsymbol{\theta})+\bar{U}_s)} \Big) \mu_0(d\boldsymbol{\theta}) \right)^2$$
$$\leq 4C e^{-2\lambda(t-s)} \int_D |\Delta\Theta_t(\boldsymbol{\theta})|^2 + \Big( |\Delta\Theta_s(\boldsymbol{\theta})|^2 + U_s(\boldsymbol{\theta})^2 \Big)e^{2C'(U_s(\boldsymbol{\theta})+\bar{U}_s)}\mu_0(d\boldsymbol{\theta})$$
$$\leq 4C|D'|e^{-2\lambda(t-s)} \int_D |\Delta\Theta_t(\boldsymbol{\theta})| + \Big( |\Delta\Theta_s(\boldsymbol{\theta})| + U_s(\boldsymbol{\theta})^2 \Big)e^{2C'(U_s(\boldsymbol{\theta})+\bar{U}_s)}\mu_0(d\boldsymbol{\theta}), \tag{271}$$

where for the last inequality, we assume that $|D'| \geq 1$ (or, to accommodate the more general case, just replace $|D'|$ by $\max\{|D'|, 1\}$).

To prove Lemma E.10, the first goal is to show

$$\lim_{t_0 \to \infty} \int_{t_0}^\infty \int_{t_0}^t \|\Delta\Gamma_{t,s}\|_{\hat{\nu}}^2 ds dt = 0 . \tag{272}$$

There is

$$\int_{t_0}^{\infty} \int_{t_0}^{t} \|\Delta\Gamma_{t,s}\|_{\hat{\nu}}^2 ds dt$$

$$\leq 4C|D'| \int_D \int_{t_0}^{\infty} \int_{t_0}^{t} e^{-2\lambda(t-s)} \left( |\Delta\boldsymbol{\Theta}_t(\boldsymbol{\theta})| + \left( |\Delta\boldsymbol{\Theta}_s(\boldsymbol{\theta})| + U_s(\boldsymbol{\theta})^2 \right) e^{2C'(U_s(\boldsymbol{\theta}) + \bar{U}_s)} \right) ds dt \mu_0(d\boldsymbol{\theta})$$

$$\leq 4C|D'| \int_D \left( \int_{t_0}^{\infty} \left( \int_{t_0}^{t} e^{-2\lambda(t-s)} ds \right) |\Delta\boldsymbol{\Theta}_t(\boldsymbol{\theta})| dt \right.$$

$$\left. + \int_{t_0}^{\infty} \left( \int_s^{\infty} e^{-2\lambda(t-s)} dt \right) \left( |\Delta\boldsymbol{\Theta}_s(\boldsymbol{\theta})| + U_s(\boldsymbol{\theta})^2 \right) e^{2C'(U_s(\boldsymbol{\theta}) + \bar{U}_s)} ds \right) \mu_0(d\boldsymbol{\theta})$$

$$\leq 2C|D'|\lambda^{-1} \int_D \left( \int_{t_0}^{\infty} |\Delta\boldsymbol{\Theta}_t(\boldsymbol{\theta})| dt + \int_{t_0}^{\infty} \left( |\Delta\boldsymbol{\Theta}_s(\boldsymbol{\theta})| + U_s(\boldsymbol{\theta})^2 \right) e^{2C'(U_s(\boldsymbol{\theta}) + \bar{U}_s)} ds \right) \mu_0(d\boldsymbol{\theta})$$

$$\leq 4C|D'|\lambda^{-1} \int_D \int_{t_0}^{\infty} \left( |\Delta\boldsymbol{\Theta}_s(\boldsymbol{\theta})| + U_s(\boldsymbol{\theta})^2 \right) e^{2C'(U_s(\boldsymbol{\theta}) + \bar{U}_s)} ds \mu_0(d\boldsymbol{\theta}) \,. \tag{273}$$

By our assumption, the RHS is finite for $t_0 > 0$. Hence, by taking $t_0$ large enough, the value of $\int_{t_0}^{\infty} \int_{t_0}^{t} \|\Delta\Gamma_{t,s}\|_{\hat{\nu}}^2 ds dt$ can be made arbitrarily close to zero.

The second goal is to show that $\forall t_0 > 0$,

$$\lim_{T \to \infty} \fint_{t_0}^{T} \int_0^{t_0} \|\Gamma_{t,s}\|^2 ds dt = 0 \,. \tag{274}$$

As a first step, we show that

$$\lim_{T \to \infty} \fint_{t_0}^{T} \int_0^{t_0} \|\Gamma_{t-s}^{\infty}\|_{\hat{\nu}}^2 ds dt = 0 \tag{275}$$

because $\forall \eta \in \mathcal{W}(\Omega)$, there is

$$|\langle \eta, \Gamma_{t-s}^{\infty} \eta \rangle_{\hat{\nu}}| = \int_D \left\langle \boldsymbol{b}(\boldsymbol{\theta}), e^{-t\nabla\nabla V_{\infty}(\boldsymbol{\Theta}_{\infty}(\boldsymbol{\theta}))} \boldsymbol{b}(\boldsymbol{\theta}) \right\rangle \mu_0(d\boldsymbol{\theta})$$

$$\leq e^{-\lambda(t-s)} \int_D |\boldsymbol{b}(\boldsymbol{\theta})|^2 \mu_0(d\boldsymbol{\theta}) \tag{276}$$

$$\leq e^{-\lambda(t-s)} \|\mathcal{M}_{\infty}\|_{\hat{\nu}} \|\eta\|_{\hat{\nu}}^2 \,,$$

where

$$\boldsymbol{b}(\boldsymbol{\theta}) = \int_{\Omega} \nabla\varphi(\boldsymbol{\Theta}_{\infty}(\boldsymbol{\theta}), \boldsymbol{x}) \eta(\boldsymbol{x}) \hat{\nu}(d\boldsymbol{x}). \tag{277}$$

and $\mathcal{M}_{\infty}$ is defined as $\mathcal{M}_{\infty} := \mathcal{B}_{\infty}^{\mathsf{T}} \mathcal{B}_{\infty}$, or concretely, for $\eta \in \mathcal{W}_L(\omega)$,

$$(\mathcal{M}_{\infty}\eta)(\boldsymbol{x}) := \int_{\Omega} \left( \int_D \nabla\varphi(\boldsymbol{\Theta}_{\infty}(\boldsymbol{\theta}'), \boldsymbol{x})^{\mathsf{T}} \nabla\varphi(\boldsymbol{\Theta}_{\infty}(\boldsymbol{\theta}'), \boldsymbol{x}') \mu_0(d\boldsymbol{\theta}') \right) \eta(\boldsymbol{x}') \hat{\nu}(d\boldsymbol{x}')$$

$$= \int_{\Omega} M(\boldsymbol{x}, \boldsymbol{x}', \mu_{\infty}) \eta(\boldsymbol{x}') \hat{\nu}(d\boldsymbol{x}') \,, \tag{278}$$

where

$$M(\boldsymbol{x}, \boldsymbol{x}', \mu_{\infty}) := \int_D \nabla\varphi(\boldsymbol{\Theta}_{\infty}(\boldsymbol{\theta}'), \boldsymbol{x}) \cdot \nabla\varphi(\boldsymbol{\Theta}_{\infty}(\boldsymbol{\theta}'), \boldsymbol{x}') \mu_0(d\boldsymbol{\theta}') \,. \tag{279}$$

In the ERM setting, $\mathcal{M}_{\infty}$ is effectively an $n \times n$ matrix. Thus,

$$\fint_{t_0}^{T} \int_0^{t_0} \|\Gamma_{t-s}^{\infty}\|_{\hat{\nu}}^2 ds dt \leq \fint_{t_0}^{T} \int_0^{t_0} e^{-2\lambda(t-s)} \|\mathcal{M}_{\infty}\|_{\hat{\nu}}^2 ds dt$$

$$\leq \|\mathcal{M}_{\infty}\|_{\hat{\nu}}^2 \fint_{t_0}^{T} e^{-2\lambda(t-t_0)} dt \to 0 \qquad \text{as} \quad T \to \infty \tag{280}$$

Hence, it is sufficient to show that

$$\lim_{T\to\infty} \fint_{t_0}^{T} \int_0^{t_0} \|\Delta\Gamma_{t,s}\|^2 ds dt = 0 .\tag{281}$$

We have

$$\int_{t_0}^{T} \int_0^{t_0} \|\Delta\Gamma_{t,s}\|^2 ds dt$$

$$\leq 4C|D'| \int_D \left( \int_{t_0}^{T} \left( \int_0^{t_0} e^{-2\lambda(t-s)} ds \right) |\Delta\Theta_t(\boldsymbol{\theta})| dt \right.$$

$$\left. + \int_0^{t_0} \left( \int_{t_0}^{T} e^{-2\lambda(t-s)} dt \right) \left( |\Delta\Theta_s(\boldsymbol{\theta})| + U_s(\boldsymbol{\theta})^2 \right) e^{2C'(U_s(\boldsymbol{\theta})+\bar{U}_s)} ds \right) \mu_0(d\boldsymbol{\theta})$$

$$\leq 2C|D'|\lambda^{-1} \int_D \left( \int_{t_0}^{T} e^{-2\lambda(t-t_0)} |\Delta\Theta_t(\boldsymbol{\theta})| dt \right.$$

$$\left. + \int_0^{t_0} e^{-2\lambda(t_0-s)} \left( |\Delta\Theta_s(\boldsymbol{\theta})| + U_s(\boldsymbol{\theta})^2 \right) e^{2C'(U_s(\boldsymbol{\theta})+\bar{U}_s)} ds \right) \mu_0(d\boldsymbol{\theta})$$

$$\leq 4C|D'|\lambda^{-1} \int_D \int_0^\infty \left( |\Delta\Theta_s(\boldsymbol{\theta})| + U_s(\boldsymbol{\theta})^2 \right) e^{2C'(U_s(\boldsymbol{\theta})+\bar{U}_s)} ds \mu_0(d\boldsymbol{\theta})$$

$$< \infty$$

(282)

by assumption (231). Therefore,

$$\fint_{t_0}^{T} \int_0^{t_0} \|\Delta\Gamma_{t,s}\|_\nu^2 ds dt = \frac{1}{T-t_0} \int_{t_0}^{T} \int_0^{t_0} \|\Delta\Gamma_{t,s}\|_\nu^2 ds dt \xrightarrow[T\to\infty]{} 0 .\tag{283}$$

This concludes the proof of Lemma E.10. □

### E.4.2   Interpretation of the Assumption (231)

Below, we will illustrate the assumption (231)

$$Q := \int_D \int_0^\infty \left( |\Delta\Theta_t(\boldsymbol{\theta})| + U_t(\boldsymbol{\theta})^2 \right) e^{C_1(U_t(\boldsymbol{\theta})+\bar{U}_t)} dt \mu_0(d\boldsymbol{\theta}) < \infty,\tag{284}$$

in Theorem 3.6 by giving examples that satisfy this condition.

First, consider an example where $\exists\kappa > 0, \alpha > 1$ such that $\forall\boldsymbol{\theta} \in \operatorname{supp}\mu_0$ and $\forall\, t > 0$,

$$|\Delta\Theta_t(\boldsymbol{\theta})| < \kappa(t+1)^{-\alpha},\tag{285}$$

that is, all characteristic flows share a uniform asymptotic convergence rate on the order of $t^{-\alpha}$. Then $\forall\boldsymbol{\theta} \in \operatorname{supp}\mu_0$,

$$U_t(\boldsymbol{\theta}) = \int_t^\infty |\Delta\Theta_s(\boldsymbol{\theta})| ds \leq \frac{\kappa}{\alpha-1}(t+1)^{-(\alpha-1)}\tag{286}$$

and thus

$$\bar{U}_t \leq \frac{\kappa}{\alpha-1}(t+1)^{-(\alpha-1)} .\tag{287}$$

Therefore,

$$Q \leq \int_D \int_0^\infty \left( |\Delta\Theta_t(\boldsymbol{\theta})| + U_t(\boldsymbol{\theta})^2 \right) e^{C_1(U_0(\boldsymbol{\theta})+\bar{U}_0)} dt \mu_0(d\boldsymbol{\theta})$$

$$\leq \int_0^\infty \left( \kappa(t+1)^{-\alpha} + \left( \frac{\kappa}{\alpha-1} \right)^2 (t+1)^{-2(\alpha-1)} \right) e^{\frac{2C_1\kappa}{\alpha-1}} dt,$$

(288)

which is finite as long as $\alpha > \frac{3}{2}$. Thus,

**Proposition E.12** *If $\exists\kappa > 0, \alpha > \frac{3}{2}$ such that $\forall\boldsymbol{\theta} \in \operatorname{supp}\mu_0$ and $\forall t \geq 0$,*

$$|\Delta\Theta_t(\boldsymbol{\theta})| = |\Theta_t(\boldsymbol{\theta}) - \Theta_\infty(\boldsymbol{\theta})| < \kappa(t+1)^{-\alpha},\tag{289}$$

*then the condition (231) is satisfied.*

Moreover, the assumption allows flexibility in having non-uniform convergence rate for different characteristic flows, $\Theta_t(\boldsymbol{\theta})$. Suppose that $\exists \kappa : \operatorname{supp} \mu_0 \to \mathbb{R}_+$ and $\alpha > \frac{3}{2}$ such that $\forall \theta \in \operatorname{supp} \mu_0$,

$$|\Delta \Theta_t(\boldsymbol{\theta})| < \kappa(\boldsymbol{\theta})(t+1)^{-\alpha} . \tag{290}$$

Then

$$U_t(\boldsymbol{\theta}) = \int_t^\infty |\Delta \Theta_s(\boldsymbol{\theta})| ds \le \frac{\kappa}{\alpha - 1}(t+1)^{-(\alpha - 1)} \tag{291}$$

and so

$$\begin{aligned}
Q &\le \int_D \int_0^\infty \left( |\Delta \Theta_t(\boldsymbol{\theta})| + U_t(\boldsymbol{\theta})^2 \right) e^{2C_1(U_0(\boldsymbol{\theta}))} dt \mu_0(d\boldsymbol{\theta}) \\
&\le \int_D \int_0^\infty \left( \kappa(\boldsymbol{\theta})(t+1)^{-\alpha} + \left( \frac{\kappa(\boldsymbol{\theta})}{\alpha - 1} \right)^2 (t+1)^{-2(\alpha-1)} \right) e^{\frac{2C_1 \kappa(\boldsymbol{\theta})}{\alpha - 1}} dt \\
&\le C_2 \int_D \left( \kappa(\boldsymbol{\theta}) + \kappa(\boldsymbol{\theta})^2 \right) e^{\frac{2C_1 \kappa(\boldsymbol{\theta})}{\alpha - 1}} \mu_0(d\boldsymbol{\theta}) .
\end{aligned} \tag{292}$$

Therefore,

**Proposition E.13** *Suppose* $\exists \alpha > \frac{3}{2}$ *and a function* $\kappa : \operatorname{supp} \mu_0 \to \mathbb{R}_+$, *which satisfies*

$$\int_D \left( \kappa(\boldsymbol{\theta}) + \kappa(\boldsymbol{\theta})^2 \right) e^{\frac{2C_1 \kappa(\boldsymbol{\theta})}{\alpha - 1}} \mu_0(d\boldsymbol{\theta}) < \infty, \tag{293}$$

*such that* $\forall \boldsymbol{\theta} \in \operatorname{supp} \mu_0$,

$$|\Delta \Theta_t(\boldsymbol{\theta})| = |\Theta_t(\boldsymbol{\theta}) - \Theta_\infty(\boldsymbol{\theta})| \le \kappa(\boldsymbol{\theta})(t+1)^{-\alpha} . \tag{294}$$

*Then the condition* (231) *is satisfied.*

### E.5 Relationship between Theorem 3.6 and [11]

As a comparison to our result, Chizat [11, Theorem 3.8] shows that under assumptions including (45) as well as the uniqueness and sparseness of the global minimizer, an alternative type of particle gradient descent (with a different homogeneity degree in the loss function and under the conic metric, which give rise to gradient flow in Wasserstein-Fisher-Rao metric instead of Wasserstein metric) converges to the global minimizer for large enough $m$ (depending exponentially on $d$) with a uniform rate. This implies that in that setting, $\lim_{t\to\infty} \lim_{m\to\infty} m \| f_t^{(m)} - f_t \|_\nu^2 = \lim_{m\to\infty} \lim_{t\to\infty} m \| f_t^{(m)} - f_t \|_\nu^2 = 0$, $\mathbb{P}_0$-almost surely.

## F The Monte-Carlo bound and variation norm

The bound (41) on the long-time fluctuations motivates us to control the term $\int_D \|\varphi(\boldsymbol{\theta}, \cdot)\|_\nu^2 \mu_\infty(d\boldsymbol{\theta})$ using a suitable choice of regularization in (3). In the following, we restrict our attention to the shallow neural networks setting, and further assume that

**Assumption F.1** $\hat{D}$ *is compact.*

Under this assumption, there is

$$\int_D \|\varphi(\boldsymbol{\theta}, \cdot)\|_\nu^2 \mu(d\boldsymbol{\theta}) = \int_D \int_\Omega |\varphi(\boldsymbol{\theta}, \boldsymbol{x})|^2 \hat{\nu}(d\boldsymbol{x}) \mu(d\boldsymbol{\theta}) \le \hat{K}_M \int_D c^2 \mu(d\boldsymbol{\theta}) , \tag{295}$$

where $\hat{K}_M = \max_{\boldsymbol{z} \in \hat{D}} \|\hat{\varphi}(\boldsymbol{z}, \cdot)\|_\nu^2$. Thus, we consider regularization with $r(\boldsymbol{\theta}) = \frac{1}{2} c^2$, in which case (3) becomes

$$\min_{\mu \in \mathcal{P}(D)} \mathcal{L}(\mu) \quad \text{with} \quad \mathcal{L}(\mu) := \frac{1}{2} \| f[\mu] - f_* \|_\nu^2 + \frac{1}{2} \lambda \int_D c^2 \mu(d\boldsymbol{\theta}) . \tag{296}$$

Interestingly, this choice of regularization leads to learning in the function space $\mathcal{F}_1$ [5] associated with $\hat{\varphi}$, which is equipped with the *variation norm* defined as

$$|\gamma_q(f)| := \inf_{\mu \in \mathcal{P}(D)} \left\{ \int_D |c|^q \mu(d\boldsymbol{\theta}); \ f(\boldsymbol{x}) = \int_D c\hat{\varphi}(\boldsymbol{z}, \boldsymbol{x}) \mu(d\boldsymbol{\theta}) \right\} = |\gamma_1(f)|^q , \qquad q \ge 1 . \tag{297}$$

We call $\int_D |c|^q \mu(d\boldsymbol{\theta})$ the *q-norm* of $\mu$. One can verify [44, Proposition 1] that indeed, using any $q \geq 1$ above yields the same norm because $\mu$, the object defining the integral representation (2), is in fact a *lifted* version of a more 'fundamental' object $\gamma = \int_{\mathbb{R}} c\mu(dc, \cdot) \in \mathcal{M}(\hat{D})$, the space of signed Radon measures over $\hat{D}$. They are related via the projection

$$\int_{\hat{D}} \chi(\boldsymbol{z}) \gamma(d\boldsymbol{z}) = \int_D c\chi(\boldsymbol{z}) \mu(d\boldsymbol{\theta}) \tag{298}$$

for all continuous test functions $\chi : \hat{D} \to \mathbb{R}$. One can also verify [11] that $\gamma_1(f) = \inf\{\|\gamma\|_{\mathrm{TV}}; f(\boldsymbol{x}) = \int_{\hat{D}} \hat{\varphi}(\boldsymbol{z}, \boldsymbol{x}) \gamma(d\boldsymbol{z})\}$, where $\|\gamma\|_{\mathrm{TV}}$ is the *total variation* of $\gamma$ [5].

The space $\mathcal{F}_1$ contains any RKHS whose kernel is generated as an expectation over features $k(\boldsymbol{x}, \boldsymbol{x}') = \int_{\hat{D}} \hat{\varphi}(\boldsymbol{z}, \boldsymbol{x}) \hat{\varphi}(\boldsymbol{z}, \boldsymbol{x}') \hat{\mu}_0(d\boldsymbol{z})$ with a base measure $\hat{\mu}_0 \in \mathcal{P}(\hat{D})$, but it provides crucial approximation advantages over such RKHS at approximating certain non-smooth, high-dimensional functions with hidden low-dimensional structure, giving rise to powerful generalization guarantees [5]. This also motivates the study of overparametrized shallow networks with the scaling as in (1), as opposed to the NTK scaling of $m^{-1/2}$ [36].

To learn in $\mathcal{F}_1$, a canonical approach is to consider the ERM problem

$$\min_{f \in \mathcal{F}_1} \tfrac{1}{2} \|f - f_*\|_{\hat{\nu}}^2 + \tfrac{1}{2} \lambda \gamma_1(f), \tag{299}$$

By (297), this is equivalent to (296). Next, we prove the following proposition, which characterizes the properties of the minimizers and shows that the measure obtained from (296) indeed has its 2-norm controlled:

**Proposition F.2** *Under Assumptions 2.1, 2.2, and F.1, the minimizers of the loss $\mathcal{L}(\mu)$ defined in (3) are all in the form*

$$\mu_\lambda(dc, d\boldsymbol{z}) = \delta_{c_\lambda}(dc) \hat{\mu}_+(d\boldsymbol{z}) + \delta_{-c_\lambda}(dc) \hat{\mu}_-(d\boldsymbol{z}) \tag{300}$$

*where $c_\lambda \geq 0$ and $\hat{\mu}_\pm \in \mathcal{P}(\hat{D})$ satisfy*

$$\forall \boldsymbol{z} \in \operatorname{supp} \hat{\mu}_- \; : \; -\hat{F}(\boldsymbol{z}) + c_\lambda \int_{\hat{D}} \hat{K}(\boldsymbol{z}, \boldsymbol{z}') \left( \hat{\mu}_+(d\boldsymbol{z}') - \hat{\mu}_-(d\boldsymbol{z}') \right) = \lambda c_\lambda,$$

$$\forall \boldsymbol{z} \in \operatorname{supp} \hat{\mu}_+ \; : \; -\hat{F}(\boldsymbol{z}) + c_\lambda \int_{\hat{D}} \hat{K}(\boldsymbol{z}, \boldsymbol{z}') \left( \hat{\mu}_+(d\boldsymbol{z}') - \hat{\mu}_-(d\boldsymbol{z}') \right) = -\lambda c_\lambda, \tag{301}$$

$$\forall \boldsymbol{z} \in \hat{D} \qquad : \; \left| -\hat{F}(\boldsymbol{z}) + c_\lambda \int_{\hat{D}} \hat{K}(\boldsymbol{z}, \boldsymbol{z}') \left( \hat{\mu}_+(d\boldsymbol{z}') - \hat{\mu}_-(d\boldsymbol{z}') \right) \right| \leq \lambda c_\lambda.$$

*In addition, the constant $c_\lambda$ is unique and positive if $\hat{F}(\boldsymbol{z})$ is not identically zero on $\hat{D}$, the closure of the supports of $\hat{\mu}_\pm$ are disjoint (i.e. $\overline{\operatorname{supp} \hat{\mu}_+} \cap \overline{\operatorname{supp} \hat{\mu}_-} = \emptyset$), and the function*

$$f_\lambda = \int_D c\hat{\varphi}(\boldsymbol{z}, \cdot) \mu_\lambda(dc, d\boldsymbol{z}) = c_\lambda \int_{\hat{D}} \hat{\varphi}(\boldsymbol{z}, \cdot) \left( \hat{\mu}_+(d\boldsymbol{z}) - \hat{\mu}_-(d\boldsymbol{z}) \right) \tag{302}$$

*is the same for all minimizers and satisfies*

$$\tfrac{1}{4} \lambda^2 |c_\lambda|^2 \hat{K}_M^{-1} \leq \|f_* - f_\lambda\|_{\hat{\nu}}^2, \qquad \|f_* - f_\lambda\|_{\hat{\nu}}^2 + \lambda |c_\lambda|^2 \leq \lambda |\gamma_1(f_*)|^2. \tag{303}$$

*where $\hat{K}_M = \max_{\boldsymbol{z} \in \hat{D}} \|\hat{\varphi}(\boldsymbol{z}, \cdot)\|_{\hat{\nu}}^2 = \max_{\boldsymbol{z} \in \hat{D}} \hat{K}(\boldsymbol{z}, \boldsymbol{z})$.*

**Remark F.3** *Note that the proposition automatically implies that $\gamma_1(f_\lambda) \leq \gamma_1(f_*) < \infty$. It also implies that*

$$\int_D |c|^q \mu_\lambda(dc, d\boldsymbol{z}) = |c_\lambda|^q = |\gamma_\lambda|_{TV}^q \leq |\gamma_1(f_*)|^q \qquad \forall q \in \mathbb{R}_+ \tag{304}$$

*where $\gamma_\lambda = \int_{\mathbb{R}} c\mu_\lambda(dc, \cdot)$. Finally note that the proposition holds if we replace the empirical loss by the population loss.*

*Proof of Proposition F.2:* The fact that this loss can only be minimized by minimizers follows from the compactness of the sets $\{\mu \in \mathcal{P}(D) : \mathcal{L}(\mu) \leq u, u \in \mathbb{R}\}$. The minimizers of $\mathcal{L}(\mu)$ must satisfy the following Euler-Lagrange equations [57]:

$$\forall (c, \boldsymbol{z}) \in D \quad : \quad -c\hat{F}(\boldsymbol{z}) + c \int_D c' \hat{K}(\boldsymbol{z}, \boldsymbol{z}') \mu(dc', d\boldsymbol{z}') + \tfrac{1}{2} \lambda |c|^2 \equiv c\hat{V}(\boldsymbol{z}) + \tfrac{1}{2} \lambda |c|^2 \geq \bar{V}, \tag{305}$$

with equality on the support of $\mu$ and where $\bar{V}$ is the expectation of the left hand side with respect to $\mu(dc, d\boldsymbol{z})$. Minimizing the left hand side of (305) over $c$ at fixed $\boldsymbol{z}$, we deduce that

$$\forall \boldsymbol{z} \in \hat{D} \quad : \quad \min_c \left( c\hat{V}(\boldsymbol{z}) + \tfrac{1}{2}\lambda|c|^2 \right) \geq \bar{V} , \tag{306}$$

with equality for $\boldsymbol{z}$ in the support of $\hat{\mu} = \int_{\mathbb{R}} \mu(dc, \cdot)$. This means that for any $\boldsymbol{z} \in \operatorname{supp}\hat{\mu}$, there can only be one $c = c(\boldsymbol{z})$ in $\operatorname{supp}\mu$, with $c(\boldsymbol{z})$ satisfying the Euler-Lagrange equation associated with (306)

$$\hat{V}(\boldsymbol{z}) + \lambda c(\boldsymbol{z}) = 0 \quad \Leftrightarrow \quad \hat{V}(\boldsymbol{z}) = -\lambda c(\boldsymbol{z}) \tag{307}$$

If we insert this equality back in $c(\boldsymbol{z})\hat{V}(\boldsymbol{z}) + \tfrac{1}{2}\lambda|c(\boldsymbol{z})|^2 = \bar{V}$, we deduce that $|c(\boldsymbol{z})| = c_\lambda$, with the constant $c_\lambda$ related to $\bar{V}$ as

$$-\frac{1}{2}\lambda|c_\lambda|^2 = \bar{V} , \tag{308}$$

and furthermore, $\forall \boldsymbol{z} \in \operatorname{supp}\hat{\mu}$,

$$\hat{V}(\boldsymbol{z}) = \begin{cases} -\lambda c_\lambda & \text{if } c(\boldsymbol{z}) = c_\lambda \\ \lambda c_\lambda & \text{if } c(\boldsymbol{z}) = -c_\lambda \end{cases} . \tag{309}$$

These considerations imply that the minimizer must be of the form (300), and if we combine (306) and (308) and evaluate the minimum on $c$ explicitly we deduce that $\hat{\mu}_\pm$ and $c_\lambda$ must satisfy the equations in (301). It is also clear from (301) that we must have $\overline{\operatorname{supp}\hat{\mu}_+} \cap \overline{\operatorname{supp}\hat{\mu}_-} = \emptyset$: indeed if there was a point $\boldsymbol{z} \in \overline{\operatorname{supp}\hat{\mu}_+} \cap \overline{\operatorname{supp}\hat{\mu}_-}$, then at that point $\hat{V}(\boldsymbol{z})$ would be discontinuous, which is not possible since this function is continuously differentiable for any $\mu$ by our assumptions on $\hat{\varphi}$. Finally, to show that we must have that $c_\lambda > 0$ if $F(\boldsymbol{z})$ is not identically zero on $\hat{D}$, note that if $c_\lambda = 0$, (305) reduces to

$$\forall (c, \boldsymbol{z}) \in D \quad : \quad -c\hat{F}(\boldsymbol{z}) + \tfrac{1}{2}\lambda|c|^2 \geq 0 \tag{310}$$

which can only be satisfied if $\hat{F}(\boldsymbol{z}) = 0$.

To show that $c_\lambda$ and the function in (302) are unique, let $\mu_\lambda$ and $\mu'_\lambda$ be two different minimizers and consider

$$f_\lambda = \int_D c\hat{\varphi}(\boldsymbol{z}, \cdot)\mu_\lambda(dc, d\boldsymbol{z}) \quad \text{and} \quad f'_\lambda = \int_D c\hat{\varphi}(\boldsymbol{z}, \cdot)\mu'_\lambda(dc, d\boldsymbol{z}) \tag{311}$$

Let us evaluate the loss on $a\mu_\lambda + (1-a)\mu'_\lambda \in \mathcal{P}(D)$ with $a \in [0, 1]$. By convexity of $\mathcal{E}_\lambda$ we have

$$\mathcal{L}(a\mu_\lambda + (1-a)\mu'_\lambda) \leq a\mathcal{L}(\mu_\lambda) + (1-a)\mathcal{L}(\mu'_\lambda) = \mathcal{L}(\mu_\lambda) = \mathcal{L}(\mu'_\lambda) \tag{312}$$

Since $a\mu_\lambda + (1-a)\mu'_\lambda$ cannot have a lower loss than this minimum, we must have equality in (312), which reduces to

$$\begin{aligned}
&\|f_* - af_\lambda - (1-a)f'_\lambda\|_{\hat{\nu}}^2 + a\lambda|c_\lambda|^2 + (1-a)\lambda|c'_\lambda|^2 \\
&= \|f_* - f_\lambda\|_{\hat{\nu}}^2 + \lambda|c_\lambda|^2 \\
&= \|f_* - f'_\lambda\|_{\hat{\nu}}^2 + \lambda|c'_\lambda|^2 ,
\end{aligned} \tag{313}$$

where $c_\lambda$ and $c'_\lambda$ are associated with $\mu_\lambda$ and $\mu'_\lambda$, respectively. Clearly these equations can only be fulfilled for all $a \in [0, 1]$ if $c_\lambda = c'_\lambda$ and $f_\lambda = f'_\lambda$ $\hat{\nu}$-a.e. on $\Omega$.

To establish (303), notice that if $\mu_\lambda$ is a minimizer and $f_\lambda$ is given by (302), then we can derive from (309) that

$$-\int_\Omega f_\lambda(\boldsymbol{x})f_*(\boldsymbol{x})\hat{\nu}(d\boldsymbol{x}) + \|f_\lambda\|_{\hat{\nu}}^2 + \lambda|c_\lambda|^2 = 0. \tag{314}$$

This gives, using Cauchy-Schwartz,

$$\lambda|c_\lambda|^2 = \int_\Omega f_\lambda(\boldsymbol{x})(f_*(\boldsymbol{x}) - f_\lambda(\boldsymbol{x}))\hat{\nu}(d\boldsymbol{x}) \leq \|f_\lambda\|_{\hat{\nu}} \|f_* - f_\lambda\|_{\hat{\nu}} . \tag{315}$$

Now notice that

$$\|f_\lambda\|_{\hat{\nu}}^2 = c_\lambda^2 \int_{\hat{D}\times\hat{D}} \hat{K}(\boldsymbol{z}, \boldsymbol{z}') \left(\hat{\mu}_+(d\boldsymbol{z}) - \hat{\mu}_-(d\boldsymbol{z})\right)\left(\hat{\mu}_+(d\boldsymbol{z}') - \hat{\mu}_-(d\boldsymbol{z}')\right) \leq 4c_\lambda^2 \hat{K}_M . \tag{316}$$

Using (316) in (315) and reorganizing gives the first inequality in (303). To establish the second, let $\mu_* \in \mathcal{M}_+(D)$ be the measure that minimizes $\int_D |c| \mu(dc, d\boldsymbol{z})$ under the constraint that $f_* = \int_D c\hat{\varphi}(\boldsymbol{z}, \cdot)\mu_*(dc, d\boldsymbol{z})$, so that $\int_D |c|\mu_*(dc, d\boldsymbol{z}) = \gamma_1(f_*)$—the measure $\mu_*$ exists since we assumed that $f_* \in \mathcal{F}_1$. Evaluated on $\mu_*$, the loss is

$$\mathcal{L}(\mu_*) = \lambda |\gamma_1(f_*)|^2. \tag{317}$$

Any minimizer $\mu_\lambda$ of $\mathcal{L}(\mu)$ must do at least as well, i.e we must have

$$\|f_* - f_\lambda\|_{\hat{\nu}}^2 + \lambda \int_D |c|^2 \mu_\lambda(dc, d\boldsymbol{z}) = \|f_* - f_\lambda\|_{\hat{\nu}}^2 + \lambda|c_\lambda|^2 \le \lambda|\gamma(f_*)|^2. \tag{318}$$

This establishes the second inequality in (303). $\qquad\square$

## G  Additional Details of the Experiments

### G.1  Setup of the experiments reported in Section 4

The numerical experiments reported in Figure 1 are under the student-teacher setting, where a teacher network gives the target function for the student network to learn, with both the student and the teacher being shallow neural networks. Both $\hat{D}$ and $\Omega$ are taken to be the unit sphere of $d = 16$ dimensions, and we take $\hat{\varphi}(\boldsymbol{z}, \boldsymbol{x}) = \max(0, \langle \boldsymbol{z}, \boldsymbol{x} \rangle)$. The teacher network has two neurons, $(c_1, \boldsymbol{z}_1)$ and $(c_2, \boldsymbol{z}_2)$, in the hidden layer, with $c_1 = c_2 = 1$ and $\boldsymbol{z}_1$ and $\boldsymbol{z}_2$ sampled i.i.d. from the uniform distribution on $\hat{D}$ and then fixed across the experiments. We vary the width of the student network in the range of $m = 128, 256, 512, 1024$ and $2048$, with their $\boldsymbol{z}$'s sampled i.i.d. from the uniform distribution on $\hat{D}$ and their $c$'s sampled i.i.d. from $\mathcal{N}(0, 1)$. We train the networks in the ERM setting, where we sample $n = 32$ vectors i.i.d. from the uniform distribution $\nu$ on $\Omega$ as the training dataset, which then define the empirical data measure $\hat{\nu}(d\boldsymbol{x}) = \frac{1}{n} \sum_{l=1}^n \delta_{\boldsymbol{x}_l}(d\boldsymbol{x})$. We use the mean squared error as the loss function. We rescale both the squared loss and the gradient by $d$ in order to adjust to the $\frac{1}{d}$ factor resulting from spherical integrals. The models are trained for 20000 epochs with learning rate (which is multiplied to the RHS of (8)) set to be 1. For each choice of $m$, we run the experiment $\kappa = 20$ times with different random initializations of the student network. The average fluctuation is defined as $\frac{1}{\kappa} \sum_{k=1}^\kappa \|f_k^{(m)} - \bar{f}^{(m)}\|_{\hat{\nu}}^2$ for the training loss and $\frac{1}{\kappa} \sum_{k=1}^\kappa \|f_k^{(m)} - \bar{f}^{(m)}\|_\nu^2$ for the exact population loss, with $\bar{f}^{(m)} = \frac{1}{\kappa} \sum_{k=1}^\kappa f_k^{(m)}$ being the averaged model, similar to the approach in [28]. The other plotted quantities – loss, TV-norm and 2-norm – are averaged across the $\kappa$ number of runs. The TV-norm (i.e., 1-norm) and 2-norm are defined as in Appendix F.

In addition, we also run the same set of experiments except for initializing the $c$'s of the student networks at 0 instead of i.i.d. from $\mathcal{N}(0, 1)$, for the unregularized case. We show the results in Figure 2. Adding regularization or using zero-initialization both result in lower TV-norm and 2-norm, and moreover, lower average fluctuation and (slightly) lower average value of the *population* loss. This demonstrates their positive effects on both approximation and generalization.

### G.2  Additional experiments using the population loss

In addition to the experiments described above, we also train the networks under the population loss. In this setting, we optimize the student networks by gradient descent under the population loss where the data distribution $\nu$ is uniform on $\Omega$, which allows an analytical formula for the loss value and the gradient using spherical integrals. We also consider both of the initialization schemes of the $c$'s of the student network described above.

The results are shown in Figure 3. We observe that the mean squared fluctuations remain at a $1/m$ scaling with a general tendency to decay over time. In the unregularized case with non-zero initialization, the mean squared fluctuation decays at the same rate for different $m$ in roughly the first $10^3$ epochs, after which it decays faster for smaller $m$. Interestingly, this coincides with the tendency for student neurons with $\boldsymbol{z}$ not aligned with the teacher neurons to slowly have their $|c|$ decrease to zero due to a finite-$m$ effect, which is also reflected in the decrease in TV-norm. Aside from this, the mean-squared fluctuations decay at similar rates for different choices of $m$, which is consistent with our theory, since their dynamics are governed by the same dynamical CLT. Moreover, the values of the fluctuations for these choices of $m$ are indeed lower than the asymptotic Monte-Carlo bound

Figure 2: The first two rows are a replicate of Figure 1, whereas *Row 3* shows the result for using zero-initialization of the $c$'s in the student network and without regularizaton.

given in (41) with $\mu_\infty$ and $f_\infty$ replaced by the target measure and function, respectively, as well as $\hat{\nu}$ by $\nu$, whose analytical expression and numerical value in this setup are given in Appendix H. We also see that the average loss values remain similar over time for different choices of $m$, justifying the approximation by a mean-field dynamics. Also, we notice that with either regularization or zero-initialization, the student neurons are aligned with one of the teacher neurons in both $z$ and $c$ after training, which then results in lower TV-norms and 2-norms than using non-zero initialization and without regularization.

The code is implemented in C++ and run on a cluster with single CPU. For $n = 128, 256, 512, 1024$ and 2048, the approximate running times of each run of 20000 epochs are 1 min, 2 min, 8 min, 30 min and 150 min, respectively.

The hyperparameter in regularization, $\lambda$, is manually selected from $0.01, 0.05$ and $0.1$.

### G.3    Additional experiments with a non-planted target

We also conduct an experiment in which the target function is not given by a teacher network. $\hat{D}, \Omega,$ $\hat{\varphi}$ as well as the widths of the student networks remain the same as in the previous experiments. The target function is $f_*(\boldsymbol{x}) = \int_{\hat{D}} \hat{\varphi}(\boldsymbol{z}, \boldsymbol{x}) \hat{\mu}_*(d\boldsymbol{z})$, where $\hat{\mu}_*$ is the uniform measure on the 1-dimensional great circle in the first 2 dimensions, i.e., $\{(\cos\theta, \sin\theta), 0, ..., 0 : \theta \in [0, 2\pi)\} \subseteq \mathbb{S}^d$. The student networks are trained using gradient descent under the population loss where the data distribution $\nu$ is uniform on $\Omega$, which allows an analytical formula for the gradient using spherical integrals, similar to the experiments in Appendix G.2.

The results are shown in Figure 4. We observe that the behavior of the fluctuations are similar to those found in Figure 3.

## H    Analytical Calculations of the Resampling Error

Derivations similar to the one presented here can be found in [53, 15, 5]. In the setting of ReLU without bias on unit sphere, we take $\hat{D} = \Omega = \mathbb{S}^d \subseteq \mathbb{R}^{d+1}$, $\hat{\varphi}(\boldsymbol{z}, \boldsymbol{x}) = \max(\langle \boldsymbol{z}, \boldsymbol{x} \rangle, 0)$, and $\nu$ is equal to the uniform measure on $\mathbb{S}^d$. In this case,

$$\hat{K}(\boldsymbol{z}, \boldsymbol{z}') = \int_\Omega \hat{\varphi}(\boldsymbol{z}, \boldsymbol{x})\hat{\varphi}(\boldsymbol{z}', \boldsymbol{x})\nu(d\boldsymbol{x}) = \frac{1}{2(d+1)\pi}(\sin\alpha + (\pi - \alpha)\cos\alpha), \qquad (319)$$

Figure 3: Results of the experiments in the student-teacher setting and where the student networks are trained by gradient descent on the *population* loss, as described in Appendix G.2. Each row corresponds to one setup. *Row 1*: Using unregularized loss and non-zero-initialization; *Row 2*: Using regularized loss with $\lambda = 0.01$ and non-zero-initialization; *Row 3*: Using unregularized loss and zero-initialization. In each row, *Column 1* plots the trajectory of the neurons, $\boldsymbol{\theta}_i = (c_i, \boldsymbol{z}_i)$, of a student network of width 128 during its training, with $x$-coordinate being the angle between $\boldsymbol{z}_i$ and that of a chosen teacher's neuron and $y$-coordinate being $c_i$. The yellow dots, blue dots and cyan curves marking their initial values, terminal values, and trajectory during training. *Columns 2-5* plot the average fluctuations (scaled by $m$), average loss, average TV norm, and average 2-norm during training, respectively, computed across $\kappa = 20$ runs with different random initializations of the student network for each choice of $m$. In *Column 2*, the *solid* curves give the average fluctuation in the exact population loss and the black horizontal *dashed* line gives the asymptotic Monte-Carlo bound in (41) computed in Appendix H for this setting. In *Column 3*, the *solid* curves give the total *population* loss, and the *dotted* curves give the unregularized *population* loss (for the regularized case only). In *Columns 4* and *5*, the horizontal *dashed* line gives the relevant norm of the teacher network.

with $\alpha$ being the angle between $\boldsymbol{z}$ and $\boldsymbol{z}'$, and

$$\int_\Omega |\hat{\varphi}(\boldsymbol{z}, \boldsymbol{x})|^2 \nu(d\boldsymbol{x}) = \frac{1}{2} \int_\Omega (\langle \boldsymbol{x}, \boldsymbol{z} \rangle)^2 \nu(d\boldsymbol{x}) = \frac{1}{2(d+1)} \tag{320}$$

Thus, taking $\mu_*$ to be the measure representing the teacher network, $\mu_* = \frac{1}{m_t} \sum_{i=1}^{m_t} \delta_{\boldsymbol{z}_i}(d\boldsymbol{z})\delta_1(dc)$, we have

$$\begin{aligned}
\int_D \|\varphi(\boldsymbol{\theta}, \cdot)\|_\nu^2 \mu_*(d\boldsymbol{\theta}) &= \int_D \int_\Omega |\varphi(\boldsymbol{\theta}, \boldsymbol{x})|^2 \nu(d\boldsymbol{x})\mu_*(d\boldsymbol{\theta}) \\
&= \int_D \frac{c^2}{2(d+1)} \mu_*(d\boldsymbol{\theta}) \\
&= \frac{1}{2(d+1)}
\end{aligned} \tag{321}$$

On the other hand,

$$\begin{aligned}
\|f_*\|_\nu^2 &= \int_\Omega \left| \int_D \varphi(\boldsymbol{\theta}, \boldsymbol{x})\mu_*(d\boldsymbol{\theta}) \right|^2 \nu(d\boldsymbol{x}) \\
&= \int_D \int_D cc' \hat{K}(\boldsymbol{z}, \boldsymbol{z}')\mu_*(d\boldsymbol{\theta})\mu_*(d\boldsymbol{\theta}') \\
&= \frac{1}{m_t^2} \sum_{i,j=1}^{m_t} \hat{K}(\boldsymbol{z}_i, \boldsymbol{z}_j)
\end{aligned} \tag{322}$$

Figure 4: Results of the experiments with a non-planted target using the population loss, as described in G.3. *Row 1*: Using unregularized loss and non-zero-initialization; *Row 2*: Using regularized loss with $\lambda = 0.01$ and non-zero-initialization; *Row 3*: Using unregularized loss and zero-initialization. In each row, *Column 1* plots the projection of the neurons' $z_i$ in the first two dimension. The other columns are under the same setting as Figure 3.

In the experiments described in the main text, we take $m_t = 2$, and $z_1$ and $z_2$ are initialized with a fixed random seed such that their angle, $\alpha_{12}$, equal to 1.766. Thus,

$$\|f_*\|_\nu^2 = \frac{1}{4(d+1)\pi}(0 + \pi) + \frac{1}{4(d+1)\pi}(\sin \alpha_{12} + (\pi - \alpha_{12})\cos \alpha_{12}) \approx 0.012 \qquad (323)$$

Together, we get a numerical value of the RHS of (41) if we replace $\mu_\infty$, $f_\infty$ and $\hat\nu$ by $\mu_*$, $f_*$ and $\nu$, respectively.