[Reviews · NeurIPS 2020]

Review 1

Summary and Contributions: Update: After discussing the paper with other reviewers, I realized there was less novelty in the paper's results than I had originally thought. The authors' response also clarified a confusion I had around the paper's implications on generalization. Given this I am reducing my score by 1 point. __________________________________________________________________________ The authors study the dynamics of single-hidden layer, wide neural networks in the mean-field limit. Specifically, they characterize the fluctuations for finite widths around the infinite-width dynamics. They provide a bound on these fluctuations that does not depend on the dimension of the data, and under certain conditions show the long-time fluctuations can be controlled with an MC type resampling error.

Strengths: Although the results are still asymptotic, this work studies the important problem of relating the behavior of finite-width networks to the previously studied infinite-width limits. The work is technically strong in its characterization of the fluctuations, and to my knowledge is the first work to do so in this limit.

Weaknesses: The main weakness of the paper is probably the empirical section. I'm quite happy with the student-teacher toy dataset case, given the complexity of the theory, but even here I found some of the results confusing. Hopefully the authors can clarify and add some more discussion. 1) Why does the average fluctuation decay with t? My understanding is that this observation is not explained by the theoretical results. 2) A motivation for the work, was the better understand the generalization properties of overparameterized NNs. This is achieved in some sense by obtaining a dimension-free bound on the generalization error, but there is no dependence on the amount of over parameterization? Specifically, in the third column of Fig. 1 the NNs perform similarly for all widths.

Correctness: The propositions seem correct and reasonable to me, although I did not check their proofs in the supplement in detail.

Clarity: I learned a lot from reading this work, as I was less familiar with the mean-field (rather than the NTK) limit. I think that speaks to the paper being well written and structured. I appreciated the interpretation of various terms in the propositions to build intuition. However, given that a lot of technical material is covered, I think perhaps an expository figure illustrating the results could make the article accessible to a wider section of the NeurIPS community.

Relation to Prior Work: As mentioned, I am less familiar with this line of work, but I felt it gave a good overview of previous contributions that I was aware of. One exception to this, which should be cited, is "Dynamics of Deep Neural Networks and Neural Tangent Hierarchy" by Huang and Yau, which studies fluctuation for the NTK scaling.

Reproducibility: Yes

Additional Feedback: It is mentioned that results hold with \hat{\nu} replaced with \nu. Is this true for all results? My understanding is that to say something about generalization, \nu would be required. Line 63: "we shall assume that Dˆ is a closed (i.e. compact with no boundary) domain" Do you mean contains its boundary? Line 64: Do these assumptions on \phi hold for other common activation functions? Tanh, erf, etc. Line 150: "kernerl" Figure 1: "chosen teacher's neuron" How is the neuron chosen?


Review 2

Summary and Contributions: The paper provides CLT-like results for the gradient-flow optimization of shallow (two-layer) neural networks in the infinite-width mean-field limit. First, existence of solutions in the (function norm-) regularized problem is established. Then Wasserstein gradient flows are considered to show (1) how a function differentiable wrt the parameter flow evolves wrt the discrepancy measure between the empirical and true measures of the parameters; and (2) how the difference between the empirical (n-neuron) shallow-network function and the true function evolves, with reference to a “resampled” difference (where the parameters of the empirical measure are resampled for all times t). Finally, an asymptotic second-moment bound reflective of Monte Carlo sampling error is provided for this difference. Experimental results illustrate some of the quantities discussed.

Strengths: Characterizing discrepancies from the mean-field limit as a function of number of neurons n is a pertinent question given the recent study of shallow-but-wide networks. The most interesting parts of this work are Propositions 3.1 and 3.2 (respectively “dynamical CLT” (1) and (2) above) and the interpretations provided: for Prop 3.1, the term that reflects lack of parameter resampling in the empirical measure; for Prop 3.2, the Volterra kernel as times s, t go to infinity. The 1/sqrt{n} scaling in these results justify their being called central limit theorems.

Weaknesses: Proposition 2.1 is tangential and not new in content or proof technique; much the same was shown in, eg, [Mei, Misiakiewicz, Montanari ‘19] and other works building upon it. The proofs of Propositions 3.1 and 3.2, the most meaningful results, are simple calculations making use of the Mean Value Theorem and Duhamel’s principle, respectively. Theorem 3.3 is a lot of work for what is not a particularly interesting result: it is asymptotic in both n and t, so yields no insight into dynamics, or any relationship between n and t. Moreover it is not truly dimension-free as the authors claim; dimension implicitly shows up in the moments of f, given that \psi is positively homogeneous, so it is rather the case that dimension shows up how one might expect in a variance bound. Moreover (and this is made clearer upon examination of experimental results), it is not useful to reason about optimization time (finite or asymptotic) without reference to a discretization scheme. The experiments make reference to “epochs”, but there is no optimization algorithm to relate to the flows discussed in the theory. Discretization of the results with appropriate step size and associated error bounds would be nontrivial additional work. The experiments as a whole also seem to raise more questions than they answer: why the greater decay of fluctuations for smaller n after many epochs? Why the alignment or lack thereof with teacher network total variation or 2-norm? Why the non-monotonic behavior of some plots?

Correctness: At this level of review, the paper generally appears correct. Claims of dimension-independence and novelty (Prop 2.1) should be reconsidered. Due to length and organization, the proof of Theorem 3.3 was not examined.

Clarity: The paper is generally well written. Some choices could make results easier to appreciate, such as putting the non-differential form of the quantity T_t(\theta) (from Appendix (61)) in the statement of Prop 3.1, making clear what the quantity actually is. Longer proofs could be revised for clarity and their organization made more obvious.

Relation to Prior Work: Could be improved; see above.

Reproducibility: Yes

Additional Feedback: [post-author feedback] My score remains unchanged because the asymptotic analysis leaves open a number of questions that cannot be answered unless the analysis is refined. My questions re: the plots were meant to point this out: there is a specific relationship (perhaps characterized by distinct regimes) between n and t that has not been spelled out; Theorem 3.3, by not guaranteeing monotonicity or indeed anything beyond the limiting case, does not yield even crude insight into behavior at (finite) timescales of interest. The extent to which the proof of Theorem 3.3 is opaque also remains a concern.


Review 3

Summary and Contributions: This paper studies the difference between the gradient descent dynamics on a finite-width two-layer network and its infinite-width limit, in the mean-field setting. The fluctuation is first characterized at a given time, and then bounded as the time goes to infinity, assuming that the infinite-width gradient flow converges to a global minimizer fast enough.

Strengths: The difference between the training dynamics of finite-width and infinite-width networks is an important quantity to study: it can help us understand how wide the network should be to ensure global convergence.

Weaknesses: In the main result Theorem 3.2, it is unclear whether we can interchange the lim sup and lim; if not, then we may not have a convergence result even for a wide network. If it is hard to show, can we replace n with perhaps n^a for some 0<a<1? Moreover, Theorem 3.2 makes some assumptions on the convergence rate of the infinite-width gradient flow (eq (71) and eq (131)); it would be better to have some concrete examples where such assumptions hold.

Correctness: I do not see any correctness issue.

Clarity: There are a few places that are confusing. 1. On line 63, it says "closed (i.e. compact with no boundary) domain"; I guess it should just be compact? 2. Since Proposition 2.1 and Theorem 3.3 are stated in formal environments, I think it is better to include all the assumptions. 3. It would be better to explain the quantities and proof ideas a bit more. For example, does T_t defined in eq (25) correspond to changes in the kernel? Is it controlled using smoothness and convergence of mu_t?

Relation to Prior Work: The discussion of prior work is thorough as far as I know.

Reproducibility: Yes

Additional Feedback: Reply to the feedback: Thanks for the response! I think the assumptions on the convergence rates (eq. (71) and (131)) might be reasonable in the current setting, but since they look important in the proofs, they should be mentioned and discussed in the body of the paper. I also find the overall writing too technical; it would be helpful to highlight the key and new ideas.


Review 4

Summary and Contributions: The authors analyze the so-called mean-field dynamics of neural networks and prove that the deviations from the predicted evolution remain bounded throughout training in the asymptotic limit. This allows to obtain estimate of Monte-Carlo type resampling errors, with its to-be-expected 1/n^1/2 behavior.

Strengths: The paper provides a refinement of the analysis of refs [17, 7, 19, 22] and provide Monte-Carlo type control on the mean-field dynamics,

Weaknesses: While the authors studied the deviation of neural networks from their infinite width limit, the setting is still asymptotic in both particule and times, and finite size correction beyond that would be welcome.

Correctness: As far are the reviewer can tell, the result are correct, and intuitively to be expected.

Clarity: The paper is fairly well written.

Relation to Prior Work: It seems to the reviewer that the appropriate literature is cited.

Reproducibility: Yes

Additional Feedback: All in all, this looks like a solid, if not surprising, contribution.

[Author Response · NeurIPS 2020]

We thank the reviewers for their helpful feedback. The reviews emphasized the significance of our results for the
analysis of finite-width neural networks (**R1**, **R3**) and the strength of our technical contributions (**R1**, **R2**). One primary
concern was that our results are only asymptotic. We address this concern and other questions below.

**1. Theory**

**R2** & **R4**: *Being asymptotic in both $n$ and $t$, Theorem 3.3 is not very interesting.* We agree that nonasymptotic bounds
would be highly desirable, but such results are difficult and have only been established in limited settings that are not
applicable to neural network training (e.g. weak interactions in [1]) beyond the *lazy* regime. Despite being asymptotic
in $n$ (just like the classical CLT), our results give insights on the evolution of the deviations from mean-field limit and
set the groundwork for nonasymptotic results. In particular, they reveal a benign dependency on ambient dimension
(see point below). Regarding the dependence on $t$, existing bounds on the fluctuations (cf. [2]) grow exponentially in $t$,
whereas we show a finite bound as $t \to \infty$. We also discuss the decay of the fluctuations at finite $t$ in the first remark
under Sec. 2 below. Finally, we check the validity of our results in numerical experiments under finite $n$ and $t$.

**R1** & **R3**: *Assumption on $\hat{D}$.* We assume it is a *closed manifold* – a compact manifold without boundary – like a sphere.
**R1**: *Do the assumptions on activations hold for* $\tanh$ *and* $\mathrm{erf}$? Yes. The only requirements are the Universal
Approximation Theorem and our smoothness assumptions in Sec. 2.1 in the paper.
**R2**: *Is Prop. 2.1 tangential?* Not really - it shows by adding the regularization we can control the 2-norm of the loss
minimizer (consistent with Col. 1 & 5, Row 2 in Fig. 1), on which the fluctuation bound in Thm. 3.3 crucially depends.
**R2**: *The bound in Theorem 3.3 is not truly dimension free.* Our bound depends on the norm of the target function
in the variation-norm space introduced in Sec. 2.1. As the reviewer points out, the dependency on dimension is
implicit through this norm, which is shown in [3] to depend polynomially in the dimension for functions with hidden
dependency on low-dim structures.
**R2**: *Results under time discretization are lacking.* Indeed, we leave this for future work, though our continuous-time
analysis already yields insights. Moreover, we see from Col. 3 of Fig. 1 that the loss evolutions under gradient descent
nicely agree across different $n$, showing empirical consistency of the discretization scheme.
**R3**: *Are $\limsup_{t \to \infty}$ and $\lim_{n \to \infty}$ exchangeable in Thm. 3.3? How about changing $n$ to $n^a$ for $a \in (0, 1)$?* Not in our
result (including in the latter case), unfortunately, as the $O(n^{-1})$ scaling of the fluctuations in $n$ at finite time (thanks to
CLT and the continuity of the flow) may not be preserved at the $t \to \infty$ limit. This is worthy of future investigations.
**R3**: *Concrete examples for assumptions (71) and (131).* We show in C.1.1 and C.2.1 that (71) and (131) can be satisfied
if the flow $\mathbf{\Theta}_t(\boldsymbol{\theta})$ converges at a uniform rate of $O(t^{-\alpha})$ with $\alpha > 2$ and $\alpha > \frac{3}{2}$, respectively. Also, an alternative to
(71) is $\int_0^\infty (\mathcal{L}(\mu_t))^{1/2} dt < \infty$, and so a sufficient condition is for the loss value to decrease faster than $O(t^{-2})$.
**R3**: *More explanations of $\boldsymbol{T}_t$.* We will add: $\forall \boldsymbol{\theta} \in D$, $\boldsymbol{T}_t(\boldsymbol{\theta})$ captures the deviation of the flow $\mathbf{\Theta}_t(\boldsymbol{\theta})$ due to the "initial
deviation" $\omega_0$, i.e. $\boldsymbol{T}_t = \lim_{n \to \infty} n^{1/2}(\mathbf{\Theta}_t^{(n)} - \mathbf{\Theta}_t)$. $\boldsymbol{T}_t$ satisfies an infinite-dim. linear ODE, (86). To control it, we
show that 1) the asymptotic linear operator is PSD; 2) the source term lies in the range of the linear operator; and 3)
finite time perturbations are controlled.

**2. Experimental results**

**R1**: *Why do the average fluctuations decay with $t$?* While Thm. 3.3 only speaks about the $t \to \infty$ limit,
we can study the long-term behavior of $\mathbb{E}_0 \|g_t\|_{\hat{\nu}}$ by analyzing the $t$-asymptotic version of (25) or (86),
$\dot{\boldsymbol{T}}_t = -(\mathcal{A}_\infty^{(K)} + \mathcal{A}_\infty^{(V)})\boldsymbol{T}_t + \boldsymbol{b}_\infty$. (Note that this also describes the exact dynamics of the fluctuations if we
set $\mu_0 = \mu_\infty$.) In the unregularized case, we expect that $f_\infty$ interpolates the data, and hence $\mathcal{A}_\infty^{(V)} = 0$. Thus,
the solution to above is $\boldsymbol{T}_t = (1 - e^{-t\mathcal{A}_\infty^{(K)}})(\mathcal{A}_\infty^{(K)})^\dagger \boldsymbol{b}_\infty$. Also, in the ERM setting, $\mathcal{A}_t^{(K)}$ is a PSD operator with
finitely many nonzero eigenvalues, and hence its nonzero eigenspaces are spanned by eigenfunctions $v_1, ..., v_k$
associated with eigenvalues $\lambda_1, ..., \lambda_k > 0$. By Lemma C.3, we can express $\boldsymbol{b}_\infty = \sum_{i=1}^k c_i v_i$. Using (26), we
get $\|g_t\|_\nu^2 = \|\bar{g}_\infty\|_\nu^2 - \langle \boldsymbol{b}_\infty, (I - e^{-2t\mathcal{A}_\infty^{(K)}})(\mathcal{A}_\infty^{(K)})^\dagger \boldsymbol{b}_\infty \rangle = \|\bar{g}_\infty\|_\nu^2 - \sum_{i=1}^k \lambda_j^{-1} (1 - e^{-2\lambda_j t}) c_i^2$, which decreases
monotonically in $t$. This is consistent with the long-term behavior of the fluctuations in Col. 2, Rows 1 & 3 in Fig. 1.
**R2**: *Why do the fluctuations decrease faster for smaller $n$?* This is due to finite-$n$ effects in the fluctuation dynamics,
which are captured in (63) but not (25), and which decrease as $n$ grows.
**R2**: *Why the alignment or lack thereof with teacher network's TV- and 2-norm in Fig. 1?* When (and only when)
regularized, both the TV- and 2-norm of the student are controlled by those of the teacher, consistent with Prop. 2.1.
**R2**: *Why are some of the behaviors plotted in Fig. 1 non-monotonic?* Thm. 3.3 does not guarantee monotonic decay of
the fluctuations during finite time, but only prescribes its behavior as $t \to \infty$. The calculations above of the fluctuation
decay is also an asymptotic analysis. As for the norms, their non-monotonic evolution indicates that the regularization's
effect become relatively stronger later in training, when the function reconstruction loss is low.
**R1**: *In Col. 3 of Fig. 1, why is the loss independent from width?* We first note that Col. 3 plots the training / population
loss, as the study of generalization is beyond the scope of this paper (which is why we don't distinguish between $\nu$ and
$\hat{\nu}$). The good agreement of loss evolution for different $n$ validates empirically the convergence to a mean-fields solution.
**R1**: *How do we choose the teacher's neuron in Fig. 1?* We chose the teacher's neurons to have $c = 1$ and $\boldsymbol{z}$ randomly
sampled on the hypersphere. For the plots on Col. 1, we chose one of the neurons as the "marker".

58

[1] Durmus et al. "An Elementary...". [2] Mei, Misiakiewicz, Montanari, "Mean-field...". [3] Bach, "Breaking...".


[Meta-Review · NeurIPS 2020]

The paper provides CLT-like results for the dynamics of single-hidden layer, wide neural networks in the mean-field limit. The authors also show that under certain conditions the long-time fluctuations can be controlled with an MC type resampling error. The reviewers had a positive assessment of the finite width analysis and the strength of some of the technical contributions. They did however raise a variety of concerns regarding the asymptotic nature of results (both in n and t), assumptions on Dhat, and lack of results with discretization. While some of these concerns were alleviated based on the authors’ response, the more critical reviewers maintained their score and one positive reviewer slightly decreased theirs from 8 to 7. I agree with the reviewers that CLT type results for finite width is indeed interesting. In particular, compared with prior work such as [16] the dependence in time in their results is not exponential. I also agree with R2 that “the asymptotic analysis leaves open a number of questions that cannot be answered unless the analysis is refined”. Given the other papers I am handling I believe this paper is slightly above the acceptance threshold. I strongly urge the authors to address the excellent points raised by the reviewers in their final manuscript.